# *ShEPhERD*: DIFFUSING SHAPE, ELECTROSTATICS, AND PHARMACOPHORES FOR BIOISOSTERIC DRUG DESIGN

**Keir Adams**\*, **Kento Abeywardane**\*, **Jenna Fromer, & Connor W. Coley**
Massachusetts Institute of Technology, Cambridge, MA 02139, USA
{keir,kento,jfromer,ccoley}@mit.edu

## ABSTRACT

Engineering molecules to exhibit precise 3D intermolecular interactions with their environment forms the basis of chemical design. In ligand-based drug design, bioisosteric analogues of known bioactive hits are often identified by virtually screening chemical libraries with shape, electrostatic, and pharmacophore similarity scoring functions. We instead hypothesize that a generative model which learns the joint distribution over 3D molecular structures and their interaction profiles may facilitate 3D interaction-aware chemical design. We specifically design *ShEPhERD*[1], an SE(3)-equivariant diffusion model which jointly diffuses/denoises 3D molecular graphs and representations of their shapes, electrostatic potential surfaces, and (directional) pharmacophores to/from Gaussian noise. Inspired by traditional ligand discovery, we compose 3D similarity scoring functions to assess *ShEPhERD*'s ability to conditionally generate novel molecules with desired interaction profiles. We demonstrate *ShEPhERD*'s potential for impact via exemplary drug design tasks including natural product ligand hopping, protein-blind bioactive hit diversification, and bioisosteric fragment merging.

## 1 INTRODUCTION

Designing new molecules to attain specific functions via physicochemical interactions with their environment is a foundational task across the chemical sciences. For instance, early-stage drug discovery often involves tuning the 3D shape, electrostatic potential (ESP) surface, and non-covalent interactions of small-molecule ligands to promote selective binding to a protein target (Bissantz et al., 2010; Huggins et al., 2012). Homogeneous catalyst design requires developing organometallic, organic, or even peptidic catalysts that stabilize reactive transition states via specific noncovalent interactions (Raynal et al., 2014; Fanourakis et al., 2020; Knowles & Jacobsen, 2010; Wheeler et al., 2016; Toste et al., 2017; Metrano & Miller, 2018). Supramolecular chemistry similarly optimizes host-guest interactions for applications across photoresponsive materials design, biomedicine, and structure-directed zeolite synthesis (Qu et al., 2015; Stoffelen & Huskens, 2016; Corma et al., 2004).

This essential challenge of designing molecules with targeted 3D interactions manifests across myriad tasks in ligand-based drug design. In medicinal chemistry, bioisosteric scaffold hopping aims to swap-out substructures within a larger molecule while preserving bioactivity (Langdon et al., 2010). Often, the swapped scaffolds share biochemically-relevant features such as electrostatics or pharmacophores, which describe both non-directional (hydrophobic, ionic) and directional (hydrogen bond acceptor/donor, aromatic $\pi$-$\pi$, halogen bonding) non-covalent interactions. When scaffold hopping extends to entire ligands, "ligand hopping" can help identify synthesizable analogues of complex natural products that mimic their 3D biochemical interactions (Grisoni et al., 2018). In hit expansion, ligand hopping is also used to diversify known bioactive hits by proposing alternative actives, ranging from topologically-similar "me-too" compounds to distinctly new chemotypes (Schneider et al., 2006; Wermuth, 2006). Notably, ligand hopping *does not require* knowledge of the protein target. Lastly, bioisosteric scaffold hopping extends beyond altering individual molecules; Wills et al. (2024) used "bioisosteric fragment merging" to replace a *set* of fragments that independently bind a protein with one ligand that captures the fragments' aggregate 3D binding interactions.

---

\*These authors contributed equally
[1]*ShEPhERD*: **Sh**ape, **E**lectrostatics, and **Ph**armacophore **E**xplicit **R**epresentation **D**iffusion

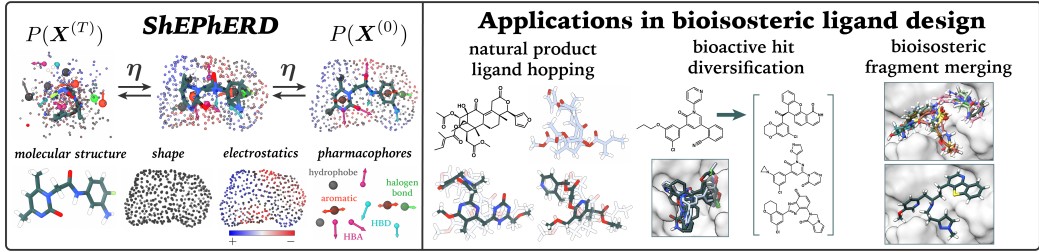

Figure 1: We introduce *ShEPhERD*, a diffusion model that jointly generates 3D molecules and their shapes, electrostatics, and pharmacophores. By explicitly modeling 3D molecular interactions, *ShEPhERD* can be applied across myriad challenging ligand-based drug design tasks including natural product ligand hopping, bioactive hit diversification, and bioisosteric fragment merging.

Here, we consider three archetypal tasks in bioisosteric drug design: 3D similarity-constrained ligand hopping, protein-blind bioactive hit diversification, and bioisosteric fragment merging (Fig. 1). The unifying theme across these tasks is identifying new molecular structures which are chemically dissimilar from known matter in terms of their molecular graphs, but which are highly similar with respect to their 3D intermolecular interaction profiles. To find such bioisosteric analogues, traditional design campaigns will virtually screen chemical libraries with 3D similarity scoring functions to query molecules' 3D shape, electrostatic, and/or pharmacophore similarity with respect to a reference molecule (Oprea & Matter, 2004; Rush et al., 2005; Zavodszky et al., 2009; Sanders et al., 2012). However, similarity-based virtual screening has acute drawbacks: it cannot explore beyond known chemical libraries, it is inefficient by virtue of being undirected, and it can be quite slow when multiple geometries of conformationally-flexible molecules must be scored.

We instead develop a broadly applicable generative modeling framework that enables efficient 3D bioisosteric molecular design. We are motivated by multiple observations: (1) As elaborated above, ligand-based drug discovery requires designing novel molecular structures that form specific 3D interactions. (2) Harris et al. (2023) have found that 3D generative models for *structure-based* drug design struggle to generate ligands that form biochemical interactions (e.g., hydrogen bonds) with the protein, despite training on protein-ligand complexes. This suggests the need for new strategies which explicitly model molecular interactions. (3) Numerous chemical design scenarios beyond drug design require engineering the physicochemical interactions of small molecules. But, such settings are often data-restricted, necessitating zero/few-shot generative approaches. To address these challenges, we introduce *ShEPhERD*, a 3D generative model which learns the relationship between the 3D chemical structures of small molecules and their shapes, electrostatics, and pharmacophores (henceforth collectively called "interaction profiles") in *context-free* environments. Specifically:

- We define explicit point cloud-based representations of molecular shapes, ESP surfaces, and pharmacophores that are amenable to symmetry-preserving SE(3)-equivariant diffusion modeling.

- We formulate a joint denoising diffusion probabilistic model (DDPM) that learns the joint distribution over 3D molecules (atom types, bond types, coordinates) and their 3D shapes, ESP surfaces, and pharmacophores. In addition to diffusing/denoising attributed point clouds for shape and electrostatics, we natively model the directionality of pharmacophores by diffusing/denoising vectors on the unit sphere. We sample from specific interaction-conditioned distributions via inpainting.

- Inspired by virtual screening, we craft shape, electrostatic, and pharmacophore similarity functions to score (1) the self-consistency of jointly generated molecules and interaction profiles, and (2) the similarity between conditionally generated molecules and target interaction profiles. We show that *ShEPhERD* can generate diverse 3D molecular structures with substantially enriched interaction-similarity to target profiles, even upon geometric relaxation with semi-empirical DFT.

- After training on drug-like datasets, we demonstrate *in silico* that *ShEPhERD* can facilely design small-molecule mimics of natural products via ligand hopping, diversify bioactive hits while preserving protein-binding modes, and merge fragments into bioisosteric ligands, all out-of-the-box.

We anticipate that *ShEPhERD* will prove immediately useful for ligand-based drug design campaigns that require the *de novo* design of new molecules with precise 3D interactions. However, we stress *ShEPhERD*'s general applicability to other areas of interaction-aware molecular design, such as organocatalyst design. We especially envision that *ShEPhERD* will be extended to model other structural characteristics beyond the shape, electrostatic, and pharmacophore profiles treated here.

## 2    RELATED WORK

**3D similarity scoring for ligand-based drug design.** Ligand-based drug design commonly applies shape, electrostatic, and/or pharmacophore similarity scoring functions to screen for molecules with similar 3D interactions as a known bioactive molecule (Fiedler et al., 2019; Rush et al., 2005; Jackson et al., 2022). Shape similarity functions typically use atom-centered Gaussians to compute the volumetric overlap between active and query molecules (Grant & Pickup, 1995; Grant et al., 1996). Many methods additionally attribute scalar (e.g., charge) or categorical (e.g., pharmacophore type) features to these Gaussians in order to score electrostatic (Good et al., 1992; Bolcato et al., 2022; OpenEye, Cadence Molecular Sciences, 2024) or pharmacophore similarity (Sprague, 1995; Dixon et al., 2006; Taminau et al., 2008; Wahl, 2024). To better capture intermolecular interactions, other methods quantify the Coulombic or pharmacophoric "potential" felt by a chemical probe at points near the molecule's surface (Cheeseright et al., 2006; Vainio et al., 2009; Cleves et al., 2019). We similarly use surface representations of molecular shape and electrostatic interactions, but combine this with volumetric point-cloud *and* vector representations of pharmacophores to model directional interactions such as hydrogen bonding and non-covalent $\pi$-effects (Wahl, 2024). Notably, we develop our own 3D similarity scoring functions which natively operate on our chosen representations.

**Symmetry-preserving generation of molecules in 3D.** Many generative models for small molecules have been developed to sample chemical space (Gómez-Bombarelli et al., 2018; Segler et al., 2018; Jin et al., 2018; Vignac et al., 2022; Bilodeau et al., 2022; Anstine & Isayev, 2023). Our work is most related to 3D approaches which directly generate molecules in specific conformations. These works usually preserve translational and rotational symmetries of atomistic systems by generating structures with E(3)- or SE(3)-invariant internal coordinates (Gebauer et al., 2022; Luo & Ji, 2022; Roney et al., 2022), or more recently, by generating Euclidean coordinates with equivariant networks (Vignac et al., 2023; Peng et al., 2023; Irwin et al., 2024). We generate molecular structures with an equivariant DDPM (Ho et al., 2020; Hoogeboom et al., 2022), a strategy that has seen use in protein-conditioned ligand and linker design (Igashov et al., 2024; Schneuing et al., 2022). Besides molecular structures, we jointly diffuse/denoise explicit representations of the molecule's shape, ESP surface, and pharmacophores with their vector directions. To do so, we employ spherical harmonics-based SE(3)-equivariant Euclidean Neural Networks (E3NNs) (Geiger & Smidt, 2022) to encode/decode coordinates, vectors, and scalar features across our heterogeneous representations.

**3D interaction-aware molecular generation.** Prior works have partially explored shape- and pharmacophore-aware molecular generative design. Multiple methods generate either (1D) SMILES or (2D) molecular graphs conditioned on 3D representations of target shapes and/or pharmacophores (Skalic et al., 2019; Imrie et al., 2021; Zhu et al., 2023; Xie et al., 2024), or use 3D similarity scores to fine-tune unconditional generative models (Papadopoulos et al., 2021; Neeser et al., 2023), which we include as a baseline (App. A.1). Since these generative models do not directly predict 3D structures, they require conformer generation as a post-processing step. On the other hand, Adams & Coley (2022) found that generating structures natively in 3D yields more chemically diverse molecules with higher 3D shape-similarity compared to a competing shape-conditioned 1D approach. Multiple other methods have since been developed for shape-conditioned 3D generation (Chen et al., 2023; Lin et al., 2024; Le et al., 2024). Regarding pharmacophores, Ziv et al. (2024) applied a pretrained DDPM to inpaint 3D molecules given fixed N and O atoms, constrained to be hydrogen bond donors and acceptors, respectively. Yet, they neglect other HBD/HBA definitions and ignore important non-covalent interactions like aromatic $\pi$-$\pi$ interactions, hydrophobic effects, and halogen bonding. Electrostatics-aware molecular generation has been considered only by Bolcato et al. (2022), via exchanging pre-enumerated chemical fragments with similar electrostatics. But, they do not conditionally generate 3D molecules given a global ESP surface. Our work *unifies* and *extends* prior work on interaction-aware 3D generative design by comprehensively modeling shape, electrostatics, and arbitrary pharmacophores (including their directionality) in a single general framework.

**3D generative structure-based drug design (SBDD).** Tangential to our work are models that generate ligands inside a protein pocket by training on protein-ligand complexes with (Lee et al., 2024; Sako et al., 2024; Zhung et al., 2024; Wang et al., 2022; Huang et al., 2024) or without (Peng et al., 2022; Schneuing et al., 2022; Guan et al., 2023) explicit encodings of their interactions. In contrast, we consider context-free molecular design. By modeling the interaction profiles of ligands *only*, *ShEPhERD* is more general than (yet still applicable to) SBDD. We also retain the freedom to train on arbitrary chemical spaces that may be larger, denser, or more diverse than the ligands in the PDB.

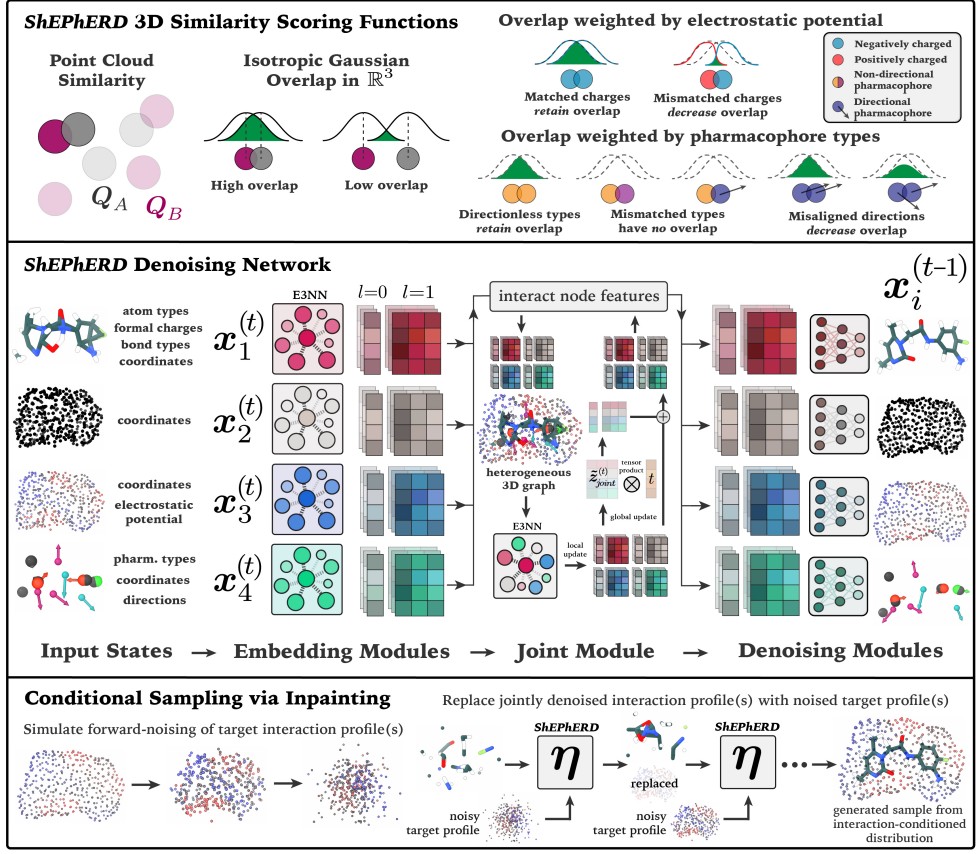

Figure 2: **(Top)** Visualization of our 3D point-cloud similarity scoring functions, which evaluate surface/shape, electrostatic, and pharmacophore similarity via weighted Gaussian overlaps. **(Middle)** *ShEPhERD*'s denoising network architecture, which uses SE(3)-equivariant neural networks to (1) embed noisy input 3D molecules and their interaction profiles into scalar and vector node features, (2) jointly interact the node features, and (3) denoise the input states. **(Bottom)** *ShEPhERD* uses inpainting to sample chemically diverse 3D molecules that exhibit desired interaction profiles.

## 3 METHODOLOGY

We seek to sample chemically diverse molecular structures from 3D similarity-constrained chemical space, where the constraints are defined by the 3D shape, electrostatics, and/or pharmacophores ("interaction profiles") of a reference 3D molecular system. To do so, we develop *ShEPhERD*, a DDPM that learns the *joint* distribution over 3D molecular graphs and their interaction profiles. At inference, we sample from specific interaction-conditioned distributions over 3D chemical space via *inpainting*. Below, we formally define our chosen representations for 3D molecules and their interaction profiles, and also define the 3D similarity scoring functions used in our evaluations. We then detail our joint DDPM and sampling protocols. Fig. 2 visualizes our overall methodology.

### 3.1 DEFINING REPRESENTATIONS OF MOLECULES AND THEIR INTERACTION PROFILES

**3D molecular graph.** We define an organic molecule with $n_1$ atoms in a specific conformation as a 3D molecular graph $\boldsymbol{x}_1 = (\boldsymbol{a}, \boldsymbol{f}, \boldsymbol{C}, \boldsymbol{B})$ where $\boldsymbol{a} \in \{\text{H}, \text{C}, \text{O}, ...\}^{n_1}$ lists the atom types ($N_a$ options); $\boldsymbol{f} \in \{-2, -1, 0, 1, 2\}^{n_1}$ specifies the formal charge of each atom; $\boldsymbol{C} \in \mathbb{R}^{n_1 \times 3}$ specifies the atomic coordinates; and $\boldsymbol{B} \in \{0, 1, 2, 3, 1.5\}^{n_1 \times n_1}$ is the adjacency matrix specifying the covalent bond order between pairs of atoms. For modeling purposes, the categorical variables $\boldsymbol{a}, \boldsymbol{f}, \boldsymbol{B}$ are represented as one-hot continuous features so that $\boldsymbol{a} \in \mathbb{R}^{n_1 \times N_a}$, $\boldsymbol{f} \in \mathbb{R}^{n_1 \times 5}$, and $\boldsymbol{B} \in \mathbb{R}^{n_1 \times n_1 \times 5}$.

**Shape.** We define the shape $\boldsymbol{x}_2 = \boldsymbol{S}_2 \in \mathbb{R}^{n_2 \times 3}$ as a point cloud with $n_2$ points sampled from the solvent-accessible surface of $\boldsymbol{x}_1$. $n_2$ is fixed regardless of $n_1$. We use *surface* points to better decouple the shape representation from the molecule's exact coordinates or 2D graph (App. A.5).

**Electrostatic potential (ESP) surface.** We represent the ESP surface $\boldsymbol{x}_3 = (\boldsymbol{S}_3, \boldsymbol{v})$ by the Coulombic potential at each point in a surface point cloud $\boldsymbol{S}_3 \in \mathbb{R}^{n_3 \times 3}$, with $\boldsymbol{v} \in \mathbb{R}^{n_3}$. We compute $\boldsymbol{v}$ from the per-atom partial charges of $\boldsymbol{x}_1$, which are computed via semi-empirical DFT (App. A.5).

**Pharmacophores.** The set of $n_4$ pharmacophores are represented as $\boldsymbol{x}_4 = (\boldsymbol{p}, \boldsymbol{P}, \boldsymbol{V})$ where $\boldsymbol{p} \in \mathbb{R}^{n_4 \times N_p}$ is a matrix of one-hot encodings of the $N_p$ pharmacophore types; $\boldsymbol{P} \in \mathbb{R}^{n_4 \times 3}$ is the pharmacophores' coordinates; and $\boldsymbol{V} \in \{\mathbb{S}^2, \boldsymbol{0}\}^{n_4}$ contains the relative unit or zero vectors specifying the directionality of each pharmacophore. The directional pharmacophores include hydrogen bond acceptors (HBA) and donors (HBD), aromatic rings, and halogen-carbon bonds. Directionless pharmacophores ($\boldsymbol{V}[k] = \boldsymbol{0}$) include hydrophobes, anions, cations, and zinc binders. App. A.5 details how we extract pharmacophores from a molecule by adapting known SMARTS patterns.

### 3.2 SHAPE, ELECTROSTATIC, AND PHARMACOPHORE SIMILARITY SCORING FUNCTIONS

To formulate our scoring functions, we first define a point cloud $\boldsymbol{Q} \in \mathbb{R}^{n_Q \times 3}$ where each point $\boldsymbol{r}_k$ is assigned an isotropic Gaussian in $\mathbb{R}^3$ (Grant & Pickup, 1995; Grant et al., 1996). We measure the Tanimoto similarity sim $\in [0, 1]$ between two point clouds $\boldsymbol{Q}_A$ and $\boldsymbol{Q}_B$ using first order Gaussian overlap $O_{A,B} = \sum_{a \in \boldsymbol{Q}_A} \sum_{b \in \boldsymbol{Q}_B} w_{a,b} (\frac{\pi}{2\alpha})^{\frac{3}{2}} \exp\left(-\frac{\alpha}{2} \|\boldsymbol{r}_a - \boldsymbol{r}_b\|^2\right)$ and $\text{sim}^*(\boldsymbol{Q}_A, \boldsymbol{Q}_B) = \frac{O_{A,B}}{O_{A,A} + O_{B,B} - O_{A,B}}$ where $\alpha$ is a Gaussian width and $w_{a,b}$ is a weighting factor. Note that 3D similarities are sensitive to SE(3) transformations of $\boldsymbol{Q}_A$ with respect to $\boldsymbol{Q}_B$. We characterize the similarity at their optimal alignment by $\text{sim}(\boldsymbol{Q}_A, \boldsymbol{Q}_B) = \max_{\boldsymbol{R}, \boldsymbol{t}} \text{sim}^*(\boldsymbol{R}\boldsymbol{Q}_A^T + \boldsymbol{t}, \boldsymbol{Q}_B)$ where $\boldsymbol{R} \in SO(3)$ and $\boldsymbol{t} \in T(3)$. We align using automatic differentiation. Note that $n_{Q_A}$ need not equal $n_{Q_B}$.

**Shape scoring.** The *volumetric* similarity between two atomic point clouds $\boldsymbol{C}_A$ and $\boldsymbol{C}_B$ is defined as $\text{sim}^*_{\text{vol}}(\boldsymbol{C}_A, \boldsymbol{C}_B)$ with $w_{a,b} = 2.7$ and $\alpha = 0.81$ (Adams & Coley, 2022). We newly define the *surface* similarity between two surfaces $\boldsymbol{S}_A$ and $\boldsymbol{S}_B$ as $\text{sim}^*_{\text{surf}}(\boldsymbol{S}_A, \boldsymbol{S}_B)$ with $w_{a,b} = 1$, and $\alpha = \Psi(n_2)$. Here, $\Psi$ is a function fitted to $\text{sim}^*_{\text{vol}}$ depending on the choice of $n_2$ (App. A.6).

**Electrostatic scoring.** We define the similarity between two electrostatic potential surfaces $\boldsymbol{x}_{3,A}$ and $\boldsymbol{x}_{3,B}$ as $\text{sim}^*_{\text{ESP}}(\boldsymbol{x}_{3,A}, \boldsymbol{x}_{3,B})$ with $w_{a,b} = \exp\left(-\frac{\|\boldsymbol{v}_A[a] - \boldsymbol{v}_B[b]\|^2}{\lambda}\right)$, $\alpha = \Psi(n_3)$, and $\lambda = \frac{0.3}{(4\pi\epsilon_0)^2}$. $\epsilon_0$ is the permittivity of vacuum with units $e^2(\text{eV} \cdot \text{Å})^{-1}$. Inspired by Hodgkin & Richards (1987) and Good (1992), we use the *difference* between electrostatic potentials to increase sensitivity to their respective magnitudes (i.e., rather than simply comparing signs).

**Pharmacophore scoring.** We define the similarity between two sets of pharmacophores $\boldsymbol{x}_{4,A}$ and $\boldsymbol{x}_{4,B}$ with $\text{sim}^*_{\text{pharm}}(\boldsymbol{x}_{4,A}, \boldsymbol{x}_{4,B}) = \frac{\sum_{m \in \mathcal{M}} O_{A,B;m}}{\sum_{m \in M} O_{A,A;m} + O_{B,B;m} - O_{A,B;m}}$. Here, $\mathcal{M}$ is the set of all pharmacophore types ($|\mathcal{M}| = N_p$). $w_{a,b} = 1$ if $m$ is non-directional, or a scaling of vector cosine similarity $w_{a,b;m} = \frac{\boldsymbol{V}[a]_m^\top \boldsymbol{V}[b]_m + 2}{3}$ if $m$ is directional (Wahl, 2024). $\alpha_m = \Omega(m)$ where $\Omega$ maps each pharmacophore type to a Gaussian width (App. A.6). We take the absolute value of $\boldsymbol{V}[a]_m^\top \boldsymbol{V}[b]_m$ for aromatic groups as we assume their $\pi$ interaction effects are symmetric across their plane.

### 3.3 JOINT DIFFUSION OF MOLECULES AND THEIR INTERACTION PROFILES WITH *ShEPhERD*

*ShEPhERD* follows the DDPM paradigm (Ho et al., 2020; Sohl-Dickstein et al., 2015) to decompose the joint distribution over the tuple $\boldsymbol{X} = (\boldsymbol{x}_1, \boldsymbol{x}_2, \boldsymbol{x}_3, \boldsymbol{x}_4)$ as $P_{\text{data}}(\boldsymbol{X}) := P(\boldsymbol{X}^{(0)}) = P(\boldsymbol{X}^{(T)}) \prod_{t=1}^{T} P(\boldsymbol{X}^{(t-1)} | \boldsymbol{X}^{(t)})$, where $P_{\text{data}}$ is the data distribution, $P(\boldsymbol{X}^{(T)})$ is (roughly) a Gaussian prior, and $P(\boldsymbol{X}^{(t-1)} | \boldsymbol{X}^{(t)})$ are Markov transition distributions learnt by a neural network. This network is trained to reverse a *forward*-noising process $P(\boldsymbol{X}^{(t)} | \boldsymbol{X}^{(t-1)}) = N(\alpha_t \boldsymbol{X}^{(t-1)}, \sigma_t^2 \boldsymbol{I})$ which gradually corrupts data $\boldsymbol{X}$ into Gaussian noise $\boldsymbol{X}^{(T)}$ according to a variance preserving noise schedule given by $\sigma_t^2$ and $\alpha_t = \sqrt{1 - \sigma_t^2}$ for $t = 1, ..., T$. See Ho et al. (2020) for preliminaries on DDPMs. Here, we describe the forward and reverse processes of *ShEPhERD*'s joint DDPM.

**Forward noising processes.** We follow Hoogeboom et al. (2022) to forward-noise the 3D molecule $\boldsymbol{x}_1 = (\boldsymbol{a}, \boldsymbol{f}, \boldsymbol{C}, \boldsymbol{B})$. For $\boldsymbol{a} \in \mathbb{R}^{n_1 \times N_a}$, we use Gaussian noising where $\boldsymbol{a}^{(t)} = \alpha_t \boldsymbol{a}^{(t-1)} + \sigma_t \boldsymbol{\epsilon}$ for $\boldsymbol{\epsilon} \sim N(\boldsymbol{0}, \boldsymbol{I})$. The processes for $\boldsymbol{f}$ and $\boldsymbol{B}$ are similar, but we symmetrize the upper/lower triangles of $\boldsymbol{B}^{(t)}$. We apply isotropic noise to $\boldsymbol{C} \in \mathbb{R}^{n_1 \times 3}$, but center the noise at $\boldsymbol{0}$ to ensure translational invariance: $\boldsymbol{C}^{(t)} = \alpha_t \boldsymbol{C}^{(t-1)} + \sigma_t (\boldsymbol{\epsilon} - \frac{1}{n_1} \sum_{k=1}^{n_1} \boldsymbol{\epsilon}[k])$ for $\boldsymbol{\epsilon}[k] \sim N(\boldsymbol{0}, \boldsymbol{I}_3)$ and $\boldsymbol{\epsilon} \in \mathbb{R}^{n_1 \times 3}$.

For the molecular shape $x_2 = S_2 \in \mathbb{R}^{n_2 \times 3}$, we also forward-noise with isotropic Gaussian noise: $S_2^{(t)} = \alpha_t S_2^{(t-1)} + \sigma_t \epsilon$ for $\epsilon \sim N(\mathbf{0}, \mathbf{I})$. We *do not* subtract the noise's center of mass (COM), though; this ensures that the model can learn to denoise $x_2$ such that it is centered with respect to $x_1$. For the ESP surface $x_3 = (S_3, v)$, we forward-noise the surface $S_3 \in \mathbb{R}^{n_3 \times 3}$ in the same manner as $S_2$. We forward-noise $v_3 \in \mathbb{R}^{n_3}$ in the typical way: $v^{(t)} = \alpha_t v^{(t-1)} + \sigma_t \epsilon$ for $\epsilon \sim N(\mathbf{0}, \mathbf{I})$.

For the pharmacophores $x_4 = (p, P, V)$, we forward-noise their types $p$ just like the atom types $a$, and their positions $P$ just like the shape $S_2$. Diffusing the vector directions $V \in \{\mathbb{S}^2, \mathbf{0}\}^{n_4}$ is complicated since some pharmacophores are directionless. To unify their treatment, we interpret the pharmacophore vectors as Euclidean points in $\mathbb{R}^3$ (e.g., only noiseless vectors have norm 1.0 or 0.0). We then forward-noise the vectors like any point cloud: $V^{(t)} = \alpha_t V^{(t-1)} + \sigma_t \epsilon$ for $\epsilon \sim N(\mathbf{0}, \mathbf{I})$.

Whereas the above processes describe the *single-step* forward transition distributions, note that we can efficiently sample noised structures given any time horizon. For instance, we may directly sample $a^{(t)} = \overline{\alpha}_t a^{(0)} + \overline{\sigma}_t \epsilon$ for $\epsilon \sim N(\mathbf{0}, \mathbf{I})$, where $\overline{\alpha}_t = \prod_{s=1}^{t} \alpha_s$ and $\overline{\sigma}_t = \sqrt{1 - \overline{\alpha}_t^2}$.

**Reverse denoising process.** Starting from any $X^{(t)}$ (but typically pure noise $X^{(T)} \sim N(\mathbf{0}, \mathbf{I})$), the DDPM iteratively denoises $X^{(t)}$ by stochastically interpolating towards a *predicted* clean structure $\hat{X}^{(t)} \approx X^{(0)} \sim P_{\text{data}}(X^{(0)}|X^{(t)})$ resembling true samples from the data distribution. We follow the DDPM formulation where rather than predicting $X^{(0)}$ directly, the network predicts the *true noise* $\hat{\epsilon}^{(t)} \approx \epsilon$ that, when applied to data $X^{(0)}$, yields $X^{(t)} = \overline{\alpha}_t X^{(0)} + \overline{\sigma}_t \epsilon$. In this case, the single-step denoising update can be derived as: $X^{(t-1)} = \frac{1}{\alpha_t} X^{(t)} - \frac{\sigma_t^2}{\alpha_t \overline{\sigma}_t} \hat{\epsilon}^{(t)} + \frac{\sigma_t \overline{\sigma}_{t-1}}{\overline{\sigma}_t} \epsilon'$, where the additional noise $\epsilon' \sim N(\mathbf{0}, \mathbf{I})$ (set to $\epsilon' = \mathbf{0}$ for $t = 1$) makes each denoising step stochastic.

*ShEPhERD* employs a single denoising network $\eta$ that is trained to jointly predict the noises $\hat{\epsilon}_1^{(t)}, \hat{\epsilon}_2^{(t)}, \hat{\epsilon}_3^{(t)}, \hat{\epsilon}_4^{(t)} = \eta(x_1^{(t)}, x_2^{(t)}, x_3^{(t)}, x_4^{(t)}, t)$, where $\hat{\epsilon}_1^{(t)} = (\hat{\epsilon}_a^{(t)}, \hat{\epsilon}_f^{(t)}, \hat{\epsilon}_C^{(t)}, \hat{\epsilon}_B^{(t)})$, $\hat{\epsilon}_2^{(t)} = (\hat{\epsilon}_{S_2}^{(t)})$, $\hat{\epsilon}_3^{(t)} = (\hat{\epsilon}_{S_3}^{(t)}, \hat{\epsilon}_v^{(t)})$, and $\hat{\epsilon}_4^{(t)} = (\hat{\epsilon}_p^{(t)}, \hat{\epsilon}_P^{(t)}, \hat{\epsilon}_V^{(t)})$. At inference, we jointly apply the denoising updates to obtain $X^{(t-1)}$. When computing $x_1^{(t-1)}$, we remove the COM from $\hat{\epsilon}_C^{(t)}$ *and* the extra noise $\epsilon'_C$.

The forward and reverse processes of our joint DDPM are designed to be straightforward to make *ShEPhERD* flexible: One may freely adjust the exact representations of the shape, electrostatics, or pharmacophores as long as they can be represented as a point-cloud with one-hot, scalar, and/or vector attributes. One can also directly model specific marginal distributions (e.g., $P(x_1, x_2)$, $P(x_1, x_3)$, or $P(x_1, x_4)$) by simply modeling a subset of the variables $\{x_2, x_3, x_4\}$. Finally, *ShEPhERD* can be easily extended to model other structural features or interaction profiles beyond those considered here by defining their explicit structural representations and forward/reverse processes.

**Denoising network design.** We design *ShEPhERD*'s denoising network $\eta$ to satisfy three criteria:

- *Symmetry-preserving*: The noise predictions $\hat{\epsilon}_1^{(t)}, \hat{\epsilon}_2^{(t)}, \hat{\epsilon}_3^{(t)}, \hat{\epsilon}_4^{(t)}$ are T(3)-invariant and SO(3)-equivariant with respect to global SE(3)-transformations of $X^{(t)}$ in order to (1) efficiently preserve molecular symmetries, and (2) ensure $x_1^{(t)}, x_2^{(t)}, x_3^{(t)}$, and $x_4^{(t)}$ remain aligned during denoising.

- *Expressive*: $\eta$ captures both local and global relationships between $x_1^{(t)}, x_2^{(t)}, x_3^{(t)}$, and $x_4^{(t)}$.

- *General*: To promote applications across chemical design, $\eta$ accommodates other definitions of shape/electrostatics/pharmacophores and permits incorporating other structural interactions, too.

To achieve these criteria, we design $\eta$ to have three components: (1) a set of *embedding modules* which equivariantly encode the heterogeneous $x_i$ into a uniform sets of latent $l{=}0$ (scalar) and $l{=}1$ (vector) node representations; (2) a *joint module* which locally and globally interacts these latent node representations; and (3) a set of *denoising modules* which predict $\hat{\epsilon}_i$ for each $x_i$.

**The embedding modules** use SE(3)-equivariant E3NNs (we choose to use expressive EquiformerV2 modules (Liao et al., 2023)) to individually encode the 3D structures of $x_1^{(t)}, x_2^{(t)}, x_3^{(t)}$, and $x_4^{(t)}$ into latent codes $(z_i^{(t)}, \tilde{z}_i^{(t)}) = \phi_i(x_i^{(t)}, t) \ \forall \ i \in [1, 2, 3, 4]$. Here, $z_i \in \mathbb{R}^{n_i \times d}$ are invariant scalar representations of each node (e.g., atom, point, or pharmacophore), and $\tilde{z}_i \in \mathbb{R}^{n_i \times 3 \times d}$ are equivariant vector representations of each node. To make each system $x_i$ sensitive to relative translations with respect to $x_1^{(t)}$, we also include a virtual node that is positioned at the center of mass of $x_1^{(t)}$, and which remains unnoised. Prior to 3D message passing with the E3NNs, scalar

atom/point/pharmacophore features (e.g., $\boldsymbol{a}^{(t)}$, $\boldsymbol{f}^{(t)}$, $\boldsymbol{v}^{(t)}$, $\boldsymbol{p}^{(t)}$) are embedded into $l{=}0$ node features, and vector features (e.g., $\boldsymbol{V}^{(t)}$) are directly assigned as $l{=}1$ node features. The pairwise bond representations $\boldsymbol{B}^{(t)}$ for $\boldsymbol{x}_1^{(t)}$ are also embedded as $l{=}0$ edge attributes. Finally, for each $\phi_i$, we embed sinusoidal positional encodings of the time step $t$ and add these to all the $l{=}0$ node embeddings.

***The joint module*** consists of two steps to *locally* and *globally* interact the joint latent variables:

(1) We collate the coordinates of $\boldsymbol{x}_1^{(t)}$, $\boldsymbol{x}_2^{(t)}$, $\boldsymbol{x}_3^{(t)}$, and $\boldsymbol{x}_4^{(t)}$ to form a heterogeneous 3D graph where the nodes are attributed with their corresponding latent features $(\boldsymbol{z}_i^{(t)}, \tilde{\boldsymbol{z}}_i^{(t)}) \; \forall \, i \in [1, 2, 3, 4]$. We then encode this heterogeneous 3D graph with another E3NN (EquiformerV2) module $\phi_{joint}^{local}$ and residually update the nodes' latent features: $(\boldsymbol{z}_i^{(t)}, \tilde{\boldsymbol{z}}_i^{(t)}) \mathrel{+}= \phi_{joint}^{local}\left((\boldsymbol{x}_i^{(t)}, \boldsymbol{z}_i^{(t)}, \tilde{\boldsymbol{z}}_i^{(t)}) \; \forall \, i \in [1, 2, 3, 4]\right)$.

(2) We sum-pool the updated $l{=}1$ node features across each sub-graph, concatenate, and then embed with an equivariant feed-forward network $\phi_{joint}^{global}$ to obtain a global $l{=}1$ code describing the overall system: $\tilde{\boldsymbol{z}}_{joint}^{(t)} = \phi_{joint}^{global}\left(\mathrm{Cat}\left[\left(\sum_{k=1}^{n_i} \tilde{\boldsymbol{z}}_i^{(t)}[k]\right) \; \forall \, i \in [1, 2, 3, 4]\right]\right)$. We then apply equivariant tensor products between $\tilde{\boldsymbol{z}}_{joint}^{(t)} \in \mathbb{R}^{d \times 3}$ and an $l{=}0$ embedding of $t$. This yields $l{=}0$ and $l{=}1$ global latent features $(\boldsymbol{z}_{joint}^{(t)}, \tilde{\boldsymbol{z}}_{joint}^{(t)})$, which are residually added to the node representations $(\boldsymbol{z}_i^{(t)}, \tilde{\boldsymbol{z}}_i^{(t)})$.

***The denoising modules*** predict the noises $\hat{\boldsymbol{\epsilon}}_i$ from the node-level features $(\boldsymbol{z}_i^{(t)}, \tilde{\boldsymbol{z}}_i^{(t)})$. For $\boldsymbol{x}_2$, $\boldsymbol{x}_3$, and $\boldsymbol{x}_4$, the scalar noises $(\hat{\boldsymbol{\epsilon}}_{\boldsymbol{v}}^{(t)}, \hat{\boldsymbol{\epsilon}}_{\boldsymbol{p}}^{(t)})$ are predicted from the corresponding $(l{=}0)$ $\boldsymbol{z}_i^{(t)}$ codes using multi-layer perceptrons (MLPs), whereas the vector noises $(\hat{\boldsymbol{\epsilon}}_{\boldsymbol{S}_2}^{(t)}, \hat{\boldsymbol{\epsilon}}_{\boldsymbol{S}_3}^{(t)}, \hat{\boldsymbol{\epsilon}}_{\boldsymbol{P}}^{(t)}, \hat{\boldsymbol{\epsilon}}_{\boldsymbol{V}}^{(t)})$ are predicted from the $(l{=}1)$ $\tilde{\boldsymbol{z}}_i^{(t)}$ codes using equivariant feed-forward networks ("E3NN-style" coordinate predictions). For $\boldsymbol{x}_1$, $\hat{\boldsymbol{\epsilon}}_{\boldsymbol{a}}^{(t)}$ and $\hat{\boldsymbol{\epsilon}}_{\boldsymbol{f}}^{(t)}$ are predicted from $\boldsymbol{z}_1^{(t)}$ with simple MLPs. $\hat{\boldsymbol{\epsilon}}_{\boldsymbol{B}}^{(t)}$ is predicted from pairs $(\boldsymbol{z}_1^{(t)}[k], \boldsymbol{z}_1^{(t)}[j])$ using a permutation-invariant MLP (App. A.7.1). $\hat{\boldsymbol{\epsilon}}_{\boldsymbol{C}}^{(t)}$ is predicted from $(\boldsymbol{z}_1^{(t)}, \tilde{\boldsymbol{z}}_1^{(t)})$ using E3NN-style *and* EGNN-style (Satorras et al., 2021) coordinate predictions (App. A.7.1).

**Training.** We train the denoising network $\boldsymbol{\eta}$ with unweighted L2 regression losses $l_i = ||\hat{\boldsymbol{\epsilon}}_i - \boldsymbol{\epsilon}_i||^2$ between the predicted and true noises. Importantly, our framework permits us to train models which directly learn certain marginal distributions. In our experiments, we train models to learn $P(\boldsymbol{x}_1)$, $P(\boldsymbol{x}_1, \boldsymbol{x}_2)$, $P(\boldsymbol{x}_1, \boldsymbol{x}_3)$, $P(\boldsymbol{x}_1, \boldsymbol{x}_4)$, and $P(\boldsymbol{x}_1, \boldsymbol{x}_3, \boldsymbol{x}_4)$. Note that since $\boldsymbol{x}_3$ defines an (attributed) surface $\boldsymbol{S}_3$, jointly modeling $(\boldsymbol{x}_2, \boldsymbol{x}_3)$ is redundant; $\boldsymbol{x}_3$ implicitly models $\boldsymbol{x}_2$. App. A.7 provides details on training protocols, choice of noise schedules, feature scaling, and model hyperparameters.

**Sampling.** For *unconditional generation*, we first sample $\boldsymbol{X}^{(T)}$ from isotropic Gaussian noise, and then denoise for $T$ steps to sample $\boldsymbol{X}^{(0)}$. We then argmax $\boldsymbol{a}^{(0)}$, $\boldsymbol{f}^{(0)}$, $\boldsymbol{B}^{(0)}$, and $\boldsymbol{p}^{(0)}$ to obtain discrete atom/bond/pharm. types, and round each $\boldsymbol{V}^{(0)}[k]$ to have norm 1.0 or 0.0. To help break the spherical symmetry of $\boldsymbol{X}^{(t)}$ at early time steps, we strategically add extra noise to the reverse process (App. A.8). For *conditional generation*, we use inpainting (Lugmayr et al., 2022; Schneuing et al., 2022) to sample $\boldsymbol{x}_1^{(0)}$ conditioned on target interaction profiles. Namely, we first simulate the forward-noising of the target profiles $(\boldsymbol{x}_2^*, \boldsymbol{x}_3^*, \boldsymbol{x}_4^*) \to (\boldsymbol{x}_2^*, \boldsymbol{x}_3^*, \boldsymbol{x}_4^*)^{(t)} \; \forall \, t \in [1, T]$. Then, during the reverse denoising process, we replace the *ShEPhERD*-denoised $(\boldsymbol{x}_2^{(t)}, \boldsymbol{x}_3^{(t)}, \boldsymbol{x}_4^{(t)})$ with the noisy target $(\boldsymbol{x}_2^*, \boldsymbol{x}_3^*, \boldsymbol{x}_4^*)^{(t)}$. Like other molecule DDPMs, we must specify $n_1$ and $n_4$ for each sample.

## 4 EXPERIMENTS

We train and evaluate *ShEPhERD* using two new datasets. Our first dataset (***ShEPhERD*-GDB17**) contains 2.8M molecules sampled from medicinally-relevant subsets of GDB17 (Ruddigkeit et al., 2012; Awale et al., 2019; Bühlmann & Reymond, 2020). Each molecule contains $\leq 17$ non-hydrogen atoms with element types in {H, C, N, O, S, F, Cl, Br, I}, and includes one conformation optimized with GFN2-xTB in the gas phase (Bannwarth et al., 2019). Our second dataset (***ShEPhERD*-MOSES-aq**) contains 1.6M drug-like molecules from MOSES (Polykovskiy et al., 2020) with up to 27 non-hydrogen atoms. Each molecule contains one conformation optimized with GFN2-xTB in implicit water. In all experiments, hydrogens are treated explicitly. Whereas we use *ShEPhERD*-GDB17 to evaluate *ShEPhERD* in straightforward unconditional and conditional generation

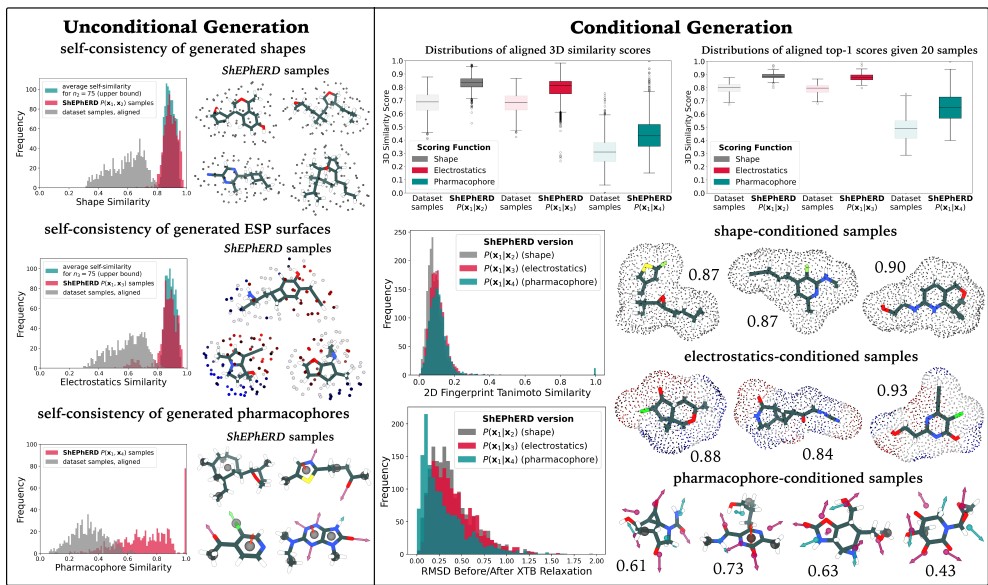

Figure 3: **(Left)** Self-consistency of jointly generated 3D molecules and their shapes, ESP surfaces, or pharmacophores, as assessed via 3D similarity between the generated *vs.* true interaction profiles of the generated molecules. Shape and ESP consistency are bounded due to randomness in surface sampling. **(Right)** Distributions of 3D interaction similarities between *ShEPhERD*-generated or dataset-sampled 3D molecules (post-relaxation and realignment) and 100 target molecules from *ShEPhERD*-GDB17, including the top-1 scores given 20 samples per target. *ShEPhERD* generates 3D molecules with low *graph* similarity to the target molecule, and which have stable geometries as measured by heavy-atom RMSD upon xTB-relaxation. Also shown are top-scoring samples overlaid on their target profiles (surfaces are upsampled for visualization), labeled with 3D similarity scores.

settings, we use *ShEPhERD*-MOSES-aq to challenge *ShEPhERD* to design drug-like analogues of natural products, to diversify bioactive hits, and to merge fragments from a fragment screen.

**Unconditional joint generation**. We first evaluate *ShEPhERD*'s ability to jointly generate 3D molecules and their interaction profiles in a self-consistent way. Namely, a well-trained model that learns the joint distribution should generate interaction profile(s) that match the true interaction profile(s) of the generated molecule. Fig. 3 reports distributions of the 3D similarity between generated and true interaction profiles across 1000 samples (with $n_1 \in [11, 60]$, $n_4 \sim P_{\text{data}}(n_4|n_1)$) from models trained on *ShEPhERD*-GDB17 to learn $P(\boldsymbol{x}_1, \boldsymbol{x}_2)$, $P(\boldsymbol{x}_1, \boldsymbol{x}_3)$, or $P(\boldsymbol{x}_1, \boldsymbol{x}_4)$.[2] When we compare against the (optimally aligned) similarities $\text{sim}_{\text{surf}}$, $\text{sim}_{\text{ESP}}$, and $\text{sim}_{\text{pharm}}$ between the *true* profiles of the generated molecules and those of random molecules from the dataset, we confirm that *ShEPhERD*'s generated profiles have substantially enriched similarities to the true profiles in all cases. Interestingly, *ShEPhERD* is more self-consistent when generating shapes or ESP surfaces than when generating pharmacophores. We partially attribute this performance disparity to the discrete nature of the pharmacophore representations and the requirement of specifying $n_4$; *ShEPhERD* occasionally generates molecules that have more true pharmacophores than generated pharmacophores, as demonstrated by the samples shown in Fig. 3 (i.e., some have missing HBAs).

**Interaction-conditioned generation**. We now evaluate *ShEPhERD*'s ability to sample from interaction-conditioned chemical space. To do so, we reuse the same $P(\boldsymbol{x}_1, \boldsymbol{x}_2)$, $P(\boldsymbol{x}_1, \boldsymbol{x}_3)$, and $P(\boldsymbol{x}_1, \boldsymbol{x}_4)$ models trained on *ShEPhERD*-GDB17, but use *inpainting* to sample from the interaction-conditioned distributions $P(\boldsymbol{x}_1|\boldsymbol{x}_2)$, $P(\boldsymbol{x}_1|\boldsymbol{x}_3)$, and $P(\boldsymbol{x}_1|\boldsymbol{x}_4)$. Specifically, we extract the true interaction profiles from 100 random target molecules (held out from training), and use *ShEPhERD* to inpaint new 3D molecular structures given these target profiles. After generating 20 structures per target profile and discarding (without replacement) any invalid structures (App. A.3) or those with

---

[2]We compute the true profiles of generated molecules *after* they are relaxed with xTB and realigned to the unrelaxed structures by minimizing heavy-atom RMSD; this avoids optimistically scoring generated molecules that are highly strained. Nonetheless, the average RMSD upon relaxation is <0.1 Å for all unconditional models trained on *ShEPhERD*-GDB17. App. A.4 reports such unconditional metrics (validity, novelty, etc.).

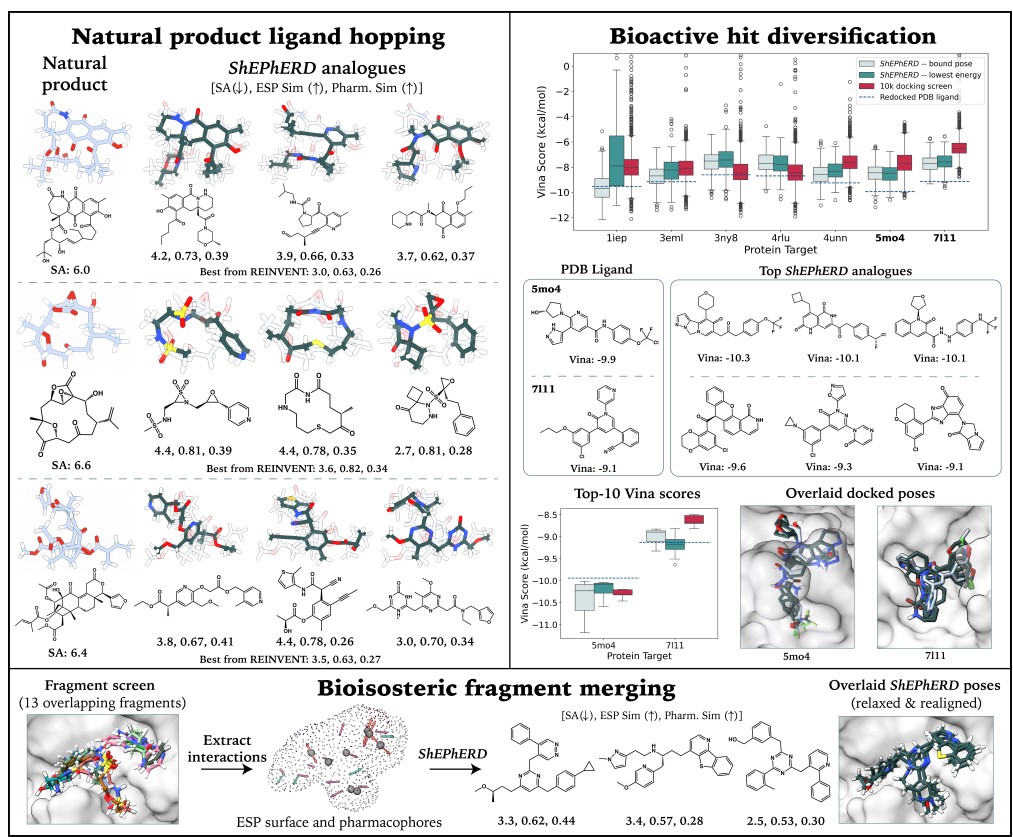

Figure 4: **(Left)** Examples of *ShEPhERD*-generated analogues of natural product targets, labeled by SA score, ESP similarity, and pharmacophore similarity to the target. Similarities are computed after xTB-relaxation and ESP-optimal realignment. **(Right)** Distributions of Vina scores for ≤500 samples from *ShEPhERD* when conditioning on the bound or lowest-energy pose of co-crystal PDB ligands across 7 proteins. We compare against the Vina scores of 10K virtually screened molecules from *ShEPhERD*-MOSES-aq. For 5mo4 and 7l11, we show top-scoring *ShEPhERD*-generated ligands (conditioned on low-energy poses), and overlay a selection from their top-10 docked poses on the PDB ligands. **(Bottom)** *ShEPhERD*'s bioisosteric fragment merging workflow. We extract the ESP surface and pharmacophores of 13 fragments from a fragment screen, and show *ShEPhERD*-generated ligands with low SA score and high 3D similarities to the fragments' interaction profiles.

2D graph similarities >0.3 to the target molecule, we relax the generated structures with xTB and compute their optimally-realigned 3D similarities to the target. Fig. 3 plots the distributions of 3D similarity scores between all valid (sample, target) pairs as well as the top-1 scores amongst the 20 samples per target. We compare against analogous similarity distributions for randomly sampled molecules from the dataset. Overall, *ShEPhERD* generates structures with very low graph similarity (≥94% of valid samples have graph similarity <0.2) but significantly enriched 3D similarities to the target, for all versions of the model. Qualitatively, *ShEPhERD* can generate molecular structures that satisfy very detailed target interactions, including multiple directional pharmacophores (Fig. 3).

**Natural product ligand hopping**. Numerous clinically-approved drugs have structures derived from natural products due to their rich skeletal complexity, high 3D character, and wide range of pharmacophores that impart uniquely selective biological function. But, the structural complexity of natural products limits their synthetic tractability. As such, designing synthetically-accessible small-molecule analogues of natural products that mimic their precise 3D interactions is a preeminent task in scaffold/ligand hopping. To imitate this task, we select three complex natural products from CO-CONUT (Sorokina et al., 2021), including two large macrocycles and a fused ring system with 9 stereocenters. We then apply *ShEPhERD* (trained to learn $P(x_1, x_3, x_4)$ on *ShEPhERD*-MOSES-aq) to generate drug-like molecules conditioned *jointly* on the ESP surface and pharmacophores of the lowest-energy conformer of each natural product, again via inpainting. We emphasize that these natural products are out-of-distribution compared to the drug-like molecules contained in *ShEP-*

*hERD*-MOSES-aq. Upon generating 2500 samples from the conditional distribution $P(\boldsymbol{x}_1|\boldsymbol{x}_3, \boldsymbol{x}_4)$ for each natural product, *ShEPhERD* identifies small-molecule mimics that attain high ESP and pharmacophore similarity to the natural products (Fig. 4), despite having much simpler chemical structures as assessed via SA score (Ertl & Schuffenhauer, 2009). Crucially, *ShEPhERD* generates molecules with higher 3D similarity compared to 2500 molecules sampled from the dataset, and compared to 10K molecules optimized by REINVENT (Blaschke et al., 2020) (App. A.1).

**Bioactive hit diversification**. Whereas stucture-based drug design aims to design high-affinity ligands given the structure of the protein target, ligand-based drug design attempts to develop and optimize bioactive hit compounds in the absence of protein information. A common task in ligand-based drug design is diversifying the chemical structures of previously identified bioactive hits (i.e., from a phenotypic experimental screen) through interaction-preserving scaffold hopping, often as a means to reduce toxicity, increase synthesizability, or evade patent restrictions. We simulate bioactive hit diversification by using *ShEPhERD* to generate analogues of 7 experimental ligands from the PDB. To evaluate their likelihood of retaining bioactivity, we use Autodock Vina (Trott & Olson, 2010; Eberhardt et al., 2021) to dock the generated ligands to their respective proteins, treating the Vina docking scores as a weak surrogate for bioactivity. To best imitate ligand-based design, we condition *ShEPhERD* on the ESP surface and pharmacophores of the *lowest-energy* conformer of each PDB ligand, rather than their bound poses (we also simulate this scenario for comparison). We then use inpainting to generate 500 samples from $P(\boldsymbol{x}_1|\boldsymbol{x}_3, \boldsymbol{x}_4)$, and dock the valid samples. Fig. 4 shows the distributions of Vina scores for *ShEPhERD*-generated analogues, compared against a docking screen of 10K random compounds from *ShEPhERD*-MOSES-aq. *Despite having no explicit knowledge of the protein targets*, *ShEPhERD* enriches Vina scores in multiple cases. For the top-scoring generated ligands for 5mo4 and 7l11, *ShEPhERD* generates substantial scaffold hops that yield diverse 2D graph structures relative to the experimental ligands. Nevertheless, upon docking, the generated molecules still explore poses that closely align with the experimental crystal poses.

**Bioisosteric fragment merging**. Fragment screening seeks to identify protein-ligand binding modes by analyzing how small chemical fragments bind to a protein of interest. Clusters of protein-bound fragment hits can then be analyzed to design high-affinity ligands. Multiple methods have been developed to *link* fragment hits into a single ligand containing the original fragments and the new linker. Recently, Wills et al. (2024) showed that *merging* (not *linking*) the fragments to form a bioisosteric ligand (which may not contain the exact fragment hits) can diversify ligand hypotheses while preserving the fragments' important binding interactions. While *ShEPhERD could* link fragments, *ShEPhERD* is uniquely suited to bioisosteric fragment merging as it can condition on *aggregate* fragment interactions. We use *ShEPhERD* to merge a set of 13 fragments experimentally identified to bind to the antiviral target EV-D68 3C protease (Lithgo et al., 2024; Wills et al., 2024). We extract $n_4 = 27$ pharmacophores by clustering common motifs and selecting interactions identified by Fragalysis (Diamond Light Source, 2024) (App. A.2). We also compute an aggregate ESP surface by sampling points from the surface of the overlaid fragments and averaging the fragments' ESP contributions at each point. We then condition *ShEPhERD* on these profiles to sample 1000 structures ($n_1 \in [50, 89]$) from $P(\boldsymbol{x}_1|\boldsymbol{x}_3, \boldsymbol{x}_4)$ via inpainting. Fig. 4 shows samples with SA $\leq 4.0$ that score in the top-10 by combined ESP and pharmacophore similarity. Visually, *ShEPhERD* generates structures that align well to the fragments and preserve many of their binding interactions, even though $n_1$ and $n_4$ are significantly out-of-distribution from *ShEPhERD*-MOSES-aq (App. A.9).

## 5    CONCLUSION

We introduced *ShEPhERD*, a new 3D molecular generative model that facilitates interaction-aware chemical design by learning the joint distribution over 3D molecular structures and their shapes, electrostatics, and pharmacophores. Empirically, *ShEPhERD* can sample chemically diverse molecules with highly enriched interaction-similarity to target structures, as assessed via custom 3D similarity scoring functions. In bioisosteric drug design, *ShEPhERD* can design small-molecule mimics of complex natural products, diversify bioactive hits while enriching docking scores despite having no knowledge of the protein, and merge fragments from experimental fragment screens into bioisosteric ligands. We anticipate that future work will creatively extend *ShEPhERD* to other areas of interaction-aware chemical design such as structure-based drug design and organocatalyst design.

REPRODUCIBILITY STATEMENT

Our main text and appendices provide all critical details necessary to understand and reproduce our work, including our training and sampling protocols. To ensure reproducibility, we make our datasets and all training, inference, and evaluation code available on Github at `https://github.com/coleygroup/shepherd` and `https://github.com/coleygroup/shepherd-score`.

ACKNOWLEDGMENTS

The authors would like to thank Wenhao Gao for support with SynFormer. This research was supported by the Office of Naval Research under grant number ONR N00014-21-1-2195. This material is based upon work supported by the National Science Foundation Graduate Research Fellowship under Grant No. 2141064. The authors acknowledge the MIT SuperCloud and Lincoln Laboratory Supercomputing Center for providing HPC resources that have contributed to the research results reported within this paper.

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

# A  APPENDIX

## CONTENTS

## A.1  COMPARING *ShEPhERD* TO REINVENT AND VIRTUAL SCREENING FOR NATURAL PRODUCT LIGAND HOPPING

REINVENT, a state-of-the-art baseline for generative molecular design and optimization, applies a reinforcement learning policy to iteratively update a SMILES recursive neural network (RNN) with a provided reward function (Blaschke et al., 2020). We applied REINVENT to the three natural product ligand hopping tasks defined in section 4, using a combination of our ESP and pharmacophore 3D similarity scoring functions (section 3.2) as REINVENT's reward function. We followed the REINVENT implementation in the Practical Molecular Optimization (PMO) benchmark (Gao et al., 2022). REINVENT was pretrained on the ZINC database (Sterling & Irwin, 2015) and was deployed with a batch size of 64, $\sigma = 500$, an experience replay of 24, and an oracle budget of $10,000$. Training was performed using an Adam optimizer with a learning rate of $5 \times 10^{-4}$. Any SMILES which failed during pharmacophore scoring were assigned a score of 0.

Since REINVENT generates molecules in a 1D SMILES representation, we generate up to 5 conformers for each SMILES in order to apply our 3D similarity scoring functions as the reward function. The procedure to compute the reward for a single generated SMILES is as follows: 1) embed 5 conformers with RDKit's `EmbedMultipleConfs` function, which uses ETKDG; 2) optimize each conformer with MMFF94 for a maximum of 200 steps; 3) cluster the conformers with Butina clustering using an RMSD threshold of 0.1 Å; 4) relax each remaining conformer with xTB in implicit water; 5) extract the ESP surface $x_3$ with $n_3 = 400$ and pharmacophore profile $x_4$ of each relaxed conformer; 6) align each conformer to the target natural product by optimizing our 3D ESP scoring function; 7) calculate the ESP and pharmacophore similarity scores of the ESP-aligned conformers; 8) add the ESP and pharmacophore similarity scores to obtain one combined score per conformer; and 9) take the maximum score across the conformers as the reward.

Fig. 5 compares the distributions of ESP and pharmacophore similarity for samples obtained by (1) using *ShEPhERD* (trained on *ShEPhERD*-MOSES-aq) to sample 2500 molecules from $P(x_1|x_3, x_4)$ via inpainting; (2) virtually screening (VS) 2500 random 3D molecules from *ShEPhERD*-MOSES-aq; and (3) optimizing REINVENT with an oracle/sampling budget of 10,000. Note that REINVENT was pretrained on ZINC, and *ShEPhERD*'s training set (MOSES-aq) is a small subset of ZINC. Each 3D molecule generated by *ShEPhERD* was relaxed with xTB prior to realigning the relaxed structure to the natural product (via maximizing ESP similarity) and scoring the ESP and pharamcophore similarity of the aligned pose. 3D molecules sampled from *ShEPhERD*-MOSES-aq (which are already xTB-relaxed) were directly aligned to the natural product in the same manner. Samples from REINVENT were scored using the procedure outlined in the preceding paragraph. For both *ShEPhERD* and REINVENT, we only compare *valid* samples that have SA scores lower than 4.5. This means that although we initially obtain 2500 and 10000 samples from *ShEPhERD* and REINVENT, respectively, we only compare ∼500 samples from *ShEPhERD* against ∼9000 samples from REINVENT, for each case study. Despite the fewer number of samples, *ShEPhERD* still finds molecules beneath the SA-score threshold that score higher than molecules optimized by REINVENT, for all three natural products. Both *ShEPhERD* and REINVENT find much better molecules than those obtained by randomly sampling from the dataset.

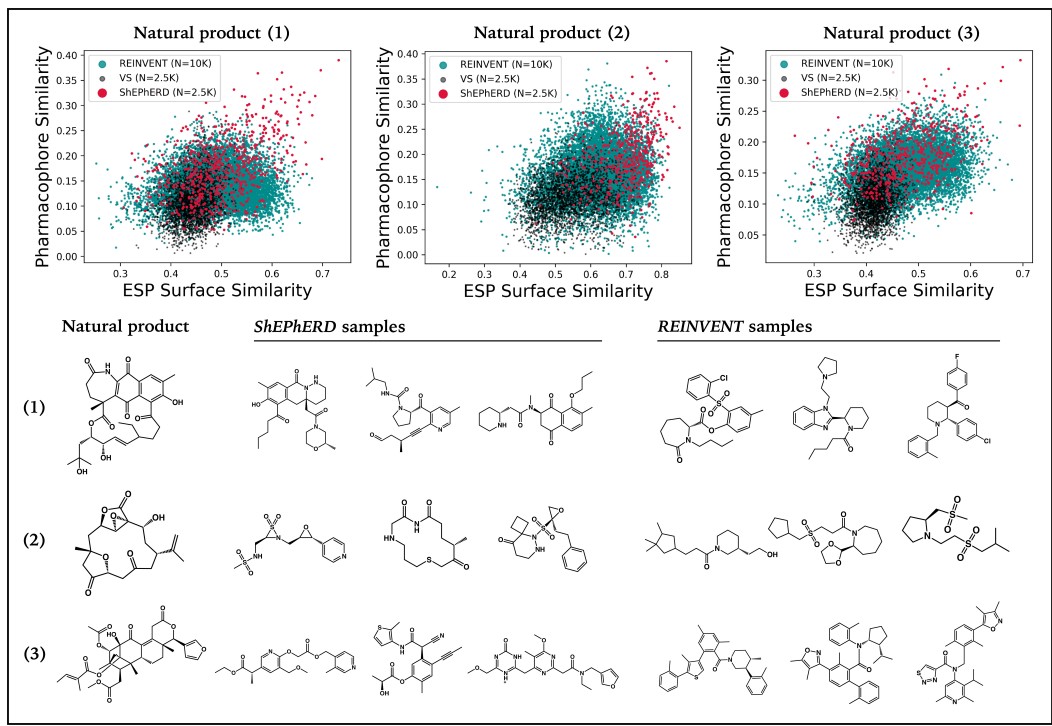

Figure 5: Distributions of 3D ESP and pharmacophore similarity to each of three natural product targets for small-molecules sampled via *ShEPhERD*, REINVENT, or virtual screening (VS). Similarity scores are computed after optimizing the molecular geometry with xTB and aligning the structure to the natural product by maximizing ESP similarity. For *ShEPhERD* and REINVENT, only valid samples with SA score $< 4.5$ are included. Also visualized are examples of top-scoring samples generated by *ShEPhERD* and REINVENT.

We also emphasize that unlike REINVENT, *ShEPhERD* is *not* trained to directly optimize 3D similarity scores to each natural product. Indeed, we use the same *ShEPhERD* model with the same weights for each natural product target. Nor does *ShEPhERD* employ any inference optimization strategies beyond conditional generation via inpainting. Hence, although *ShEPhERD* is already quite capable out-of-the-box, we expect *ShEPhERD* to be able to find even better small-molecule mimics by combining inpainting with other inference-time optimization strategies. For instance, one could use a genetic algorithm which iteratively mutates and evolves the best-scoring samples by partial forward-noising and subsequent denoising.

### A.1.1   REINVENT CONVERGENCE

We plot the average score of each REINVENT batch vs. the iteration number for each natural product target in Fig. 6. For the first natural product, REINVENT appears converged. For the second, REINVENT appears close to converged. For the third natural product, REINVENT could likely find better scoring molecules if trained for longer. We emphasize that we could also run *ShEPhERD* for longer, too. Although REINVENT may not necessarily be converged for each natural product, we follow the PMO benchmark (Gao et al., 2022) and limit the number of oracle calls to 10,000. In practice, even allowing for just 10,000 REINVENT samples required multiple days of computation per target because of the expensive conformer sampling, xTB optimization, and alignment with our scoring functions. We also note that in contrast to REINVENT, *ShEPhERD* is not specifically trained to optimize similarity for any natural product (or for any particular molecule). Fig. 6 also plots SA score distributions for both REINVENT- and *ShEPhERD*-generated molecules.

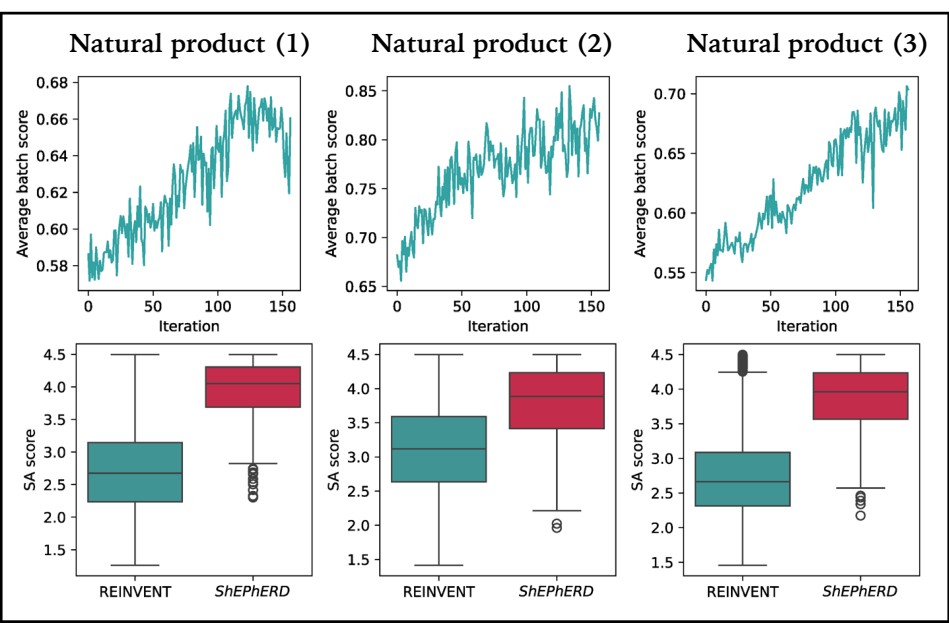

Figure 6: (**Top**) The average score (ESP+pharmacophore 3D similarity) across a batch for each iteration of the optimization for REINVENT. Only natural product (1) seems to have fully converged. All optimizations were concluded at 10,000 oracle calls. (**Bottom**) The distributions of SA score for REINVENT and *ShEPhERD* for molecules with SA score $< 4.5$.

## A.2 Additional details on experiments

### A.2.1 Unconditional joint generation on *ShEPhERD*-GDB17

We define self-consistency as the 3D similarity between the *ShEPhERD*'s generated interaction profile (i.e., $x_2$, $x_3$, or $x_4$) and the true interaction extracted from the generated molecule $x_1$ after xTB relaxation and alignment to the unrelaxed pose via minimizing heavy atom RMSD. We align via heavy-atom RMSD (rather than aligning with the scoring functions) for our self-consistency evaluations in order to preserve the natively generated pose of $x_1$ as much as possible. Since the $P(x_1, x_2)$ and $P(x_1, x_3)$ models employ $n_2, n_3 = 75$, all self-consistency similarity evaluations also use 75 surface points when extracting surface and ESP interaction profiles from the (xTB-relaxed and RMSD-aligned) generated structures. Note that this differs from our other experiments involving conditional generation, which employ $n_2, n_3 = 400$ for similarity scoring.

For each valid *ShEPhERD*-generated molecule, the lower bound of self-consistency is calculated by (globally) aligning a sampled molecule from *ShEPhERD*-GDB17 to the true interaction profile. Since surface sampling is stochastic (App. A.5), the self-similarity between two surfaces extracted from the same $x_1$ may not equal 1.0. Hence, we compare the self-consistency of *ShEPhERD*'s jointly generated samples to upper bounds for the surface and ESP models, calculated as the expectation of the self-similarity between repeated extractions of the same profile from the same $x_1$. We estimate these upper bounds by resampling the true profiles for each of the xTB-relaxed generated molecules 5 times and computing the average similarity between the resampled profiles. The upper bound for pharmacophore self-consistency is 1.0 since pharmacophore extraction is deterministic.

### A.2.2 Conditional generation on *ShEPhERD*-GDB17

We use our 3D similarity scoring functions to evaluate the conditional *ShEPhERD* models. For each *ShEPhERD*-generated sample that is valid after xTB relaxation, we score the similarity of the relaxed molecule to the target interaction profile after optimal alignment using the relevant scoring function. For surface and ESP similarity, we use $n_2, n_3 = 400$ to lessen the effect of stochastic surface sampling on the scores – we only notice small deviations in scores (generally $<0.03$) due to stochasticity (Fig. 19), which should be considered when comparing any two similarity scores. As a baseline, we also sample 20 molecules from the *ShEPhERD*-GDB17 dataset for each target and score their similarity after optimally aligning the sampled molecules to the target interaction profile.

### A.2.3 Natural product ligand hopping

We selected the three natural products in Fig. 4 by manually searching the COCONUT database (`https://coconut.naturalproducts.net/`) for natural product with complex structures that are meaningfully out of distribution compared to *ShEPhERD*-MOSES-aq in terms of their molecular size, 3D conformation, number of pharmacophores, and/or stereochemistry. Notably, all selected natural products have SA scores exceeding 6.0 and have intricate 3D shapes.

Each natural product underwent an extensive conformer search and optimization to identify the lowest-energy conformer as evaluated with xTB in implicit water. Up to 1000 conformers of each natural product were initially enumerated with RDKit before undergoing geometry relaxation with xTB. We extracted the interaction profiles of the lowest-energy conformer following App. A.5.

For each natural product, we sampled 2500 structures from $P(x_1 | x_3^*, x_4^*)$ by inpainting, where $(x_3^*, x_4^*)$ are the ESP and pharmacophore interaction profiles of the natural product. We use the version of *ShEPhERD* trained to learn $P(x_1, x_3, x_4)$ on *ShEPhERD*-MOSES-aq. We used $n_3 = 75$, specified $n_4$ based on the target $x_4^*$, and swept over 25 values of $n_1$ in the range $[36, 80]$ to force *ShEPhERD* to generate both small and larger molecules, even though smaller molecules are less likely to occupy the entire volume of the target natural product (and hence less likely to score well). We sampled 100 structures for each choice of $n_1$. For the valid samples, we relaxed the generated 3D molecular conformation with xTB, and realigned the relaxed structure to the natural product by optimizing our ESP similarity scoring function after resampling $x_3$ for both structures at $n_3 = 400$. We used ESP-based alignment instead of pharmacophore-based alignment as ESP surfaces better capture the 3D shapes of the molecules, and because ESP-based alignments are more robust. Given the ESP-aligned relaxed structure, we scored its ESP and pharmacophore similarity to the natural product. We consider both scores in our evaluations and comparisons to REINVENT (App. A.1).

### A.2.4 BIOACTIVE HIT DIVERSIFICATION

We used the same PDB co-crystal structures as used in (Zheng et al., 2024) and use the prepared receptor files `.pdbqt` AutoDock Vina files provided by Therapeutic Data Commons (TDC) (Huang et al., 2021). These targets are **1iep** (Nagar et al., 2001; 2002), **3eml** (Jaakola et al., 2008a;b), **3ny8** (Wacker et al., 2010a;b), **4rlu** (Li et al., 2014; Dong et al., 2015), **4unn** (Read et al., 2014; Naik et al., 2015), **5mo4** (Cowan-Jacob, 2016; Wylie et al., 2017), and **7l11** (Deshmukh et al., 2020; Zhang et al., 2021). We use the corresponding co-crystal ligands: STI, ZMA, JRZ, HCC, QZZ, Nilotinib (NIL), and XF1.

For each experimental ligand, the lowest-energy conformer was identified by initially embedding 1000 conformers with RDKit and relaxing each with xTB in implicit water. Then we extract the interaction profiles of the lowest energy conformer following App. A.5. We use the crystal structure pose from the PDB (Berman et al., 2000) to also extract the interaction profiles and obtain the "PDB pose" conformation. We added hydrogens to these PDB poses using RDKit's `AddHs` function with `addCoords=True`, but did *not* relax the structures with xTB. The ESP surfaces were computed from the partial charges of the native conformation (with added hydrogens) using a single point xTB calculation in implicit water.

For the two poses of each experimental ligand, we generate 500 samples from $P(x_1|x_3^*, x_4^*)$ by inpainting, where $(x_3^*, x_4^*)$ are the ESP and pharmacophore interaction profiles of the ligand. We use the *ShEPhERD* model that was trained to learn $P(x_1, x_3, x_4)$ on *ShEPhERD*-MOSES-aq. In a similar manner to the natural products, we used $n_3 = 75$ and specified $n_4$ based on the target $x_4^*$. We swept over 25 values of $n_1$ in the range $[n_1^* - 12, n_1^* + 12]$ where $n_1^*$ is the total number of atoms in the PDB ligand. We generate 20 structures for each choice of $n_1$. For valid samples, we extract SMILES strings. Different from the other experiments, we do not use xTB for relaxation as we simply dock the valid molecules from a SMILES representation.

To evaluate valid samples, we implement an adapted version of the `Vina_smiles` class from the TDC oracles. For each SMILES, we 1) embed a conformer with RDKit ETKDG with a random seed of 123456789; 2) optimize with MMFF94 for 200 steps; 3) prepare the conformer for docking with Meeko (forlilab, n.d.); 4) dock the molecule with Autodock Vina v1.2.5 (Trott & Olson, 2010; Eberhardt et al., 2021) using a random seed of 987654321 and exhaustiveness of 32; and 5) obtain a Vina score in kcal/mol, which we use as a (poor) proxy for binding affinity.

Fig. 7 showcases examples for the 5mo4 and 7l11 docking targets. For each target, we show the PDB ligand in its crystal structure pose, the top-scoring redocked pose, and its ensemble of top-10 scoring poses. We then visualize the top scoring pose of three top-scoring *ShEPhERD*-generated ligands, overlaid on the best docking pose of the PDB ligand. Although the top-scoring docked pose of the *ShEPhERD*-generated ligands may not align perfectly with the top-scoring pose of the PDB ligand, it is evident that the generated ligands explore poses that engage in the same binding interactions as PDB ligands when we compare their ensembles of docked poses. Moreover, as Fig. 4 illustrates, in each case we can identify a docked pose for each of the *ShEPhERD*-generated ligands that closely aligns with the top pose of the PDB ligand.

Fig. 8 shows the distributions of the top-10 Vina docking scores for *ShEPhERD*-generated ligands compared to the top-10 docking scores obtained by virtually screening 10K molecules from *ShEPhERD*-MOSES-aq. Despite having no knowledge of the protein pocket and only generating <500 (valid) ligands per target, *ShEPhERD* enriches the top-1 Vina score relative to the redocked PDB ligand for all 7 targets, 3/7 targets relative to the docking screen when conditioning on the lowest energy conformer, and 5/7 targets relative to the docking screen when conditioning on the PDB ligand's bound pose. Note that conditioning on the PDB ligand's bound pose indirectly leaks information about the protein pocket. We emphasize that for the 2/7 cases (3ny8 and 4rlu) where the docking screen outperforms (in terms of top-1 Vina score) the *ShEPhERD*-generated ligands when conditioning on the PDB ligand's bound pose, the Vina docking score of the PDB ligand is relatively low (<9 kcal/mol). Since we use *ShEPhERD* to diversify the PDB ligand while preserving its 3D interactions – *not* optimize docking score – we do not necessarily expect *ShEPhERD* to find higher-scoring ligands compared to an unbiased docking screen which can explore larger regions of chemical space.

To investigate the potential of using *ShEPhERD* to optimize docking score, we also condition *ShEPhERD* on the lowest energy conformer and docked pose of the best scoring compound from the 10K

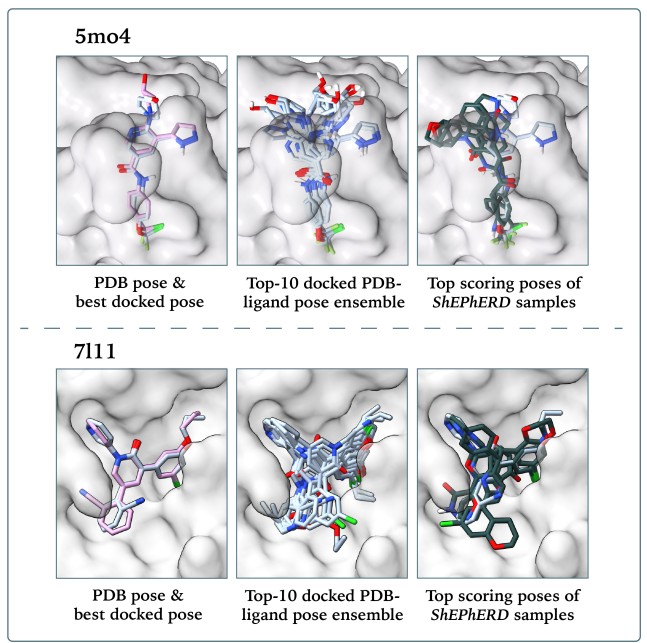

Figure 7: (**Left**) The true binding pose of the experimental ligand extracted from the PDB (pink) overlaid on the top-scoring re-docked pose (light blue). (**Middle**) Ensemble of the top-10 docked poses for the experimental ligand, ranked by Vina docking score. (**Right**) Top-1 Vina-scoring pose for each of the three select *ShEPhERD* samples highlighted in Fig. 4 (green-grey), overlaid on the top-scoring re-docked pose for the experimental ligand (light blue). The top and bottom rows show results for 5mo4's and 7l11's experimental co-crystal ligands, respectively. Although the top-1 docked pose for the *ShEPhERD*-generated ligands may not perfectly align with the top scoring pose of the experimental ligand, the ensembles of their docked poses do closely match, indicating that the experimental ligands and the *ShEPhERD*-diversified analogues explore common binding modes. We note that true experimental binding poses often do not match the top-1 docked poses simulated from docking programs like AutoDock Vina, and hence we show the ensembles of docked poses to account for this uncertainty.

docking screen. Results are shown in Fig. 9. With this strategy, *ShEPhERD* generates new ligands with higher top-1 Vina scores for 5/7 targets relative to the best scoring compounds from the 10K docking screen (which served as the conditioning information to *ShEPhERD*). Notably, *ShEPhERD* drastically improves Vina scores on the 3ny8 target compared to when conditioning on the experimental PDB ligand, shifting the mean of the top-10 scores from $-9.96$ to $-11.70$ kcal/mol (Fig. 8). This PDB ligand scores poorly (the worst among all PDB ligands). Another interesting phenomenon is that neither the 10K screened molecules nor the *ShEPhERD*-generated molecules (conditioned on the best from the screen) manage to surpass the docking score of 7l11's experimental PDB ligand. However, *ShEPhERD does* improve this docking score when directly conditioning on the 7l11 PDB ligand (Fig. 8), highlighting *ShEPhERD*'s flexibility in structure-based drug design.

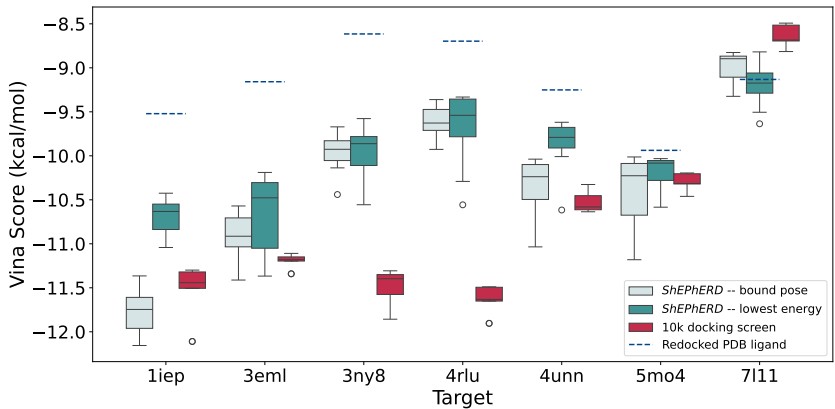

Figure 8: The distributions of the top-10 scoring *ShEPhERD*-generated samples when conditioning on the PDB ligand's lowest energy conformer (teal) or bound pose (light blue), for all protein targets. We compare to the distributions for the top-10 scoring compounds from the 10K docking screen (red). The Vina score of the redocked PDB ligand is denoted by the dotted line. Note that in each case, only 500 molecules were generated by *ShEPhERD* – in contrast to the 10K sampled in the virtual screen. Furthermore, we only docked the <500 samples for which we could extract a valid SMILES string from the generated 3D conformation.

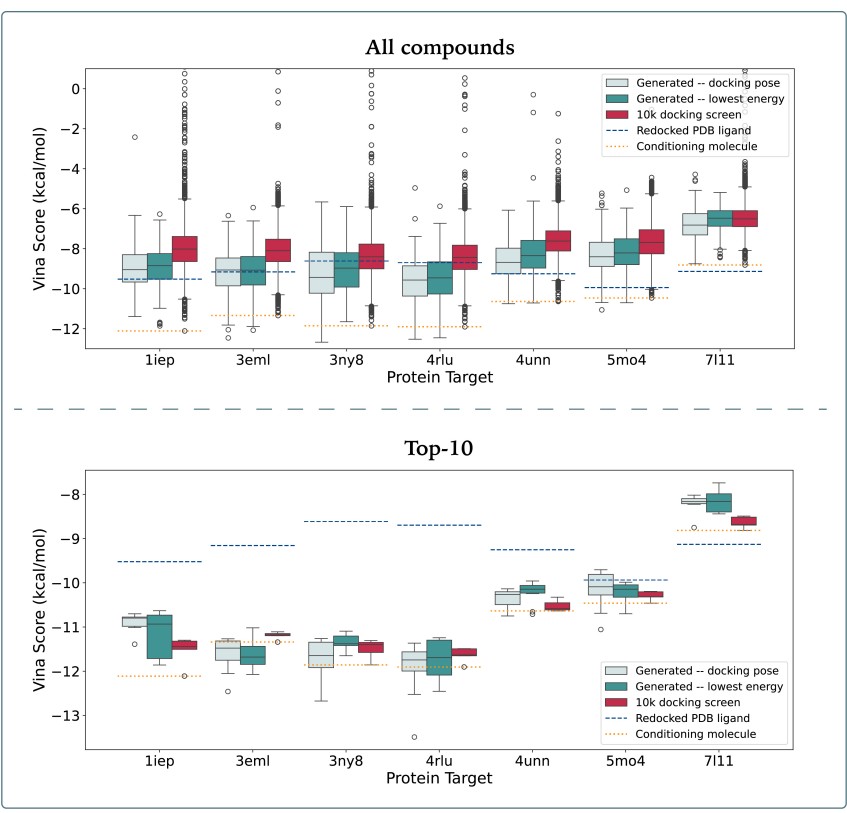

Figure 9: Docking results for *ShEPhERD*-generated molecules when conditioning on the best scoring ligand from the 10K docking screen. We show results when conditioning on the lowest energy conformer (teal) or top docking pose (light blue) of the top-scoring screened ligand. We also show the original 10K docking screen (red), the scores of the redocked PDB ligands (blue dotted line), and the scores of the conditioning molecule (orange dotted line). The top plot shows the full Vina score distributions, while the bottom plot only shows the distributions for the top-10 scoring compounds.

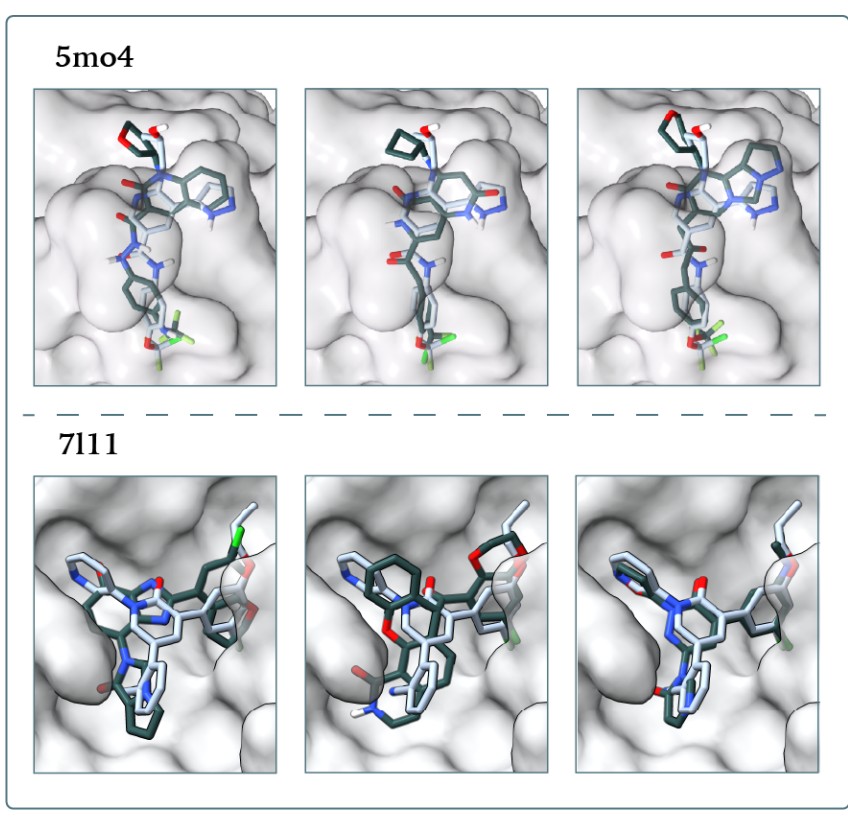

Figure 10: The individually visualized docking poses for each sampled molecule for PDB targets 5mo4 and 7l11 from Fig. 4.

A.2.5 BIOISOSTERIC FRAGMENT MERGING

We demonstrate *ShEPhERD*'s native capacity for bioisosteric fragment merging on the antiviral target EV-D68 3C protease. Starting from the 25 fragments hits in the catalytic pocket identified by Wills et al. (2024), we downselected 13 fragments that still span the P1, P2, and P3 regions of the active site through visual inspection using Fragalysis (Diamond Light Source, 2024) and ChimeraX (Meng et al., 2023). Specifically, we removed fragments that did not have common interactions, that significantly deviated from the fragment ensemble's aggregate shape, or added no extra information. The final list of fragments were: x0147_0A, x1071_0A, x1083_0A, x1084_0A, x1140_0A, x1163_0A, x1498_0A, x1537_0A, x1594_0A, x1919_0A, x2021_0A, x2099_0A, and x2135_0A.

For each fragment, we added hydrogens using RDKit's `AddHs` function with `addCoords=True`. We then generated an aggregate shape representation by sampling from a surface with a probe radius of 0.6. We compute partial charges with an xTB single point calculation in implicit water for each fragment separately. Then, we compute the ESP at each of the surface points for each fragment, and average the ESP contributions at each point. See App. A.5 for more details.

Extraction of the pharmacophores was a manual task based on selecting those that interacted with the pocket (as detected by Fragalysis and ChimeraX). After manual downselection, pharmacophores of the same type underwent clustering if there were at least two within a threshold distance. These threshold distances were 1.0 Å for HBAs, 2.0 Å for HBDs, 1.5 Å for aromatic groups, and 2.0 Å for hydrophobes. These were somewhat arbitrary cutoffs that yielded reasonable pharmacophore hypotheses upon visual inspection. Table 1 lists the final 27 selected pharmacophores.

Table 1: The 27 pharmacophores used to condition *ShEPhERD* for the bioisosteric fragment merging demonstration. Only hydrogen bond acceptors (HBA), hydrogen bond donors (HBD), aromatic groups, and hydrophobes are contained in these 27 pharmacophores.

| Type | Coordinates (P) | Vector (V) |
|---|---|---|
| HBA | (-7.5155, -3.7035, -9.0635) | (0.1900, 0.7041, -0.6842) |
| | (-6.7175, -4.8185, -8.8258) | (0.8205, -0.2287, -0.5240) |
| | (-2.3550, -7.3310, -0.0453) | (0.7719, 0.4600, -0.4388) |
| | (-6.5197, -4.3363, -4.1827) | (0.5106, 0.7063, -0.4905) |
| | (-6.9450, -6.9160, -5.0510) | (-0.1783, -0.4378, -0.8812) |
| | (-3.2780, -7.2170, 2.2860) | (-0.7287, 0.6270, -0.2755) |
| | (-5.7270, -2.3950, -0.6220) | (0.4491, -0.6321, 0.6316) |
| | (-1.9580, -8.8820, 3.3910) | (0.7956, -0.5564, -0.2396) |
| | (-5.0980, -6.4300, -1.5740) | (0.4851, 0.6853, -0.5432) |
| | (-5.8830, -1.1980, 0.0490) | (0.4090, -0.3130, 0.8572) |
| | (-5.6690, 0.0200, -2.0780) | (0.5578, 0.5389, -0.6312) |
| HBD | (-9.4350, -2.6960, -8.2680) | (0.1103, 0.6563, -0.7464) |
| | (-3.2780, -7.2170, 2.2860) | (0.7506, 0.2245, 0.6214) |
| | (-1.9580, -8.8820, 3.3910) | (-0.2298, 0.1607, 0.9599) |
| | (-6.9480, -2.0740, -1.9750) | (-0.3293, 0.3106, -0.8917) |
| Aromatic | (-7.6710, -4.5475, -8.2231) | (-0.5639, -0.4342, -0.7025) |
| | (-11.6032, -2.0268, -6.4003) | (0.3775, 0.9134, -0.1523) |
| | (-4.6030, -8.0915, -1.0500) | (0.4302, -0.8132, -0.3919) |
| | (-9.5312, -4.0857, -7.3044) | (-0.5073, -0.3479, -0.7884) |
| Hydrophobe | (-7.6710, -4.5475, -8.2231) | (0.0, 0.0, 0.0) |
| | (-11.6032, -2.0268, -6.4003) | (0.0, 0.0, 0.0) |
| | (-4.6030, -8.0915, -1.0500) | (0.0, 0.0, 0.0) |
| | (-9.5312, -4.0857, -7.3044) | (0.0, 0.0, 0.0) |
| | (-5.6776, -0.4634, -1.0174) | (0.0, 0.0, 0.0) |
| | (-6.9299, -5.4087, -1.1730) | (0.0, 0.0, 0.0) |
| | (-6.7990, -4.5530, -3.9980) | (0.0, 0.0, 0.0) |
| | (-7.9010, 0.1820, -0.7890) | (0.0, 0.0, 0.0) |

Using the extracted ESP ($x_3$) and pharmacophore ($x_4$) interaction profiles, we used the version of *ShEPhERD* trained to learn $P(x_1, x_3, x_4)$ on *ShEPhERD*-MOSES-aq to generate 1000 structures from $P(x_1|x_3, x_4)$ via inpainting. We used $n_3 = 75$, specified $n_4 = 27$, and swept over 40 values of $n_1$ in the range $[50, 89]$ at 25 samples each. For the valid samples, we relaxed the generated 3D molecular conformation with xTB, and realigned the relaxed structure to the natural product by optimizing our ESP similarity scoring function after resampling $x_3$ for both structures at $n_3 = 400$. We used ESP-based alignment instead of pharmacophore-based alignment as ESP surfaces better capture 3D shapes, and because ESP-based alignments are more robust. Given the ESP-aligned relaxed structure, we compute the ESP and pharmacophore similarity scores. Fig. 11 visualizes four *ShEPhERD*-generated samples with neutral charge and SA$\leq$ 4.0 that were in the top-10 by combined ESP and pharmacophore similarity scores.

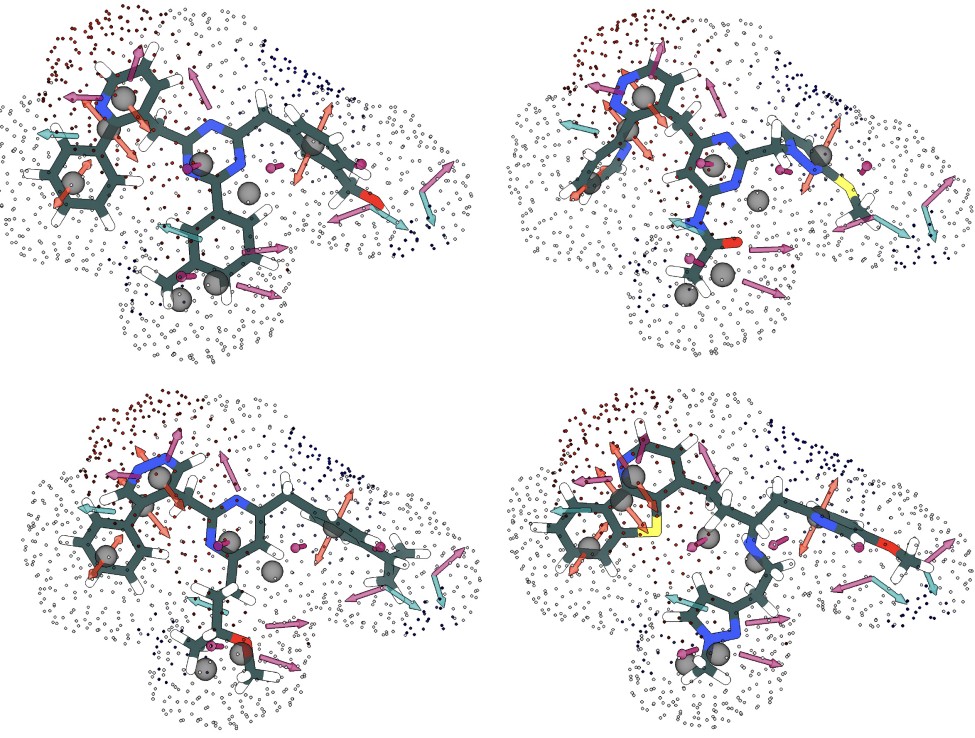

Figure 11: Examples of four top-scoring *ShEPhERD*-generated ligands obtained via bioisosteric fragment merging. Shown are the xTB-relaxed structures of the generated 3D molecules after being realigned to the fragments' ESP and pharmacophore interaction profiles. Alignment is performed by optimizing ESP similarity. The ligands are overlaid on the target interaction profiles. The target ESP surface has been upsampled for visualization.

## A.3 Defining and analyzing validity of generated 3D molecules

Table 2: Global validity reported as percentages for 1000 unconditional and 2000 conditional samples from various *ShEPhERD* models trained on *ShEPhERD*-GDB17. Here, "Validity" is the percentage of valid molecules (see definition below); "Validity xTB" is the percentage of valid molecules after xTB optimization; "Graph consis." is the percentage of samples where the atom connectivity remains the same *given* that the molecules are valid both pre- and post- xTB-optimization; and "Valid neutral" is the percentage of valid samples that have molecular charge $= 0$.

**Unconditional Generation**

| Samples | $P(\boldsymbol{x}_1, \boldsymbol{x}_2)$ | $P(\boldsymbol{x}_1, \boldsymbol{x}_3)$ | $P(\boldsymbol{x}_1, \boldsymbol{x}_4)$ |
|---|---|---|---|
| Validity (%) | 96.2 | 92.7 | 73.7 |
| Validity xTB (%) | 96.3 | 93.0 | 73.7 |
| Graph consis. (%) | 99.1 | 99.5 | 98.4 |
| Valid neutral (%) | 84.1 | 93.2 | 89.7 |

**Conditional Generation**

| Samples | $P(\boldsymbol{x}_1|\boldsymbol{x}_2)$ | $P(\boldsymbol{x}_1|\boldsymbol{x}_3)$ | $P(\boldsymbol{x}_1|\boldsymbol{x}_4)$ |
|---|---|---|---|
| Validity (%) | 96.0 | 91.9 | 80.7 |
| Validity xTB (%) | 96.2 | 92.9 | 79.9 |
| Graph consis. (%) | 97.4 | 92.8 | 93.8 |
| Valid neutral (%) | 84.2 | 82.5 | 72.6 |

**Defining validity.** We define validity as the fraction of molecules that can converted from the atomic point cloud $(\boldsymbol{a}, \boldsymbol{C})$ (i.e., element types and coordinates)[3] to an RDKit `Mol` object via RDKit's internal xyz2mol functionality that satisfy valencies as well as other constraints enumerated below. Empirically, we find that this method of extracting molecules from xyz-formatted structures is simple, convenient, and quite reliable when (1) explicit hydrogens are included in the 3D structure, (2) the 3D molecular geometry is of high quality, and (3) the overall charge of the molecule is known.

In particular, RDKit is used to (1) generate a `Mol` object from $(\boldsymbol{a}, \boldsymbol{C})$ formatted as an `.xyz` file; (2) determine bonds via a rule-based algorithm based on interatomic distances; (3) sanitize the molecule; (4) ensure that there are no radical electrons (which aren't included in our datasets); (5) check that structure is not fragmented; and (6) check that there are no more than six atomic formal charges that are non-zero (which usually indicates that a valid resonance structure of an aromatic ring could not be determined). In an effort to handle charged molecules, we run this procedure for molecular charges in this order: [0, +1, -1, +2, -2] and break the loop if extraction of a molecule is successful. Finally, an xTB single point calculation is run to obtain partial charges. The molecular charge is specifically used to determine the bonds, to sanitize the molecule, and to run the single point calculation. The sampled $\boldsymbol{x}_1$ is determined invalid if any of the aforementioned steps fails.

We point out that in some cases, generated molecules may fail our stringent validity checks due to having imperfect molecular geometries (e.g., a bond length is slightly out of distribution). To consider these cases and to measure a notion of distance from the training distribution, we perform an xTB local geometry optimization with the generated $(\boldsymbol{a}, \boldsymbol{C})$ (formatted as an `.xyz` file) and repeat the validity checks. If the natively generated structure was determined invalid pre-optimization, the xTB optimization is performed with a neutral overall charge; otherwise the same charge was used as was determined upon initial molecule extraction with RDKit.

If both the pre- and post-optimized samples are valid, we check if the graphs are consistent by matching their SMILES strings. If consistent, we also compute the strain energy and heavy-atom RMSD between the relaxed and unrelaxed conformations.

---

[3]Although *ShEPhERD generates* formal charges $\boldsymbol{f}$ and the bonds $\boldsymbol{B}$, we ignore these outputs when extracting a molecule from $\boldsymbol{x}_1$. We argue that the validity of a generated 3D molecular structure should depend only on the generated 3D structure, *not* on a jointly generated 2D graph. Hence, we use the diffusion processes over $\boldsymbol{f}$ and $\boldsymbol{B}$ primarily as a way to help regularize the diffusion processes over $\boldsymbol{a}$ and $\boldsymbol{C}$. Moreover, by defining validity only with respect to the element types and coordinates, our validity checks may be applied to other 3D molecular generative models which do not generate the 2D molecular graph.

**Analyzing validity.** Metrics of validity for unconditional and conditional across all samples from *ShEPhERD-GDB17* are presented in Table 2. There is a trend of relatively high validity ($\geq 96\%$) for the joint generation of $x_1$ and $x_2$, slightly lower validity (91-93%) for the joint generation of $x_1$ and $x_3$, and a significant decrease for the joint generation of $x_1$ and $x_4$. We hypothesize that the lower validity of the pharmacophore-based models is because of the discrete nature of the pharmacophore interaction profile relative to the surface or ESP-attributed surface. Notably, validity improves in the pharmacophore-conditioned sampling scheme compared to the unconditional setting. Constraining the sampling space by specifying the pharmacophores during inpainting may be an easier task as it forces the model to form certain groups/substructures. Specifying the pharmacophores based on a target profile also avoids relying on *ShEPhERD*'s jointly-generated $x_4$ (which may have errors) to guide the denoising of $x_1$. One way to improve joint pharmacophore generation may be to introduce a noise schedule that denoises the pharmacophores earlier than $(a, C)$, as a means to implicitly condition the denoising of the 3D molecules on less noisy pharmacophore representations.

Graph consistency is high for most models which implies that (1) the inter-atomic geometry of valid structures are within the bounds for the the bond-determining rules, and (2) the generated conformations are precise enough that relaxation with xTB is stable. The fraction of valid molecules with neutral charge ranges from $84.1\% - 93.2\%$ for unconditional samples and $72.6\% - 84.5\%$ for the conditional samples. This is somewhat surprising because the training distribution only contains neutral species. We attribute this behavior to the presence of zwitterions (neutral molecules that contain balanced atoms/groups of opposite charge) in the training sets; the model has learned to generated charged substructures, but hasn't been trained long enough to learn to neutralize the overall molecule every time. Encouragingly, there is a higher rate of generating neutral species for the $P(x_1, x_3)$ and $P(x_1, x_4)$ models relative to the $P(x_1, x_2)$ model, which perhaps implies that using representations that implicitly encode information about the charge improves the model's ability to learn the correct distribution of molecular charge. In contrast, the conditional $P(x_1|x_3)$ and $P(x_1|x_4)$ models perform worse than $P(x_1|x_2)$ model in terms of generating neutral molecules. This may be because the model learns to associate certain ESP or pharmacophoric criteria with the presence of charged groups (e.g., carboxylate anions yield negative ESP surfaces), and hence introduces charged groups more often while failing to neutralize the overall molecule.

Fig. 12 details how validity is impacted by the selection of the number of atoms ($n_1$) for unconditionally generated samples (top row) and conditionally generated samples (bottom row) from models trained on *ShEPhERD*-GDB17. In both unconditional and conditional settings, versions of *ShEPhERD* that jointly model shape or electrostatics attain high validity rates when the number of atoms is in-distribution of the training set (Fig. 23). On the other hand, the pharmacophore models generally generate more invalid structures as the number of atoms increase. We hypothesize that this is related to the difficulty of modeling discrete pharmacophores and satisfying all (jointly-generated or conditional) pharmacophore criteria as the generated molecules become larger and hence more complex. We emphasize that in all cases, *ShEPhERD* shows the capacity to extrapolate into unseen chemical space (e.g., $n_1$ exceeding the bounds of the training datasets), albeit at lower validity rates.

Table 3 and Figures 13-14 show the strain energy and heavy-atom RMSD of *ShEPhERD*-generated molecular structures compared to their xTB-relaxed geometries. Note that RMSDs are computed after realigning (by minimizing RMSD) the relaxed geometry to the unrelaxed geometry. Mean strain energies remain relatively low for the model trained on *ShEPhERD*-GDB17 (mean strain energies <10 kcal/mol). Indeed, these strain energies are typically lower than the strain energies of molecules with ***MMFF94**-optimized* geometries that have their heavy-atom RMSD and strain energies measured after relaxing the MMFF94 geometries with xTB. However, the high standard deviations in the strain energy distributions indicate that there are still many generated molecules that are highly strained (up to nearly 1000 kcal/mol). Nevertheless, almost all of the RMSD's are <1Å which implies that the generated molecules are close to the true xTB geometry. Indeed, the unconditionally-generated samples for models trained on *ShPhERD*-GDB17 have average heavy-atom RMSDs below 0.08 Å. We note that slightly distorted atomic positions can vastly increase strain energy, which makes strain energy an unforgiving (but still useful) measure of absolute geometric quality. We notice a similar trend to validity where samples from $P(x_1, x_2)$ model perform the best in terms of strain energy while samples from $P(x_1, x_4)$ model perform the worst, though the RMSD's are quite low across all the unconditional models.

The models trained on *ShEPhERD*-MOSES-aq perform worse in terms of both the strain energy and RMSD metrics (Table 3), though the trends with respect to $n_1$ are similar (Fig. 14). We note that

the molecules in *ShEPhERD*-MOSES-aq are generally larger than those in *ShEPhERD*-GDB17. For instance, the largest molecules in MOSES have 27 non-hydrogen atoms, whereas the largest molecules in GDB17 have 17 non-hydrogen atoms. *ShEPhERD*-MOSES-aq also contains fewer training examples than *ShEPhERD*-GDB17; we expect the performance of models trained on *ShEPhERD*-MOSES-aq will improve with further training on larger drug-like datasets.

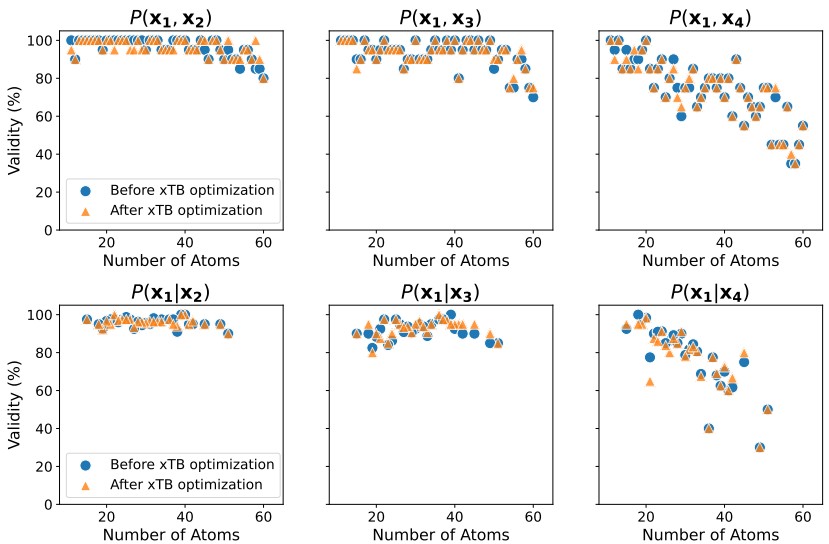

Figure 12: Validity as a function of the number of generated atoms ($n_1$) for 3D molecules $\boldsymbol{x}_1$ generated by various *ShEPhERD* models trained on *ShEPhERD*-GDB17 in (**Top Row**) unconditional or (**Bottom Row**) conditional settings. Blue circles and orange triangles indicate the validity of generated molecular structures before and after xTB optimization, respectively.

Table 3: The overall mean and standard deviation of strain energies and RMSDs for unconditional samples from versions of *ShEPhERD* trained on *ShEPhERD*-GDB17 and *ShEPhERD*-MOSES-aq (Fig. 30). We compute the metrics from the valid samples amongst 1000 total samples from each model. The column "MMFF94" is a baseline that subsamples 1000 molecules from each respective dataset, generates an MMFF94-level conformer for each molecule, and measures the strain energy and heavy-atom RMSD upon xTB relaxation. Molecules are relaxed with xTB in the gas phase for *ShEPhERD*-GDB17, and in implicit water for *ShEPhERD*-MOSES-aq.

### *ShEPhERD*-GDB17

|  | $P(\boldsymbol{x}_1, \boldsymbol{x}_2)$ | $P(\boldsymbol{x}_1, \boldsymbol{x}_3)$ | $P(\boldsymbol{x}_1, \boldsymbol{x}_4)$ | MMFF94 |
|---|---|---|---|---|
| **Strain Energy** | **2.81** | **5.80** | **5.87** | **7.34** |
| (kcal/mol) | $\pm 5.32$ | $\pm 73.17$ | $\pm 47.36$ | $\pm 9.20$ |
| **RMSD** | **0.084** | **0.081** | **0.089** | **0.193** |
| (Å) | $\pm 0.133$ | $\pm 0.144$ | $\pm 0.155$ | $\pm 0.184$ |

### *ShEPhERD*-MOSES-aq

|  | $P(\boldsymbol{x}_1, \boldsymbol{x}_3, \boldsymbol{x}_4)$ | MMFF94 |
|---|---|---|
| **Strain Energy** | **17.43** | **7.99** |
| (kcal/mol) | $\pm 124.02$ | $\pm 3.26$ |
| **RMSD** | **0.273** | **0.337** |
| (Å) | $\pm 0.292$ | $\pm 0.274$ |

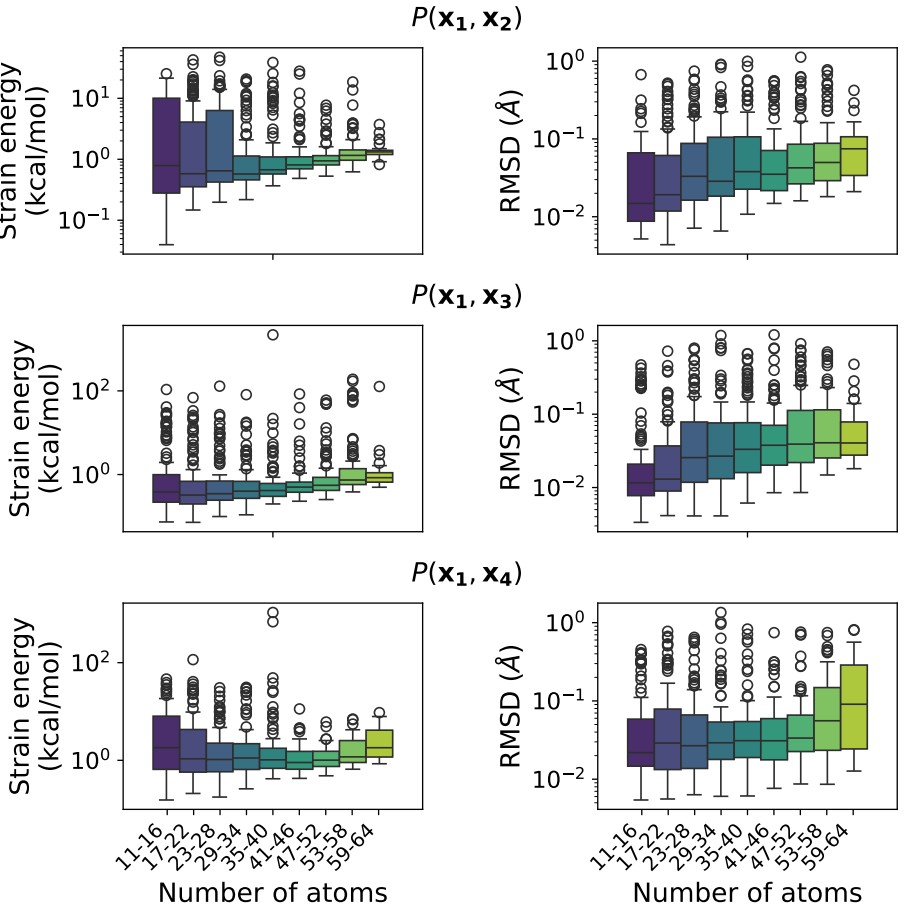

Figure 13: Strain energy and heavy-atom RMSD for samples unconditionally generated by various models trained *ShEPhERD*-GDB17. The box plots are binned by the number of atoms. Though the strain energies can be high, the RMSDs are quite low. Summary statistics are reported in Table 3.

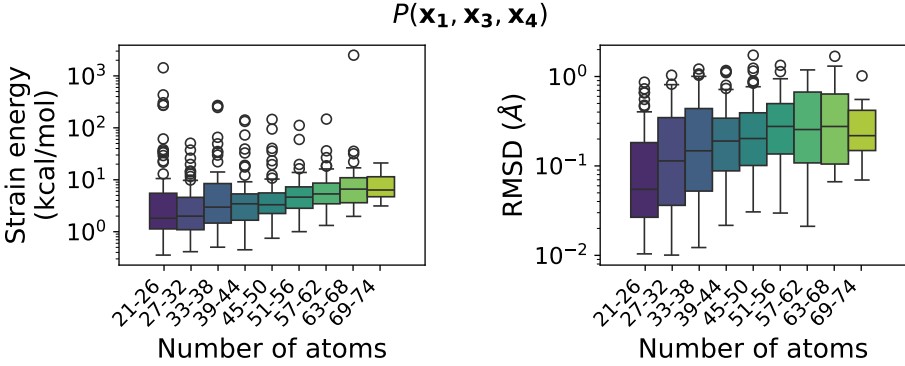

Figure 14: Strain energy and heavy-atom RMSD for valid samples (amongst 1000 total samples) unconditionally generated by *ShEPhERD* trained to learn $P(x_1, x_3, x_4)$ on *ShEPhERD*-MOSES-aq. The box plots are binned by the number of atoms. Though the strain energies can be high, the RMSDs are quite low. Summary statistics are reported in Table 3.

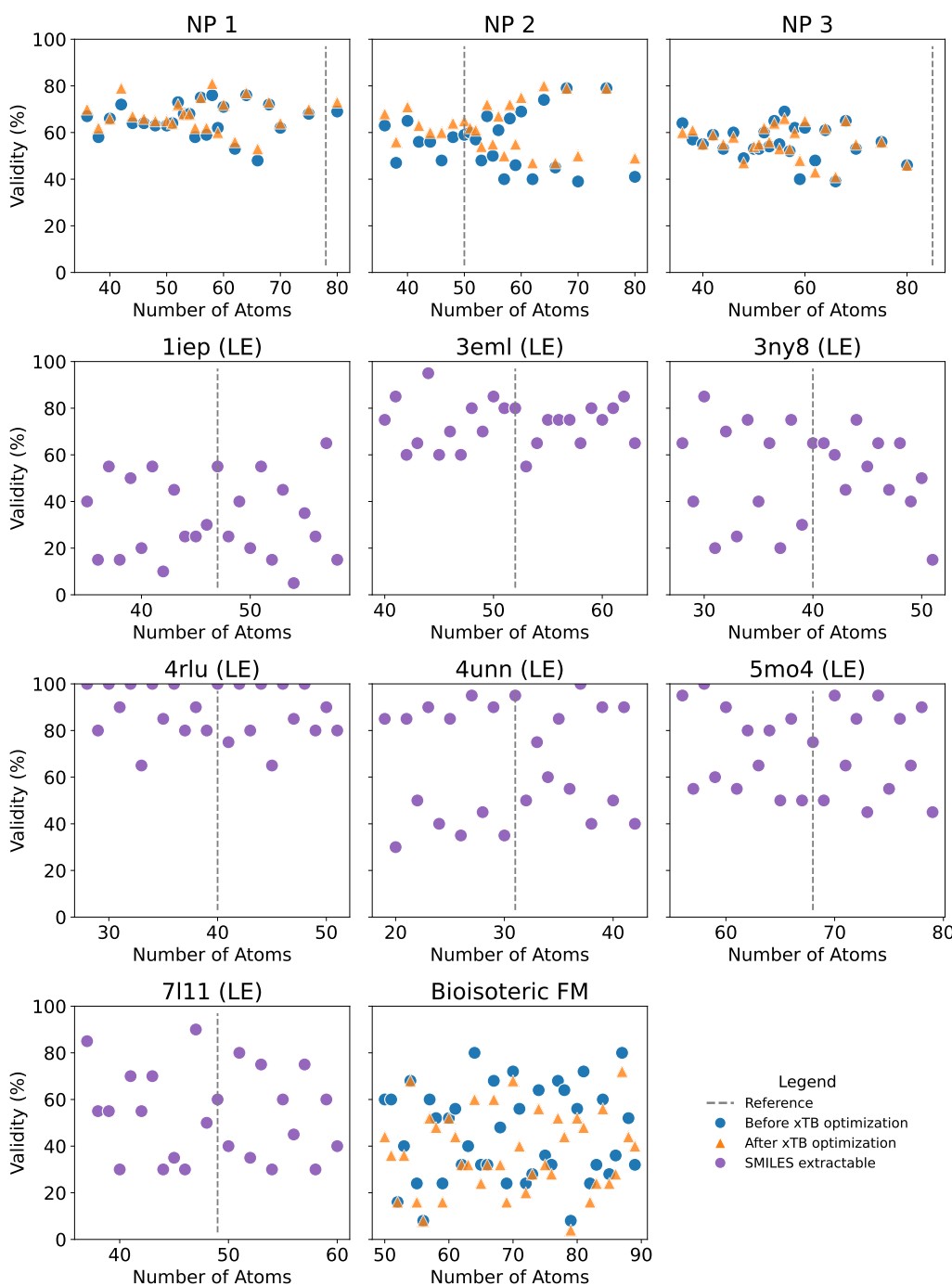

Figure 15: Validity as a function of the number of generated atoms for ligands generated with *ShEP-hERD* for the natural product (NP) ligand hopping, bioactive hit diversification, and bioisosteric fragment merging (FM) experiments. For the NP analog generation and FM experiments, we report validity both before and after relaxation of $x_1$ in implicit water with xTB. For the bioactive hit diversification experiments, we only check the ability to extract SMILES strings from the generated $x_1$ pre-xTB-optimization. There are no clear patterns in the validity of the molecules generated by *ShEPhERD* in these experiments, but we attribute the overall lower validity rates to the difficulties of each task and the need for out-of-distribution sampling (Fig. 22), particularly for the NP and FM experiments. Validity rates for each experiment are summarized in Table 4.

Table 4: Global validity metrics for molecules generated by *ShEPhERD* for the natural product ligand hopping, bioactive hit diversification, and bioisosteric fragment merging experiments.

### Natural product ligand hopping

|                  | NP 1 | NP 2 | NP 3 |
|------------------|------|------|------|
| Validity (%)     | 65.6 | 56.5 | 55.6 |
| Validity xTB (%) | 67.6 | 62.4 | 56.0 |
| Graph consis. (%)| 78.5 | 88.6 | 68.2 |
| Valid neutral (%)| 50.5 | 72.3 | 47.0 |

### Bioactive hit diversification

|                  | 1iep | 3eml | 3ny8 | 4rlu | 4unn | 5mo4 | 7l11 |
|------------------|------|------|------|------|------|------|------|
| Validity (%)     | 32.7 | 73.3 | 52.3 | 88.5 | 66.5 | 71.5 | 53.5 |
| Valid neutral (%)| 47.1 | 60.0 | 75.7 | 78.8 | 55.2 | 71.4 | 62.6 |

### Bioisosteric fragment merging

|                  | Bioisosteric FM |
|------------------|-----------------|
| Validity (%)     | 45.0            |
| Validity xTB (%) | 37.5            |
| Graph consis. (%)| 76.6            |
| Valid neutral (%)| 46.0            |

A.4    Unconditional and conditional generation metrics

**Novelty** is defined as the fraction of valid molecules that are not contained in the training set. **Uniqueness** is defined as the fraction of valid molecules that are only generated once across the samples from a particular model in a given experiment. Novelty and uniqueness metrics for all generated samples (in both unconditional- and conditional-generation experiments) are reported in Table 5. *ShEPhERD* attains nearly 100% novelty and uniqueness for all cases.

Fig. 16 also reports specific property distributions (SA Score, logP, QED, Fsp3) of unconditionally-generated molecules from versions of *ShEPhERD* trained on *ShEPhERD*-GDB17 or *ShEPhERD*-MOSES-aq. Property distributions for generated molecules are compared against the distributions from a random sampling of molecules from the corresponding datasets.

Table 5: Novelty and uniqueness metrics across different versions of *ShEPhERD* trained on either *ShEPhERD*-GDB17 or *ShEPhERD*-MOSES-aq. The first six rows report metrics for *ShEPhERD* models when trained on *ShEPhERD*-GDB17 and applied to either unconditional ($P(\boldsymbol{x}_1, \boldsymbol{x}_i)$) or conditional ($P(\boldsymbol{x}_1 | \boldsymbol{x}_i)$) generation. The remaining rows report metrics for *ShEPhERD* when trained to learn $P(\boldsymbol{x}_1, \boldsymbol{x}_3, \boldsymbol{x}_4)$ on *ShEPhERD*-MOSES-aq. $P(\boldsymbol{x}_1, \boldsymbol{x}_3, \boldsymbol{x}_4)$ indicates this model when applied to unconditionally generate 1000 samples, whereas $P(\boldsymbol{x}_1 | \boldsymbol{x}_3, \boldsymbol{x}_4)$ indicates this model when applied to conditional generation in the natural product (NP), bioactive hit diversification, and bioisosteric fragment merging (FM) experiments.

| | Samples | Novelty (%) | Uniqueness (%) |
|---|---|---|---|
| ***ShEPhERD*-GDB17** | $P(\boldsymbol{x}_1, \boldsymbol{x}_2)$ | 96.4 | 99.5 |
| | $P(\boldsymbol{x}_1, \boldsymbol{x}_3)$ | 96.7 | 99.6 |
| | $P(\boldsymbol{x}_1, \boldsymbol{x}_4)$ | 96.0 | 99.9 |
| | $P(\boldsymbol{x}_1 | \boldsymbol{x}_2)$ | 99.4 | 97.9 |
| | $P(\boldsymbol{x}_1 | \boldsymbol{x}_3)$ | 99.3 | 97.6 |
| | $P(\boldsymbol{x}_1 | \boldsymbol{x}_4)$ | 99.0 | 96.0 |
| ***ShEPhERD*-MOSES-aq** | $P(\boldsymbol{x}_1, \boldsymbol{x}_3, \boldsymbol{x}_4)$ | 100.0 | 100.0 |
| NP 1 | $P(\boldsymbol{x}_1 | \boldsymbol{x}_3, \boldsymbol{x}_4)$ | 100.0 | 99.7 |
| NP 2 | $P(\boldsymbol{x}_1 | \boldsymbol{x}_3, \boldsymbol{x}_4)$ | 100.0 | 100.0 |
| NP 3 | $P(\boldsymbol{x}_1 | \boldsymbol{x}_3, \boldsymbol{x}_4)$ | 100.0 | 100.0 |
| 1iep (lowest energy) | $P(\boldsymbol{x}_1 | \boldsymbol{x}_3, \boldsymbol{x}_4)$ | 100.0 | 100.0 |
| 1iep (pose) | $P(\boldsymbol{x}_1 | \boldsymbol{x}_3, \boldsymbol{x}_4)$ | 100.0 | 100.0 |
| 3eml (lowest energy) | $P(\boldsymbol{x}_1 | \boldsymbol{x}_3, \boldsymbol{x}_4)$ | 100.0 | 100.0 |
| 3eml (pose) | $P(\boldsymbol{x}_1 | \boldsymbol{x}_3, \boldsymbol{x}_4)$ | 100.0 | 100.0 |
| 3ny8 (lowest energy) | $P(\boldsymbol{x}_1 | \boldsymbol{x}_3, \boldsymbol{x}_4)$ | 100.0 | 100.0 |
| 3ny8 (pose) | $P(\boldsymbol{x}_1 | \boldsymbol{x}_3, \boldsymbol{x}_4)$ | 100.0 | 100.0 |
| 4rlu (lowest energy) | $P(\boldsymbol{x}_1 | \boldsymbol{x}_3, \boldsymbol{x}_4)$ | 99.5 | 100.0 |
| 4rlu (pose) | $P(\boldsymbol{x}_1 | \boldsymbol{x}_3, \boldsymbol{x}_4)$ | 99.8 | 100.0 |
| 4unn (lowest energy) | $P(\boldsymbol{x}_1 | \boldsymbol{x}_3, \boldsymbol{x}_4)$ | 100.0 | 100.0 |
| 4unn (pose) | $P(\boldsymbol{x}_1 | \boldsymbol{x}_3, \boldsymbol{x}_4)$ | 100.0 | 100.0 |
| 5mo4 (lowest energy) | $P(\boldsymbol{x}_1 | \boldsymbol{x}_3, \boldsymbol{x}_4)$ | 100.0 | 100.0 |
| 5mo4 (pose) | $P(\boldsymbol{x}_1 | \boldsymbol{x}_3, \boldsymbol{x}_4)$ | 100.0 | 100.0 |
| 7l11 (lowest energy) | $P(\boldsymbol{x}_1 | \boldsymbol{x}_3, \boldsymbol{x}_4)$ | 100.0 | 100.0 |
| 7l11 (pose) | $P(\boldsymbol{x}_1 | \boldsymbol{x}_3, \boldsymbol{x}_4)$ | 100.0 | 100.0 |
| Bioisosteric FM | $P(\boldsymbol{x}_1 | \boldsymbol{x}_3, \boldsymbol{x}_4)$ | 100.0 | 100.0 |

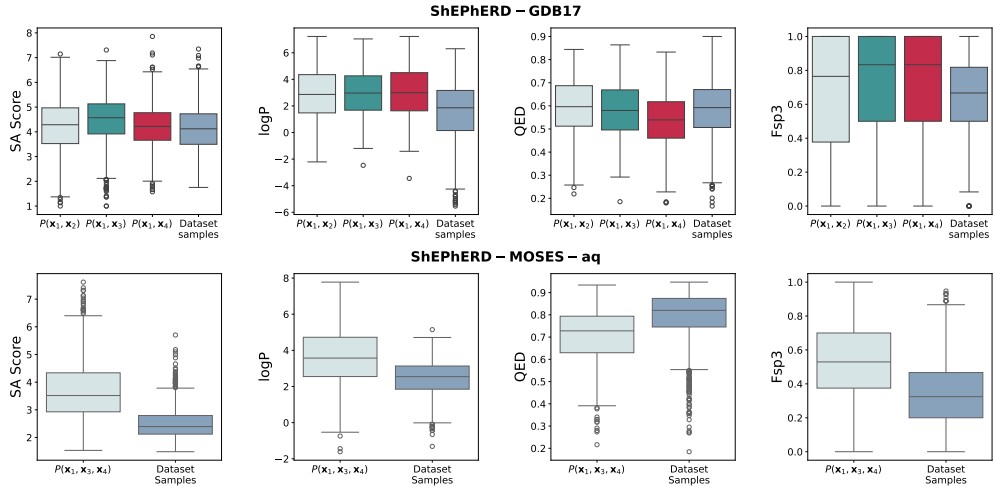

Figure 16: Distributions of molecular properties for unconditionally-generated molecules compared to samples from the respective datasets. SA Score is the synthetic accessibility score, logP is the octanol-water partition coefficient, QED is a measure of drug-likeness, and Fsp3 is the fraction of sp3 carbons in a molecule. (**Top**) The distributions relevant to *ShEPhERD*-GDB17. The plots show the property distributions for the valid molecules amongst 1000 *ShEPhERD*-generated samples from $P(\boldsymbol{x}_1, \boldsymbol{x}_2)$, $P(\boldsymbol{x}_1, \boldsymbol{x}_3)$, and $P(\boldsymbol{x}_1, \boldsymbol{x}_4)$. "Dataset Samples" refers to 1000 randomly sampled molecules from the *ShEPhERD*-GDB17 dataset. (**Bottom**) The same distributions relevant to *ShEPhERD*-MOSES-aq. The plots show the property distributions for the valid molecules amongst 1000 *ShEPhERD*-generated samples from $P(\boldsymbol{x}_1, \boldsymbol{x}_3, \boldsymbol{x}_4)$. "Dataset Samples" refers to 1000 randomly sampled molecules from the *ShEPhERD*-MOSES-aq dataset.

### A.5 Calculating interaction profiles from 3D molecular structures

#### A.5.1 Shapes/surfaces

We represent the shape of a molecule as a point cloud sampled from the solvent accessible surface with probe radius $r_{\text{probe}}$. Here, we describe how to extract $x_2 = S_2 \in \mathbb{R}^{n_2 \times 3}$ from $x_1 = (a, C)$ with $C \in \mathbb{R}^{n_1 \times 3}$.

First, we sample points on the surfaces of $n_1$ unit spheres using Fibonacci sampling. The number of sampled points for each sphere is $25(r_{a[k]}/1.7)^2$ where $r_{a[k]}$ is the van der Waals (vdW) radius of the $k$th atom in Å. The points are then scaled to have norm $r_{a[k]} + r_{\text{probe}}$, and the sampled spheres are translated to their respective atomic coordinates in $C$. Points that fall within $r_{a[k]} + r_{\text{probe}}$ of any other atom are filtered out. Next, we use Open3D (Zhou et al., 2018) to generate a `TriangleMesh` using the ball pivoting algorithm with a ball radius of 1.2. Finally, we sample a surface point cloud with approximately evenly spaced points using Open3D's `sample_points_poisson_disk` method. This final step is stochastic. We used $r_{\text{probe}} = 0.6$ for the modeling in this work, but tuning of similarity scoring function parameters (App. A.6) originally used $r_{\text{probe}} = 1.2$.

#### A.5.2 Electrostatic potential surfaces

We obtain the electrostatic potential (ESP) surface by computing the Coulombic potential at each point on a surface point cloud. Here, we describe how to extract $x_3 = (S_3, v)$ from $x_1 = (a, C)$.

We obtain $S_3 \in \mathbb{R}^{n_3 \times 3}$ using the same procedure as for $S_2$. We then obtain the partial charges $q \in \mathbb{R}^{n_1}$ from xTB by running a single point calculation on $x_1$. Through internal testing, we found that using partial charges obtained from the cheaper MMFF94 to be of insufficient quality and detrimental to the performance of both *ShEPhERD* and the ESP similarity scoring function. Computing partial charges with xTB is also favorable, as single point calculations can be computed for arbitrary atomistic systems that aren't parameterized by the MMFF94 force field. Given the partial charges of each atom, we can compute the electrostatic potential at each surface point:

$$v[k] = \frac{1}{4\pi\epsilon_0} \sum_{j=1}^{n_3} \frac{q[k]}{\|r[k] - r[j]\|^2}$$

where $\epsilon_0$ is the vacuum permittivity with units $e^2(\text{eV} \cdot \text{Å})^{-1}$.

#### A.5.3 Pharmacophores

Pharmacophores $x_4 = (p, P, V)$ are abstracted representations of chemical substructures that are associated with common biochemical interactions. They can be directional or non-directional.

The directional pharmacophores considered in this work are:

- **Hydrogen bond acceptor (HBA)** – typically an electronegative nitrogen or oxygen with lone pairs. Vector(s) point from the heavy atom to the direction of a lone pair based on sp, sp2, or sp3 geometries.

- **Hydrogen bond donor (HBD)** – typically an electronegative nitrogen or oxygen with an attached hydrogen. Vector(s) point from the heavy atom to the attached hydrogen(s)

- **Aromatic ring** – aromatic rings can form special interactions such as $\pi$-$\pi$ stacking or cation-$\pi$ interactions. We position an aromatic pharmacophore at the centroid of an aromatic ring. Two antiparallel vectors point in opposite directions, orthogonal to the plane of the aromatic ring.

- **Halogen-carbon bond** – a halogen (F, Cl, Br, I) that is connected to a carbon can form halogen bonds that can display HBA/HBD-like properties. The vector points from the halogen and in the anti-parallel direction to the neighboring carbon.

The non-directional pharmacophores are:

- **Hydrophobe** – a sulfur atom, a halogen atom, or a group of atoms that are hydrophobic. We cluster groups of hydrophobic atoms such that distinct hydrophobes are $> 2$ Å apart. Note that aromatic rings are also hydrophobes.

- **Anion** – a negatively charged atom or the centroid of a common anionic group.
- **Cation** – a positively charged atom or the centroid of a common cationic group.
- **Zinc binder** – a functional group capable of coordinating with a zinc ion.

We use SMARTS patterns from Pharmer, RDKit, and Pmapper (Koes & Camacho, 2011; Landrum et al., 2006a; Kutlushina et al., 2018) to identify pharmacophores in a molecule, with three notable changes: 1) feature definitions of positive/negative *ionizable* are altered to be strictly anions/cations; 2) all hydrophobes within 2 Å are clustered using Berenger & Tsuda (2023)'s protocol; and 3) separate pharmacophore identities for halogens were made due to their unique interaction geometry (e.g., sigma holes) that warranted a distinction from HBA/HBD pharmacophores (Lin & MacKerell Jr, 2017; Politzer et al., 2013). Most of the pharmacophore extraction utilizes altered code from RD-Kit (`Features.FeatDirUtilsRD` and `Features.ShowFeats` modules) (Landrum et al., 2006b). Some pharmacophores can be associated with two or more vectors (e.g., a carbonyl group has strong HBA qualities in the directions of the two lone pairs), but we average these together for ease of modeling in this work – except for aromatic rings where we do not average but choose one randomly, instead. Zinc binders, anions, and cations are rare in our datasets which significantly decreases the probability of generating these pharmacophores with *ShEPhERD*. Examples of each type of pharmacophore are shown in Fig. 17.

The placement of pharmacophore positions $P$ are either at the location of the identified atom or at the centroid of a larger substructure – depending on the SMARTS feature definition. For example, some Zn binder SMARTS feature definitions from RDKit place different weights on certain atoms (Fig. 17). Each pharmacophore is associated with a vector $V$ and can either be a unit vector ($|V[k]| = 1$) for directional pharmacophores or the zero vector ($|V[k]| = 0$) for directionless pharmacophores. For directional pharmacophores, a unit vector in the direction of a potential interaction (e.g., in the direction of a hydrogen for an HBD or a lone pair for an HBA) is extracted. It is common for multiple pharmacophores to be located at the same position. For example, aromatic rings and halogen-carbon bonds are also hydrophobes (Fig. 17), and hydroxyl groups are both HBDs and HBAs.

## A.6 PARAMETERIZATION OF 3D SIMILARITY SCORING FUNCTIONS

Recall from Sec. 3 that for two point clouds $Q_A$ and $Q_B$ we model each point $r_k$ as an isotropic Gaussian in $\mathbb{R}^3$ (Grant & Pickup, 1995; Grant et al., 1996). We compute the overlap between them $O_{A,B}$ with the first order Gaussian overlap. The Tanimoto similarity function $\text{sim}^*(Q_A, Q_B) \in [0, 1]$ is used to define a similarity of two point clouds based on their overlaps:

$$O_{A,B} = \sum_{a \in Q_A} \sum_{b \in Q_B} w_{a,b} \left(\frac{\pi}{2\alpha}\right)^{\frac{3}{2}} \exp\left(-\frac{\alpha}{2}\|r_a - r_b\|^2\right)$$

$$\text{sim}^*(Q_A, Q_B) = \frac{O_{A,B}}{O_{A,A} + O_{B,B} - O_{A,B}}$$

Here, $\alpha$ parameterizes the Gaussian width and $w_{a,b}$ is a weighting factor that is tuned for each representation. Note that $n_{Q_A}$ need not equal $n_{Q_B}$, generally.

Since 3D similarities are sensitive to SE(3) transformations of $Q_A$ with respect to $Q_B$, their optimal alignment can be defined as:

$$\text{sim}(Q_A, Q_B) = \max_{R,t} \text{sim}^*(RQ_A^T + t, Q_B)$$

where $R \in SO(3)$ and $t \in T(3)$. The optimization landscape is non-convex and noisy. For global optimizations (e.g., aligning random 3D molecules from the dataset to a target natural product), we optimize the alignment of $Q_A$ with respect to $Q_B$ with $N$ different initializations of $(R, t)$. We found that $N = 50$ is sufficient for convergence. For local optimizations (e.g., aligning a 3D molecule generated by *ShEPhERD* to a target natural product), we use the original alignment (e.g., the pose natively generated by *ShEPhERD*) as the only initialization and optimize directly via gradient descent.

For initializations of *global* alignment optimization, $t$ is set so that the center of mass (COM) of $Q_A$ is aligned with the COM of $Q_B$. One of the initializations uses the identity $R = I$ (to keep the initial orientation). We also use alignments to four orientations relative to the principal moments

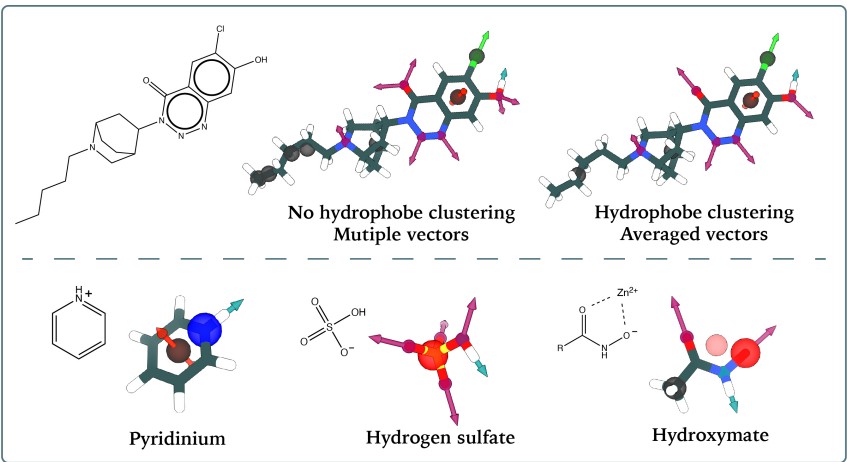

Figure 17: Examples of each type of pharmacophore. (**Top**) A molecule is shown that contains common pharmacophores, including hydrophobes (grey), HBA (purple), HBD (light blue), aromatic rings (orange), and halogen-carbon bonds (green). The left 3D structure highlights how multiple vectors are used for HBAs when they have multiple lone pairs, while the right structure shows these vectors averaged into a single direction (not shown here but the same applies for HBDs). Additionally, the clustering scheme from (Berenger & Tsuda, 2023) is used to reduce the number of hydrophobes. This molecule is an slightly altered example from `https://greglandrum.github.io/rdkit-blog/posts/2023-02-24-using-feature-maps.html`. (**Bottom**) Examples are provided for pharmacophores that are rare in our datasets: cations, anions, and zinc binders. The pyridinium structure features a cation (large dark blue sphere) on the nitrogen atom. The hydrogen sulfate example shows an anion (large red sphere), with the SMARTS pattern covering the entire group, placing the feature at the centroid. The hydroxamate example contains both a cation on the oxygen and a zinc binder (pink sphere). Although the entire hydroxamate group is classified as a Zn binder according to the SMARTS pattern, the pharmacophore weights are specifically assigned to the two oxygens capable of coordinating a zinc ion, which places the pharmacophore between the oxygens rather than at the group centroid.

of inertia of $\boldsymbol{Q}_B$ (the positive and negative orientations for the two largest principal components). For all other initializations when $N > 5$, we use Fibonacci sampling to obtain evenly spaced out rotations in $SO(3)$.

We use unit quaternions to represent $\boldsymbol{R}$ to decrease the number of parameters and increase optimization stability. We achieve optimal alignment through automatic differentiation and use Adam optimizer for a maximum of 200 steps with a learning rate of 0.1. By implementing in PyTorch and Jax, we run batched computations across the $N$ initializations.

### A.6.1 SHAPE SIMILARITY SCORING FUNCTION

Recall that we follow Adams & Coley (2022) in defining the *volumetric* shape similarity between two atomic point clouds $\boldsymbol{C}_A$ and $\boldsymbol{C}_B$ as $\mathrm{sim}^*_{\mathrm{vol}}(\boldsymbol{C}_A, \boldsymbol{C}_B)$ with $w_{a,b} = 2.7$ and $\alpha = 0.81$. The *surface* shape similarity between two surfaces $\boldsymbol{S}_A$ and $\boldsymbol{S}_B$ is defined as $\mathrm{sim}^*_{\mathrm{surf}}(\boldsymbol{S}_A, \boldsymbol{S}_B)$ with $w_{a,b} = 1$, and $\alpha = \Psi(n_2)$. Here, $\Psi$ is a function fitted to $\mathrm{sim}^*_{\mathrm{vol}}$ depending on the choice of $n_2$.

We fit $\Psi(n_2)$ by minimizing RMSE between $\mathrm{sim}^*_{\mathrm{vol}}$ and $\mathrm{sim}^*_{\mathrm{surf}}$ for $n_2 \in [50, 100, 150, 200, 300, 400]$ on 1K randomly sampled molecules from `https://www.kaggle.com/datasets/art3mis/chembl22` with the number of heavy atoms ranging from 6 to 62. For parameter fitting, we used a probe radius of 1.2 Å when sampling the surfaces. This differs from the 0.6 Å probe radius that we ultimately used when modeling and scoring the similarities for $\boldsymbol{x}_2$ and $\boldsymbol{x}_3$ in our experiments, but we do not expect this discrepancy to meaningfully affect our analyses. The conformations were generated with RDKit ETKDG and optimized with MMFF94. Results and correlations from tuned $\alpha$'s are found in Fig. 18. We do not expect, nor want, perfect correlation with the volumetric similarity since we aim to decouple the shape representation from the atomic coordinates. $\Psi(n_2)$ is defined as an quadratic interpolation function from SciPy used to fit the optimal $\alpha$'s found in Table 6. Note that for *surface* similarity, we require $n_{S_A} = n_{S_B}$ due to the tuned $\alpha$. We show the effects of stochasticity in Figure 19.

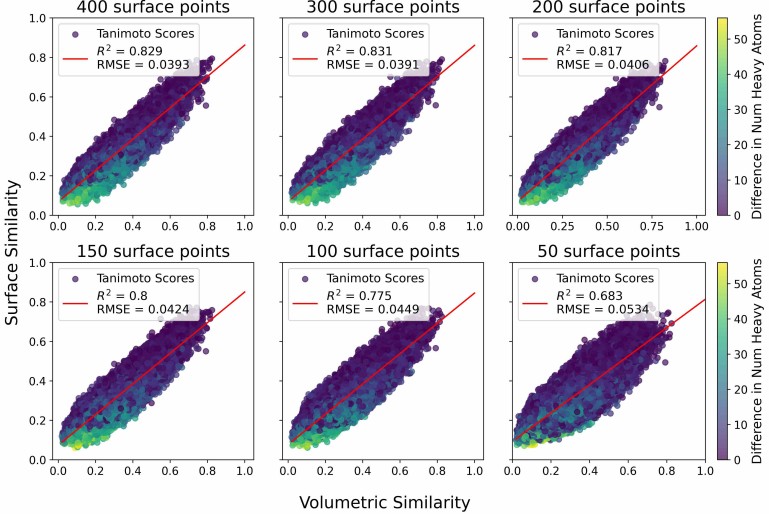

Figure 18: The correlations between surface similarity and volumetric similarity after tuning the Gaussian width parameter $\alpha$ to minimize RMSE between the two scoring functions.

Table 6: Tuned Gaussian width parameters ($\alpha$) for the surface similarity scoring function. Larger $\alpha$ corresponds to smaller width.

| Number of surface points ($n_2$) | 50 | 100 | 150 | 200 | 300 | 400 |
|---|---|---|---|---|---|---|
| Gaussian width ($\alpha$) | 0.6011 | 0.8668 | 1.022 | 1.118 | 1.216 | 1.258 |

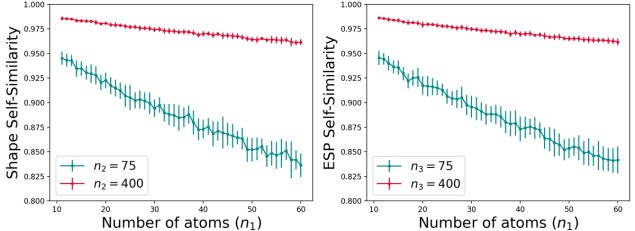

Figure 19: The effect of stochastic surface sampling on the surface and ESP scores, as measured by the average self-similarity as a function of the number of atoms $n_1$. For each of $\sim 900$ (xTB-relaxed) generated compounds from *ShEPhERD* trained on *ShEPhERD*-GDB17, we repeatedly extract the true shape $\boldsymbol{x}_2$ or ESP $\boldsymbol{x}_3$ interaction profile 5 times, score the similarity between pairs of extracted profiles, and compute the average similarity. We call this quantity the shape or ESP "self-similarity" for a given molecule. We plot the average self-similarity as a function of the number of atoms in the molecule $n_1$ when extracting $\boldsymbol{x}_2$ or $\boldsymbol{x}_3$ with either 75 or 400 surface points. As the surfaces are more densely sampled — either by reducing $n_1$ for fixed $n_2$ or $n_3$, or increasing $n_2$ or $n_3$ for fixed $n_1$ — the average self-similarity approaches 1.0. Recall that we use 75 points when evaluating the self-consistency of *ShEPhERD*'s jointly generated $(\boldsymbol{x}_1, \boldsymbol{x}_2)$ and $(\boldsymbol{x}_1, \boldsymbol{x}_3)$ samples, but use 400 points when scoring similarities between two molecules in all conditional experiments. Using 400 points for shape and ESP similarity scoring gives a much more precise estimate of interaction similarity.

### A.6.2 ESP SURFACE SIMILARITY SCORING FUNCTION

The similarity between two electrostatic potential (ESP) surfaces $\text{sim}^*_{\text{ESP}}(\boldsymbol{x}_{3,A}, \boldsymbol{x}_{3,B})$ is defined with an overlap function:

$$O^{\text{ESP}}_{A,B} = \sum_{a \in \boldsymbol{Q}_A} \sum_{b \in \boldsymbol{Q}_B} \left(\frac{\pi}{2\alpha}\right)^{\frac{3}{2}} \exp\left(-\frac{\alpha}{2}\|\boldsymbol{r}_a - \boldsymbol{r}_b\|^2\right) \exp\left(-\frac{\|\boldsymbol{v}_A[a] - \boldsymbol{v}_B[b]\|^2}{\lambda}\right)$$

where $\alpha = \Psi(n_3)$, and $\lambda = \frac{0.3}{(4\pi\epsilon_0)^2}$. $\epsilon_0$ is the permittivity of vacuum with units $e^2(\text{eV} \cdot \text{Å})^{-1}$. We chose $\lambda$ through manual inspection of optimal alignments. Fig. 19 shows the effects of stochasticity. Others have also considered different formulations of ESP scoring functions to characterize bioisosteres (Good et al., 1992; Cleves et al., 2019; Bolcato et al., 2022; Osman & Arabi, 2024).

### A.6.3 PHARMACOPHORE SIMILARITY SCORING FUNCTION

For pharmacophore similarity, we follow the the vector weighting formulation of PheSA (Wahl, 2024) and define the overlap similarity between two sets of pharmacophores $\boldsymbol{x}_{4,A}$ and $\boldsymbol{x}_{4,B}$ as:

$$O^{\text{pharm}}_{A,B;m} = \sum_{a \in \boldsymbol{Q}_{A,m}} \sum_{b \in \boldsymbol{Q}_{B,m}} w_{a,b;m} \left(\frac{\pi}{2\alpha_m}\right)^{\frac{3}{2}} \exp\left(-\frac{\alpha_m}{2}\|\boldsymbol{r}_a - \boldsymbol{r}_b\|^2\right)$$

$$\text{sim}^*_{\text{pharm}}(\boldsymbol{x}_{4,A}, \boldsymbol{x}_{4,B}) = \frac{\sum_{m \in \mathcal{M}} O_{A,B;m}}{\sum_{m \in M} O_{A,A;m} + O_{B,B;m} - O_{A,B;m}}$$

where $\mathcal{M}$ is the set of all pharmacophore types ($|\mathcal{M}| = N_p$), $\alpha_m = \Omega(m)$ where $\Omega$ maps each pharmacophore type to a Gaussian width using the parameters from Pharao (Taminau et al., 2008). $w_{a,b;m}$ is defined as:

$$w_{a,b;m} = \begin{cases} 1 & \text{if } m \text{ is non-directional,} \\ \frac{\boldsymbol{V}[a]^\top_m \boldsymbol{V}[b]_m + 2}{3} & \text{if } m \text{ is directional.} \end{cases}$$

We take the absolute value of $\boldsymbol{V}[a]^\top_m \boldsymbol{V}[b]_m$ for aromatic groups as we assume their $\pi$ interaction effects are symmetric across their plane. Note that Pharao does not define the halogen-carbon bond pharmacophore so we assign $\alpha = 1.0$.

## A.7 Additional details on model design, training protocols, and sampling

**Feature scaling**. Hoogeboom et al. (2022) and Peng et al. (2023) found that linearly scaling certain one-hot features in 3D molecular DDPMs can improve model performance as measured via sample validity, presumably because increasing/decreasing the magnitude of certain features causes them to become resolved earlier/later in the denoising process. We loosely follow these prior works to scale $a$ and $f$ by 0.25 and $B$ by 1.0. We additionally scale $v$ and $p$ by 2.0. Finally, we scale the pharmacophore directional vectors $V$ by 2.0. However, we did not rigorously tune these scaling factors. We also did not explore the use of different noise schedules for different variables, although this strategy has also been found to be helpful for 3D molecule DDPMs (Peng et al., 2023).

**Noise schedule**. *ShEPhERD* uses the same noise schedule for all diffusion/denoising processes. We choose the variance schedule $\sigma_t^2$ to be a weighted combination of the linear schedule used by RFDiffusion (Watson et al., 2023) and a cosine schedule introduced by Nichol & Dhariwal (2021). Fig. 20 plots our noise schedule in terms of $\sigma_t$, $\alpha_t$, $\overline{\sigma}_t$, and $\overline{\alpha}_t$ for $t \in [1, T]$. We use $T = 400$ for all *ShEPhERD* models.

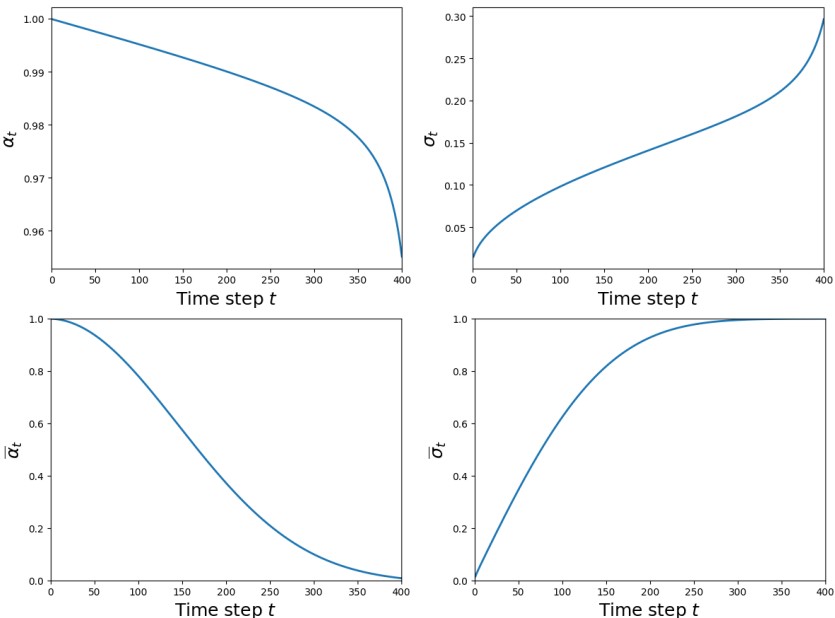

Figure 20: Noise schedule used by *ShEPhERD* for all forward noising processes in terms of $\sigma_t$, $\alpha_t$, $\overline{\sigma}_t$, and $\overline{\alpha}_t$ for $t \in [1, 400]$.

### A.7.1 Further details on *ShEPhERD*'s denoising modules

**"E3NN-style" vs. "EGNN-style" coordinate predictions**. As described in section 3.3, we predict the $l{=}1$ coordinate noises $\hat{\epsilon}_{S_2}^{(t)}$, $\hat{\epsilon}_{S_3}^{(t)}$, and $\hat{\epsilon}_P^{(t)}$ from the corresponding $l{=}1$ node features $\tilde{z}_i^{(t)}$ directly with equivariant feed forward networks that appear in EquiformerV2 and other E3NN-based architectures. In these "E3NN-style" coordinate predictions, the noise for each node $\epsilon_i[k]$ if predicted from *only* the $l{=}1$ feature $\tilde{z}_i[k]$ of that node. While each $\tilde{z}_i[k]$ implicitly contains information about the node's surrounding environment (e.g, neighboring nodes) due to *ShEPhERD*'s joint module, the actual denoising step is not defined with respect to the positions of the neighboring nodes. This notably differs from Hoogeboom et al. (2022)'s original formulation of equivariant DDPMs for 3D molecule generation, which used "EGNN-style" (Satorras et al., 2021) coordinate predictions to denoise the molecule's coordinates $C$. For a single-layer EGNN, the coordinate prediction step takes the form (using our notation):

$$C[k]' = C[k] + \sum_{j \neq k} \left( \frac{C[k] - C[j]}{d_{kj} + 1} \right) \phi_C \left( z[k], z[j], d_{kj} \right) \; ; \; d_{kj} = \left\| C[k] - C[j] \right\|_2$$

where $\phi_C$ is an MLP and $C[k]'$ can be interpreted as an updated coordinate for the node $k$, which is equivariant to E(3) transformations of $C$. With this updated coordinate, the noise $\hat{\epsilon}_i[k]$ may be obtained by computing $\hat{\epsilon}_i[k] = C[k]' - C[k]$. In "EGNN-style" coordinate predictions, therefore, the predicted coordinate/noise for node $k$ is defined with respect to the other nodes in the 3D graph.

We hypothesize that E3NN-style coordinate predictions are sufficient for predicting $\hat{\epsilon}_{S_2}^{(t)}$, $\hat{\epsilon}_{S_3}^{(t)}$, and $\hat{\epsilon}_P^{(t)}$ (e.g., the $l{=}1$ coordinates of $x_2$, $x_3$, and $x_4$), as these noise predictions do not need to be so highly sensitive to the positions of neighboring nodes. However, for the coordinates $C$ of the 3D molecule $x_1$, inter-node geometries (e.g., bond lengths and angles) are quite relevant to the quality of the molecule's conformation. Hence, we hypothesize that it may be favorable to use EGNN-style coordinate predictions to obtain $\hat{\epsilon}_C$. However, EGNN-style coordinate predictions are less expressive than E3NN-style coordinate predictions, as the difference $C[k]' - C[k]$ is just a linear combination $\sum_{j \neq k} w_j(C[k] - C[j])$ with (learnt) weights $w_j$. To denoise $C$, we thus choose to use both E3NN- and EGNN-style coordinate predictions. We first predict a set of updated coordinates $C'^{(t)}$ from $C^{(t)}$ using E3NN-style coordinate predictions, and then refine $C'^{(t)}$ with EGNN-style coordinate predictions to obtain $C''^{(t)}$. The noise $\hat{\epsilon}_C$ is then computed as $\hat{\epsilon}_C = C''^{(t)} - C^{(t)}$. Algorithm 1 outlines this process within the context of the entire denoising module for $x_1$.

**Forward-noising and denoising the covalent bond graph $B$.** Multiple works have found that jointly diffusing the 2D covalent bond graph with the 3D atomic coordinates helps improve sample quality for 3D molecule DDPMs (Peng et al., 2023; Vignac et al., 2023). While using *discrete* diffusion for bond/atom types has been found to further improve sample quality compared to diffusing/denoising continuous one-hot representations of bond/atom types, we choose to use continuous representations of bond/atom types to emphasize overall modeling simplicity, consistency, and extensibility. In this section, we clarify details regarding the forward and reverse processes for $B$.

We represent $B \in \mathbb{R}^{n_1 \times n_1 \times 5}$ as an $n_1 \times n_1$ adjacency matrix attributed with one-hot features of the covalent bond order between each pair of atoms in $x_1$. The bond types include single, double, triple, and aromatic bonds, as well as an extra bond type denoting the *absence* of a covalent bond. It is important to note that we treat bonds between pairs of atoms as *undirected* edges by symmetrizing $B^{(t)}$ in both the forward noising and reverse denoising processes. Namely, we only forward-noise and denoise the upper triangle of $B^{(t)}$, and copy the upper triangle to the lower triangle. We mask-out the diagonal of $B^{(t)}$ so that self-loops are not modeled.

To maintain permutation invariance when predicting the noise $\hat{\epsilon}_B[k, j]$ for the bond between atoms $k$ and $j$ from their $l{=}0$ node representations $z_1[k]$ and $z_1[j]$, we use a permutation-invariant MLP:

$$\hat{\epsilon}_B[k, j] = \frac{1}{2}\left( b_{kj} + b_{jk} \right)$$

$$b_{kj} = \text{MLP}_B\left( \left[ B^{(t)}[k, j], f_{RBF}(d_{kj}), z_1[k], z_1[j] \right] \right)$$

$$b_{jk} = \text{MLP}_B\left( \left[ B^{(t)}[j, k], f_{RBF}(d_{jk}), z_1[j], z_1[k] \right] \right)$$

where $d_{kj} = d_{jk} = ||C^{(t)}[k] - C^{(t)}[j]||$, $f_{RBF}$ is a radial basis function expansion using Gaussian basis functions, and $[., .]$ denotes concatenation in the feature dimension.

A.7.2 TRAINING AND SAMPLING PROCEDURES

Algorithms 2, 3, 4, 5 outline the forward pass of *ShEPhERD*'s denoising network and detail how we train and sample from *ShEPhERD*. Algorithm 1 details the forward pass of *ShePhERD*'s denoising module for $x_1$. The denoising modules for $x_2$, $x_3$, and $x_4$ are simpler and described in section 3.

**Training details**. We train *ShEPhERD* with the Adam optimizer using a constant learning rate of 3e-4 and an effective batch size ranging from 40 to 48. We clip gradients that have norm exceeding 5.0. App. A.11 describes overall training times, and App. A.7.3 lists all hyperparameters.

Each training example requires sampling a time step $t \in [1, T]$ (we use $T = 400$). Rather than uniformly sampling $t$ within that range, we upsample $t$ in the range $[50, 250]$ to accelerate model convergence, since this interval captures the most important part of the denoising trajectory. Specifically, we uniformly sample $t \in [50, 250]$ 75% of the time, $t \in [0, 50)$ 7.5% of the time, and $t \in (250, 400]$ 17.5% of the time.

---

**Algorithm 1** Denoising Module for $x_1$

---

1: **Given:**
2: $\quad (z_1, \tilde{z}_1)$
3: $\quad C^{(t)}$
4: $\quad B^{(t)}$

5: **Predict noise for $l=0$ node (atom) features:**
6: $\quad (\hat{\epsilon}_a, \hat{\epsilon}_f) = \text{MLP}_{a,f}(z_1)$

7: **Predict noise for $l=0$ edge (bond) features, using permutation-invariant MLP:**
$\qquad\qquad\qquad\qquad \triangleright \text{RBF indicates radial basis function expansion with Gaussian functions}$
8: $\quad b_{kj} = \text{MLP}_B(B^{(t)}[k, j], \text{RBF}(||C^{(t)}[k] - C^{(t)}[j]||), z_1[k], z_1[j])$
9: $\quad b_{jk} = \text{MLP}_B(B^{(t)}[j, k], \text{RBF}(||C^{(t)}[j] - C^{(t)}[k]||), z_1[k], z_1[j])$
10: $\quad \hat{\epsilon}_B[k, j] = \frac{1}{2}(b_{kj} + b_{jk})$
11: $\quad \hat{\epsilon}_B[j, k] = \frac{1}{2}(b_{kj} + b_{jk})$

12: **Predict noise for coordinates $C$:**
13: $\quad \Delta C_{E3NN} = \text{E3NN\_feed\_forward}(\tilde{z}_1)$
14: $\quad C'^{(t)} = C^{(t)} + \Delta C_{E3NN}$
15: $\quad C''^{(t)} = \text{EGNN}(z_1, C'^{(t)})$
16: $\quad \hat{\epsilon}_C^{(t)} = C''^{(t)} - C^{(t)}$

17: **return** $\hat{\epsilon}_a^{(t)}, \hat{\epsilon}_f^{(t)}, \hat{\epsilon}_B^{(t)}, \hat{\epsilon}_C^{(t)}$

---

---

**Algorithm 2** Forward Pass of *ShEPhERD*'s Denoising Network $\boldsymbol{\eta}$

---

1: **Given:**
2:      $\boldsymbol{x}_1^{(t)} = (\boldsymbol{a}^{(t)}, \boldsymbol{f}^{(t)}, \boldsymbol{C}^{(t)}, \boldsymbol{B}^{(t)})$
3:      $\boldsymbol{x}_2^{(t)} = (\boldsymbol{S}_2^{(t)})$
4:      $\boldsymbol{x}_3^{(t)} = (\boldsymbol{S}_3^{(t)}, \boldsymbol{v}^{(t)})$
5:      $\boldsymbol{x}_4^{(t)} = (\boldsymbol{p}^{(t)}, \boldsymbol{P}^{(t)}, \boldsymbol{V}^{(t)})$
6:      $t$ (time step)

7: **Add virtual node to each $\boldsymbol{x}_i$ system with positions at the COM of $\boldsymbol{C}^{(t)}$ (0 if centered):**
8: $\boldsymbol{x}_1^{(t)}, \boldsymbol{x}_2^{(t)}, \boldsymbol{x}_3^{(t)}, \boldsymbol{x}_4^{(t)} = \texttt{add\_virtual\_nodes}(\boldsymbol{x}_1^{(t)}, \boldsymbol{x}_2^{(t)}, \boldsymbol{x}_3^{(t)}, \boldsymbol{x}_4^{(t)})$

9: **Embedding modules: embed each $\boldsymbol{x}_i^{(t)}$ into $l = 0$ and $l = 1$ node features:**
10: $\boldsymbol{z}_1, \tilde{\boldsymbol{z}}_1 = \phi_1(\boldsymbol{x}_1^{(t)}, t)$                                             ▷ EquiformerV2 module
11: $\boldsymbol{z}_2, \tilde{\boldsymbol{z}}_2 = \phi_2(\boldsymbol{x}_2^{(t)}, t)$                                             ▷ EquiformerV2 module
12: $\boldsymbol{z}_3, \tilde{\boldsymbol{z}}_3 = \phi_3(\boldsymbol{x}_3^{(t)}, t)$                                             ▷ EquiformerV2 module
13: $\boldsymbol{z}_4, \tilde{\boldsymbol{z}}_4 = \phi_4(\boldsymbol{x}_4^{(t)}, t)$                                             ▷ EquiformerV2 module

14: **Joint module:**
15: *Collate each system to form a heterogeneous 3D graph with $n_1 + n_2 + n_3 + n_4$ nodes:*
16: $\boldsymbol{C}_{hetero} = \texttt{concat}([\boldsymbol{C}^{(t)}, \boldsymbol{S}_2^{(t)}, \boldsymbol{S}_3^{(t)}, \boldsymbol{P}^{(t)}], \dim = 0)$
17: $\boldsymbol{z}_{hetero} = \texttt{concat}([\boldsymbol{z}_1, \boldsymbol{z}_2, \boldsymbol{z}_3, \boldsymbol{z}_4], \dim = 0)$
18: $\tilde{\boldsymbol{z}}_{hetero} = \texttt{concat}([\tilde{\boldsymbol{z}}_1, \tilde{\boldsymbol{z}}_2, \tilde{\boldsymbol{z}}_3, \tilde{\boldsymbol{z}}_4], \dim = 0)$
19: *Encode heterogeneous graph and update node embeddings:*
20: $\boldsymbol{z}'_{hetero}, \tilde{\boldsymbol{z}}'_{hetero} = \phi_{joint}^{local}(\boldsymbol{C}_{hetero}, \boldsymbol{z}_{hetero}, \tilde{\boldsymbol{z}}_{hetero})$          ▷ EquiformerV2 module
21: $\boldsymbol{z}_{hetero} \mathrel{+}= \boldsymbol{z}'_{hetero}$
22: $\tilde{\boldsymbol{z}}_{hetero} \mathrel{+}= \tilde{\boldsymbol{z}}'_{hetero}$
23: *Pool $l = 1$ node embeddings across the nodes of each homogeneous sub-graph:*
24: $\tilde{\boldsymbol{z}}_1^{pool} = \texttt{sum}(\tilde{\boldsymbol{z}}_{hetero}[k] \; \forall k \text{ if } k \in \boldsymbol{x}_1^{(t)})$
25: $\tilde{\boldsymbol{z}}_2^{pool} = \texttt{sum}(\tilde{\boldsymbol{z}}_{hetero}[k] \; \forall k \text{ if } k \in \boldsymbol{x}_2^{(t)})$
26: $\tilde{\boldsymbol{z}}_3^{pool} = \texttt{sum}(\tilde{\boldsymbol{z}}_{hetero}[k] \; \forall k \text{ if } k \in \boldsymbol{x}_3^{(t)})$
27: $\tilde{\boldsymbol{z}}_4^{pool} = \texttt{sum}(\tilde{\boldsymbol{z}}_{hetero}[k] \; \forall k \text{ if } k \in \boldsymbol{x}_4^{(t)})$
28: *Embed global $l = 1$ code for entire system:*
29: $\tilde{\boldsymbol{z}}_{joint}^{global} = \texttt{concat}([\tilde{\boldsymbol{z}}_1^{pool}, \tilde{\boldsymbol{z}}_2^{pool}, \tilde{\boldsymbol{z}}_3^{pool}, \tilde{\boldsymbol{z}}_4^{pool}], \dim = -1)$
30: $\tilde{\boldsymbol{z}}_{joint}^{global} = \phi_{joint}^{global}(\tilde{\boldsymbol{z}}_{joint}^{global})$                          ▷ E3NN feed-forward network
31: *Apply E3NN/Equiformer's fully-connected tensor product (FCTP):*
32: $\boldsymbol{t}_{embed} = \texttt{sinusoidal\_positional\_encoding}(t)$
33: $\boldsymbol{z}_{joint}, \tilde{\boldsymbol{z}}_{joint} = \texttt{equiformer\_FCTP}([\boldsymbol{t}_{embed}, \tilde{\boldsymbol{z}}_{joint}^{global}], [\boldsymbol{t}_{embed}, \tilde{\boldsymbol{z}}_{joint}^{global}])$
34: *Residually update node embeddings:*
35: **for** each $i \in [1, 2, 3, 4]$ **do**
36:      $\boldsymbol{z}_i \mathrel{+}= \boldsymbol{z}_{joint}$
37:      $\tilde{\boldsymbol{z}}_i \mathrel{+}= \tilde{\boldsymbol{z}}_{joint}$
38: **end for**

39: **Denoising modules:**
40: $(\hat{\boldsymbol{\epsilon}}_a, \hat{\boldsymbol{\epsilon}}_f, \hat{\boldsymbol{\epsilon}}_C, \hat{\boldsymbol{\epsilon}}_B) = \texttt{denoise\_x1}(\boldsymbol{z}_1, \tilde{\boldsymbol{z}}_1)$
41: $(\hat{\boldsymbol{\epsilon}}_{S_2}) = \texttt{denoise\_x2}(\boldsymbol{z}_2, \tilde{\boldsymbol{z}}_2)$
42: $(\hat{\boldsymbol{\epsilon}}_{S_3}, \hat{\boldsymbol{\epsilon}}_v) = \texttt{denoise\_x3}(\boldsymbol{z}_3, \tilde{\boldsymbol{z}}_3)$
43: $(\hat{\boldsymbol{\epsilon}}_p, \hat{\boldsymbol{\epsilon}}_P, \hat{\boldsymbol{\epsilon}}_V) = \texttt{denoise\_x4}(\boldsymbol{z}_4, \tilde{\boldsymbol{z}}_4)$

44: **return** $(\hat{\boldsymbol{\epsilon}}_a, \hat{\boldsymbol{\epsilon}}_f, \hat{\boldsymbol{\epsilon}}_C, \hat{\boldsymbol{\epsilon}}_B, \hat{\boldsymbol{\epsilon}}_{S_2}, \hat{\boldsymbol{\epsilon}}_{S_3}, \hat{\boldsymbol{\epsilon}}_v, \hat{\boldsymbol{\epsilon}}_p, \hat{\boldsymbol{\epsilon}}_P, \hat{\boldsymbol{\epsilon}}_V)$

---

---

**Algorithm 3** Training Algorithm

---

1: **Given:**
2:  RDKit mol object `mol` containing 3D conformation of a molecule with explicit hydrogens, locally optimized with xTB.
3:  Noise schedule $\boldsymbol{\sigma_t}, \boldsymbol{\alpha_t}, \overline{\boldsymbol{\sigma}}_t, \overline{\boldsymbol{\alpha}}_t$ for $t \in [1, T]$.
4:  Denoising network $\boldsymbol{\eta}$ to be trained

5: **Obtain unnoised input molecule and its interaction profiles:**
6:  `mol = recenter_mol(mol)`     ▷ Recenter molecule to have an unweighted COM of **0**
7:  $\boldsymbol{a}, \boldsymbol{f}, \boldsymbol{C}, \boldsymbol{B} = $ `get_x1(mol)`
8:  `partial_charges = get_partial_charges_with_xtb(mol)`     ▷ single point calculation with xTB in implicit solvent or gas phase
9:  `vdw_radii = get_vdw_radii(`$\boldsymbol{a}$`)`     ▷ list of Van der Waals radii of each atom
10:  $\boldsymbol{S_2} = $ `get_solvent_surface(`$\boldsymbol{C}$`, vdw_radii, probe_radius = 0.6, n_2 = 75)`
11:  $\boldsymbol{S_3} = \boldsymbol{S_2}$
12:  $\boldsymbol{v} = $ `get_ESP_at_surface(`$\boldsymbol{S_3}, \boldsymbol{C}$`, partial_charges)`
13:  $\boldsymbol{p}, \boldsymbol{P}, \boldsymbol{V} = $ `get_pharmacophores(mol)`

14: **Linearly scale certain features:**
15:  $\boldsymbol{a} = $ `scale_a(`$\boldsymbol{a}$`)`
16:  $\boldsymbol{f} = $ `scale_f(`$\boldsymbol{f}$`)`
17:  $\boldsymbol{B} = $ `scale_B(`$\boldsymbol{B}$`)`
18:  $\boldsymbol{v} = $ `scale_v(`$\boldsymbol{v}$`)`
19:  $\boldsymbol{p} = $ `scale_p(`$\boldsymbol{p}$`)`
20:  $\boldsymbol{V} = $ `scale_V(`$\boldsymbol{V}$`)`

21: **Forward-noise input state:**
22:  Sample $t \in [1, T]$, and obtain the corresponding $\boldsymbol{\sigma_t}, \boldsymbol{\alpha_t}, \overline{\boldsymbol{\sigma}}_t$, and $\overline{\boldsymbol{\alpha}}_t$
23:  Sample independent noise $\boldsymbol{\epsilon_x} \sim \mathcal{N}(\boldsymbol{0}, \boldsymbol{1})$ for $\boldsymbol{x} = [\boldsymbol{a}, \boldsymbol{f}, \boldsymbol{C}, \boldsymbol{B}, \boldsymbol{S_2}, \boldsymbol{S_3}, \boldsymbol{v}, \boldsymbol{p}, \boldsymbol{P}, \boldsymbol{V}]$
24:  Subtract center of mass from $\boldsymbol{\epsilon_C}$
25:  Symmetrize the noise for $\boldsymbol{\epsilon_B}$
26:  Forward-noise each $\boldsymbol{x} \in [\boldsymbol{a}, \boldsymbol{f}, \boldsymbol{C}, \boldsymbol{B}, \boldsymbol{S_2}, \boldsymbol{S_3}, \boldsymbol{v}, \boldsymbol{p}, \boldsymbol{P}, \boldsymbol{V}]$ via: $\boldsymbol{x}^{(t)} = \overline{\boldsymbol{\alpha}}_t \boldsymbol{x} + \overline{\boldsymbol{\sigma}}_t \boldsymbol{\epsilon_x}$

27: **Perform forward-pass with ShEPhERD's denoising network:**
28:  $\boldsymbol{x}_1^{(t)} = (\boldsymbol{a}^{(t)}, \boldsymbol{f}^{(t)}, \boldsymbol{C}^{(t)}, \boldsymbol{B}^{(t)})$
29:  $\boldsymbol{x}_2^{(t)} = (\boldsymbol{S_2}^{(t)})$
30:  $\boldsymbol{x}_3^{(t)} = (\boldsymbol{S_3}^{(t)}, \boldsymbol{v}^{(t)})$
31:  $\boldsymbol{x}_4^{(t)} = (\boldsymbol{p}^{(t)}, \boldsymbol{P}^{(t)}, \boldsymbol{V}^{(t)})$
32:  $(\hat{\boldsymbol{\epsilon}}_a, \hat{\boldsymbol{\epsilon}}_f, \hat{\boldsymbol{\epsilon}}_C, \hat{\boldsymbol{\epsilon}}_B, \hat{\boldsymbol{\epsilon}}_{S_2}, \hat{\boldsymbol{\epsilon}}_{S_3}, \hat{\boldsymbol{\epsilon}}_v, \hat{\boldsymbol{\epsilon}}_p, \hat{\boldsymbol{\epsilon}}_P, \hat{\boldsymbol{\epsilon}}_V)$
     $= $ `ShEPhERD_forward(`$\boldsymbol{x}_1^{(t)}, \boldsymbol{x}_2^{(t)}, \boldsymbol{x}_3^{(t)}, \boldsymbol{x}_4^{(t)}, t$`)`

33: **Compute losses:**
34:  $l_x = ||\hat{\boldsymbol{\epsilon}}_{\boldsymbol{x}} - \boldsymbol{\epsilon_x}||^2$     $\forall \boldsymbol{x} \in [\boldsymbol{a}, \boldsymbol{f}, \boldsymbol{C}, \boldsymbol{B}, \boldsymbol{S_2}, \boldsymbol{S_3}, \boldsymbol{v}, \boldsymbol{p}, \boldsymbol{P}, \boldsymbol{V}]$
35:  $\mathcal{L} = \sum_{\boldsymbol{x}} l_{\boldsymbol{x}}$

36: **Optimize network parameters:**
37: Take gradient step w.r.t. the denoising network's parameters to minimize $\mathcal{L}$

---

---

**Algorithm 4** Sampling Algorithm for Unconditional Generation

---

1: **Given:**
2:     Trained ShEPhERD model $\boldsymbol{\eta_X}$ that learns $P(\boldsymbol{X})$ for $\boldsymbol{X} \subset \{\boldsymbol{x_1}, \boldsymbol{x_2}, \boldsymbol{x_3}, \boldsymbol{x_4}\}$.
3:     $n_1$                                                ▷ number of atoms to generate.
4:     (if $\boldsymbol{x_4} \in \boldsymbol{X}$) $n_4$                              ▷ number of pharmacophores to generate.
5:     Noise schedule $\boldsymbol{\sigma_t}, \boldsymbol{\alpha_t}, \overline{\boldsymbol{\sigma}}_t, \overline{\boldsymbol{\alpha}}_t$ for $t \in [1, T]$.
6:     `extra_noise_ts` – ordered list of time steps to add extra noise to coordinate states
7:     `extra_noise_scales` – list of scaling factors to apply to extra coordinate noise
8:     `harmonization_ts` – ordered list of time steps at which to perform harmonization.
9:     `harmonization_steps` – list of time horizons $\Delta t$ for each harmonization action.

10: **Sample initial states at** $t = T$ **from Gaussian priors:**
11:     Sample noises $\boldsymbol{x}^{(T)} \sim \mathcal{N}(\boldsymbol{0}, \boldsymbol{1})$ for $\boldsymbol{x} = [\boldsymbol{a}, \boldsymbol{f}, \boldsymbol{C}, \boldsymbol{B}, \boldsymbol{S_2}, \boldsymbol{S_3}, \boldsymbol{v}, \boldsymbol{p}, \boldsymbol{P}, \boldsymbol{V}]$.
12:     Remove center of mass from $\boldsymbol{C}^{(T)}$.
13:     Symmetrize $\boldsymbol{B}^{(T)}$.

14: **Denoising loop:**
15: $t = T$
16: **while** $t > 0$ **do**
17:     $\boldsymbol{x_1}^{(t)} = (\boldsymbol{a}^{(t)}, \boldsymbol{f}^{(t)}, \boldsymbol{C}^{(t)}, \boldsymbol{B}^{(t)})$.
18:     $\boldsymbol{x_2}^{(t)} = (\boldsymbol{S_2}^{(t)})$.
19:     $\boldsymbol{x_3}^{(t)} = (\boldsymbol{S_3}^{(t)}, \boldsymbol{v}^{(t)})$.
20:     $\boldsymbol{x_4}^{(t)} = (\boldsymbol{p}^{(t)}, \boldsymbol{P}^{(t)}, \boldsymbol{V}^{(t)})$.

21:     **if** `harmonization_ts[0]` $== t$ **then**     ▷ Resample states $\Delta t$ steps in forward direction
22:         `harmonization_ts.pop(0)`
23:         $\Delta t = $ `harmonization_steps.pop(0)`
24:         $\boldsymbol{x_i}^{(t+\boldsymbol{\Delta t})} = $ `forward_noise`$(\boldsymbol{x_i}^{(t)}, t, \Delta t, $ `noise_schedule`$) \; \forall \, i \in [1, 2, 3, 4]$
25:         $t = t + \Delta t$
26:         **continue**
27:     **end if**
28:                                ▷ Predict true noise with denoising network $\boldsymbol{\eta_X}$
29:     $(\hat{\boldsymbol{\epsilon}}_a, \hat{\boldsymbol{\epsilon}}_f, \hat{\boldsymbol{\epsilon}}_C, \hat{\boldsymbol{\epsilon}}_B, \hat{\boldsymbol{\epsilon}}_{S_2}, \hat{\boldsymbol{\epsilon}}_{S_3}, \hat{\boldsymbol{\epsilon}}_v, \hat{\boldsymbol{\epsilon}}_p, \hat{\boldsymbol{\epsilon}}_P, \hat{\boldsymbol{\epsilon}}_V)$
    $= $ `ShEPhERD_forward`$(\boldsymbol{x_1}^{(t)}, \boldsymbol{x_2}^{(t)}, \boldsymbol{x_3}^{(t)}, \boldsymbol{x_4}^{(t)}, t)$
30:
31:     **for** $\boldsymbol{x} \in [\boldsymbol{a}, \boldsymbol{f}, \boldsymbol{C}, \boldsymbol{B}, \boldsymbol{S_2}, \boldsymbol{S_3}, \boldsymbol{v}, \boldsymbol{p}, \boldsymbol{P}, \boldsymbol{V}]$ **do**     ▷ Apply reverse denoising equation
32:         $\boldsymbol{\epsilon}'_{\boldsymbol{x}} \sim \mathcal{N}(0, 1)$
33:         **if** $x == \boldsymbol{C}$ **then**
34:             $\boldsymbol{\epsilon}'_{\boldsymbol{x}} = \boldsymbol{\epsilon}'_{\boldsymbol{x}} - $ `get_center_of_mass`$(\boldsymbol{\epsilon}'_{\boldsymbol{x}})$
35:         **end if**
36:         **if** $x \in [\boldsymbol{C}, \boldsymbol{S_2}, \boldsymbol{S_3}, \boldsymbol{P}]$ and `extra_noise_ts[0]` $== t$ **then**
37:             `extra_noise_ts.pop(0)`
38:             $s_{\boldsymbol{\epsilon}} = $ `extra_noise_scales.pop(0)`
39:         **else**
40:             $s_{\boldsymbol{\epsilon}} = 0$
41:         **end if**
42:         $\boldsymbol{x}^{(t-1)} = \frac{1}{\alpha_t} \boldsymbol{x}^{(t)} - \frac{\sigma_t^2}{\alpha_t \overline{\sigma}_t} \hat{\boldsymbol{\epsilon}}_{\boldsymbol{x}}^{(t)} + \left( \frac{\sigma_t \overline{\sigma}_{t-1}}{\overline{\sigma}_t} + s_{\boldsymbol{\epsilon}} \right) \boldsymbol{\epsilon}'_{\boldsymbol{x}}$
43:         $\boldsymbol{x}^{(t)} = \boldsymbol{x}^{(t-1)}$
44:     **end for**
45:     $t = t - 1$
46: **end while**

47: **Final feature processing:**
48:     $(\boldsymbol{a}^{(0)}, \boldsymbol{f}^{(0)}, \boldsymbol{B}^{(0)}, \boldsymbol{v}^{(0)}, \boldsymbol{p}^{(0)}, \boldsymbol{V}^{(0)}) = $ `undo_scaling`$(\boldsymbol{a}^{(0)}, \boldsymbol{f}^{(0)}, \boldsymbol{B}^{(0)}, \boldsymbol{v}^{(0)}, \boldsymbol{p}^{(0)}, \boldsymbol{V}^{(0)})$
49:     $(\boldsymbol{a}^{(0)}, \boldsymbol{f}^{(0)}, \boldsymbol{B}^{(0)}, \boldsymbol{p}^{(0)}, \boldsymbol{V}^{(0)}) = $ `argmax_and_round`$(\boldsymbol{a}^{(0)}, \boldsymbol{f}^{(0)}, \boldsymbol{B}^{(0)}, \boldsymbol{p}^{(0)}, \boldsymbol{V}^{(0)})$

---

---

**Algorithm 5** Sampling Algorithm for Conditional Generation with Inpainting

---

1: **Given:**
2:     Trained ShEPhERD model $\boldsymbol{\eta_X}$ that learns $P(\boldsymbol{X})$ for $\boldsymbol{X} \subset \{\boldsymbol{x_1}, \boldsymbol{x_2}, \boldsymbol{x_3}, \boldsymbol{x_4}\}$
3:     $\boldsymbol{n_1}$                                                   $\triangleright$ number of atoms to generate.
4:     (if $\boldsymbol{x_4} \in \boldsymbol{X}$) $\boldsymbol{n_4}$                                  $\triangleright$ number of pharmacophores to generate
5:     Noise schedule $\boldsymbol{\sigma_t}, \boldsymbol{\alpha_t}, \overline{\boldsymbol{\sigma}}_t, \overline{\boldsymbol{\alpha}}_t$ for $t \in [1, T]$
6:     A subset of the target interaction profiles $\mathcal{I}^* \subseteq \{\boldsymbol{S_2^*}, \boldsymbol{S_3^*}, \boldsymbol{v^*}, \boldsymbol{p^*}, \boldsymbol{P^*}, \boldsymbol{V^*}\}$

7: **Simulate and store forward-noising of each target interaction profile:**
8: **for** $\boldsymbol{x}^* \in \mathcal{I}^*$ **do**
9:     $\boldsymbol{x}_{\text{noised}}^* = \{\}$
10:     $\boldsymbol{x}^{*(t=0)} = \boldsymbol{x}^*$
11:     $\boldsymbol{x}_{\text{noised}}^*[0] = \boldsymbol{x}^{*(t=0)}$
12:     **for** $t \in \texttt{range}(1, T)$ **do**:
13:         Sample $\boldsymbol{\epsilon} \sim N(\boldsymbol{0}, \boldsymbol{1})$
14:         $\boldsymbol{x}_{\text{noised}}^*[t] = \alpha_t \boldsymbol{x}_{\text{noised}}^*[t-1] + \sigma_t \boldsymbol{\epsilon}$
15:     **end for**
16: **end for**

17: **Sample initial states at $t = T$ from Gaussian priors:**
18:     Sample noises $\boldsymbol{x}^{(T)} \sim \mathcal{N}(\boldsymbol{0}, \boldsymbol{1})$ for $\boldsymbol{x} = [\boldsymbol{a}, \boldsymbol{f}, \boldsymbol{C}, \boldsymbol{B}, \boldsymbol{S_2}, \boldsymbol{S_3}, \boldsymbol{v}, \boldsymbol{p}, \boldsymbol{P}, \boldsymbol{V}]$
19:     Remove center of mass from $\boldsymbol{C}^{(T)}$
20:     Symmetrize $\boldsymbol{B}^{(T)}$

21: **Denoising loop:**
22: $t = T$
23: **while** $t > 0$ **do**
24:                     $\triangleright$ Replace denoised profiles with their corresponding forward-noised target profiles
25:     **for** $\boldsymbol{x}^{(t)} \in [\boldsymbol{S_2^{(t)}}, \boldsymbol{S_3^{(t)}}, \boldsymbol{v^{(t)}}, \boldsymbol{p^{(t)}}, \boldsymbol{P^{(t)}}, \boldsymbol{V^{(t)}}]$ **do**
26:         $\boldsymbol{x}^{(t)} = \boldsymbol{x}_{\text{noised}}^*[t]$                           $\triangleright$ if $\boldsymbol{x}_{\text{noised}}^* \in \mathcal{I}^*$
27:     **end for**

28:     $\boldsymbol{x_1^{(t)}} = (\boldsymbol{a^{(t)}}, \boldsymbol{f^{(t)}}, \boldsymbol{C^{(t)}}, \boldsymbol{B^{(t)}})$
29:     $\boldsymbol{x_2^{(t)}} = (\boldsymbol{S_2^{(t)}})$
30:     $\boldsymbol{x_3^{(t)}} = (\boldsymbol{S_3^{(t)}}, \boldsymbol{v^{(t)}})$
31:     $\boldsymbol{x_4^{(t)}} = (\boldsymbol{p^{(t)}}, \boldsymbol{P^{(t)}}, \boldsymbol{V^{(t)}})$
32:                             $\triangleright$ Predict true noise with denoising network $\boldsymbol{\eta_X}$
33:     $(\hat{\boldsymbol{\epsilon}}_a, \hat{\boldsymbol{\epsilon}}_f, \hat{\boldsymbol{\epsilon}}_C, \hat{\boldsymbol{\epsilon}}_B, \hat{\boldsymbol{\epsilon}}_{S_2}, \hat{\boldsymbol{\epsilon}}_{S_3}, \hat{\boldsymbol{\epsilon}}_v, \hat{\boldsymbol{\epsilon}}_p, \hat{\boldsymbol{\epsilon}}_P, \hat{\boldsymbol{\epsilon}}_V)$
        $= \texttt{ShEPhERD\_forward}(\boldsymbol{x_1^{(t)}}, \boldsymbol{x_2^{(t)}}, \boldsymbol{x_3^{(t)}}, \boldsymbol{x_4^{(t)}}, t)$
34:                               $\triangleright$ Apply reverse denoising equation
35:     **for** $x \in [\boldsymbol{a}, \boldsymbol{f}, \boldsymbol{C}, \boldsymbol{B}, \boldsymbol{S_2}, \boldsymbol{S_3}, \boldsymbol{v}, \boldsymbol{p}, \boldsymbol{P}, \boldsymbol{V}]$ **do**
36:         $\boldsymbol{\epsilon}_x' \sim \mathcal{N}(0, 1)$
37:         **if** $x == \boldsymbol{C}$ **then**
38:             $\boldsymbol{\epsilon}_x' = \boldsymbol{\epsilon}_x' - \texttt{get\_center\_of\_mass}(\boldsymbol{\epsilon}_x')$
39:         **end if**
40:         $x^{(t-1)} = \frac{1}{\alpha_t} x^{(t)} - \frac{\sigma_t^2}{\alpha_t \overline{\sigma}_t} \hat{\boldsymbol{\epsilon}}_x^{(t)} + \frac{\sigma_t \overline{\sigma}_{t-1}}{\overline{\sigma}_t} \boldsymbol{\epsilon}_x'$
41:         $x^{(t)} = x^{(t-1)}$
42:     **end for**
43:     $t = t - 1$
44: **end while**

45: **Final feature processing:**
46:     $(\boldsymbol{a}^{(0)}, \boldsymbol{f}^{(0)}, \boldsymbol{B}^{(0)}, \boldsymbol{v}^{(0)}, \boldsymbol{p}^{(0)}, \boldsymbol{V}^{(0)}) = \texttt{undo\_scaling}(\boldsymbol{a}^{(0)}, \boldsymbol{f}^{(0)}, \boldsymbol{B}^{(0)}, \boldsymbol{v}^{(0)}, \boldsymbol{p}^{(0)}, \boldsymbol{V}^{(0)})$
47:     $(\boldsymbol{a}^{(0)}, \boldsymbol{f}^{(0)}, \boldsymbol{B}^{(0)}, \boldsymbol{p}^{(0)}, \boldsymbol{V}^{(0)}) = \texttt{argmax\_and\_round}(\boldsymbol{a}^{(0)}, \boldsymbol{f}^{(0)}, \boldsymbol{B}^{(0)}, \boldsymbol{p}^{(0)}, \boldsymbol{V}^{(0)})$

---

### A.7.3 MODEL HYPERPARAMETERS

Table 7 lists hyperparameters relevant to training *ShEPhERD*. Table 8 lists the total number of learnable parameters for each version of *ShEPhERD* analyzed in the main text. Table 9 lists hyperparameters related to *ShEPhERD*'s denoising network architecture. Note that model hyperparameters were selected early during model development based on the memory limits of our GPUs, and were not specially tuned to optimize model performance. A notable exception is the probe radius for extracting $x_2$ and $x_3$, which was manually tuned to $0.6$ Å in order to decouple the shape/surface representation from the 2D molecular graph or exact atomic coordinates while still yielding well-defined shapes. Note that a probe radius of 0.0 corresponds to the van der Waals surface, whereas a very large probe radius would cause all molecules to have uninformative spherical shapes. If one requires the shape/surface representation to be more sensitive to the exact atomic coordinates, one can easily retrain *ShEPhERD* with a smaller probe radius.

Table 7: Hyperparameters used for training *ShEPhERD*

| Training Parameters | |
|---|---|
| **Parameter** | **Value** |
| Effective batch size | 48 |
| Learning rate | 0.0003 |
| Gradient clipping value | 5.0 |
| $T$ | 400 |
| $n_2$ | 75 |
| $n_3$ | 75 |
| surface probe radius | 0.6 |
| $a$ and $f$ scaling factors | 0.25 |
| $B$ scaling factor | 1.0 |
| $v$ scaling factor | 2.0 |
| $p$ scaling factor | 2.0 |
| $V$ scaling factor | 2.0 |

Table 8: Total number of learnable parameters for each *ShEPhERD* model

| Total Number of Learnable Parameters | |
|---|---|
| **Model** | **Number of Parameters** |
| ShEPhERD-GDB17 $P(x_1, x_2)$ | 4,375,054 |
| ShEPhERD-GDB17 $P(x_1, x_3)$ | 4,387,407 |
| ShEPhERD-GDB17 $P(x_1, x_4)$ | 4,407,321 |
| ShEPhERD-MOSES-aq $P(x_1, x_3, x_4)$ | 6,010,427 |

Table 9: Hyperparameters for *ShEPhERD*'s denoising network $\eta$

| Denoising Network Hyperparameters | |
|---|---|
| **Parameter** | **Value** |
| ***Default EquiformerV2 Parameters*** | |
| num_node_channels | 64 |
| lmax_list | [1] |
| mmax_list | [1] |
| ffn_hidden_channels | 32 |
| grid_resolution | 16 |
| num_sphere_samples | 128 |
| edge_channels | 128 |
| activation_function | silu |
| norm_type | layer_norm_sh |
| use_sep_s2_act | True |
| use_grid_mlp | True |
| use_gate_act | False |
| use_attn_renorm | True |
| use_s2_act_attn | False |
| ***Joint Module Parameters*** | |
| num_EquiformerV2_layers | 2 |
| attention_channels | 24 |
| num_attention_heads | 2 |
| radius_graph_cutoff | 5.0 |
| RBF_cutoff | 5.0 |
| ***$x_1$ Embedding Module Parameters*** | |
| num_EquiformerV2_layers | 4 |
| attention_channels | 32 |
| num_attention_heads | 4 |
| ffn_hidden_channels | 64 |
| radius_graph_cutoff | $\infty$ (fully connected) |
| RBF_cutoff | 5.0 |
| ***$x_2$, $x_3$, $x_4$ Embedding Module Parameters*** | |
| num_EquiformerV2_layers | 2 |
| attention_channels | 24 |
| num_attention_heads | 2 |
| ffn_hidden_channels | 32 |
| radius_graph_cutoff | 5.0 |
| RBF_cutoff | 5.0 |
| x3_scalar_RBF_expansion_min | -10.0 |
| x3_scalar_RBF_expansion_max | 10.0 |
| ***$x_1$ Denoising Module Parameters*** | |
| MLP_hidden_dim | 64 |
| num_MLP_hidden_layers | 2 |
| e3nn_ffn_hidden_channels | 32 |
| egnn_normalize_vectors | True |
| egnn_distance_expansion_dim | 32 |
| ***$x_2$, $x_3$, $x_4$ Denoising Module Parameters*** | |
| MLP_hidden_dim | 64 |
| num_MLP_hidden_layers | 2 |
| e3nn_ffn_hidden_channels | 32 |

A.8  Symmetry breaking in unconditional generation

We find that during unaltered unconditional generation, *ShEPhERD* tends to generate spherical molecules regardless of the choice of $n_1$ (Fig. 21). We attribute this behavior to *ShEPhERD* being unable to reliably break spherical symmetry when denoising the Euclidean coordinate components of $X^{(t)}$. This phenomenon is particularly evident for models trained on *ShEPhERD*-GDB17, which contains a number of spherical cage-like molecules in the true data distribution. But, this also occurs for models trained on *ShEPhERD*-MOSES-aq, which does not contain such molecules. Intuitively, we can understand this phenomenon by considering that we denoise molecular structures and their interaction profiles starting from isotropic Gaussian noise. At early time steps in the denoising trajectory (e.g., $t$ close to $T$), the model learns to collapse the isotropic noise into a near point-mass; this behavior minimizes the L2 denoising losses at early time steps as predicting clean structures from uninformative noisy input states is impossible. Due to *ShEPhERD*'s strict SE(3)-equivariance, though, the denoising network cannot natively break the spherical symmetry of this point mass; this ultimately leads the network to generate spherically-shaped (yet often still valid) molecules.

Fortunately, we can easily solve this problem by manually adding extra noise to the coordinate components of $X^{(t-1)}$ during the most informative time steps. With our choice of noise schedule, the states $X^{(t)}$ undergo the most significant evolution in terms of their spatial coordinates from $t = 130$ through $t = 80$. For $t \in [80, 130]$, then, we add extra (isotropic) noise to $C^{(t-1)}, S_2^{(t-1)}, S_3^{(t-1)}, P^{(t-1)}$ so that (as an example): $C^{(t-1)} = \frac{1}{\alpha_t} C^{(t)} - \frac{\sigma_t^2}{\alpha_t \overline{\sigma}_t} \hat{\epsilon}^{(t)} + \frac{\sigma_t \overline{\sigma}_{t-1}}{\overline{\sigma}_t} \epsilon' + \xi \epsilon_{\text{symmetry-breaking}}$ where $\epsilon_{\text{symmetry-breaking}} \sim N(\mathbf{0}, \mathbf{1})$ and $\xi$ scales the noise (although we just use $\xi = 1$). For $C^{(t-1)}$, we still remove the COM from $\epsilon_{\text{symmetry-breaking}}$.

Naively adding extra noise to the coordinates of the denoised states could quickly cause the model to go out-of-distribution, leading to low-quality or invalid samples. To remedy this, we use harmonization/resampling (Lugmayr et al., 2022) to re-sample the state $X^{(t=80+\Delta t)}$ given the symmetry-broken $X^{(t=80)}$ (we use $\Delta t = 20$). Empirically, we find this adequate to maintain sample quality, as evidenced by the high validity rates and low RMSDs upon xTB-relaxation for unconditional samples from *ShEPhERD* when trained on *ShEPhERD*-GDB17. Nevertheless, we admit there may be less intrusive methods of breaking symmetry; e.g., adding *non*-isotropic noise with smaller $\xi$. However, we find our symmetry-breaking procedure to be simple and effective while only marginally increasing sampling time.

Finally, we emphasize that this symmetry breaking procedure is only performed during *unconditional generation*. When inpainting during conditional generation, symmetry is naturally broken by the conditioning information, so we do not add any extra noise (although doing so could increase the diversity of conditionally-sampled structures).

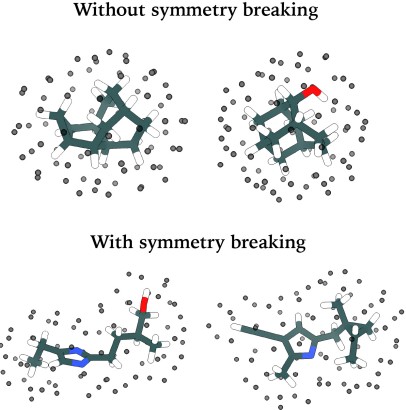

Figure 21: Without symmetry breaking during unconditional generation, *ShEPhERD* tends to generate spherical molecules, particularly for models trained on *ShEPhERD*-GDB17. Shown are unconditional samples from *ShEPhERD* trained to learn $P(x_1, x_2)$ on *ShEPhERD*-GDB17, generated either with or without symmetry breaking.

A.9 CHARACTERIZING THE NEED FOR OUT-OF-DISTRIBUTION PERFORMANCE

Multiple of our experiments, particularly those related to ligand-based drug design, require *ShEPhERD* to generate larger molecules with more pharmacophores than the molecules that *ShEPhERD* was trained on. For instance, the largest molecules in *ShEPhERD*-MOSES-aq contain 27 heavy atoms, whereas the best-scoring ligands generated in our natural product ligand hopping experiments contain more than that, presumably because these larger molecules fit the shapes of the very large natural products better than molecules with fewer heavy atoms. Moreover, in our fragment merging experiment, we conditioned *ShEPhERD* on a pharmacophore profile containing 27 pharmacophores, which is many more than those contained by the molecules in *ShEPhERD*-MOSES-aq. To quantitatively illustrate the need for out-of-distribution performance and *ShEPhERD*'s extrapolation ability, Fig. 22 shows the distributions of $n_1$ vs. $n_4$ for molecules from *ShEPhERD*-MOSES-aq compared against the molecules conditionally generated by *ShEPhERD* across our bioisosteric drug design experiments. Also included are the top-scoring ligands for each experiment, to highlight that *ShEPhERD* finds high-scoring compounds that are both in-distribution *and* meaningfully out-of-distribution. Fig. 23 shows similar distributions for our unconditional and conditional generation experiments on the *ShPhERD*-GDB17 dataset.

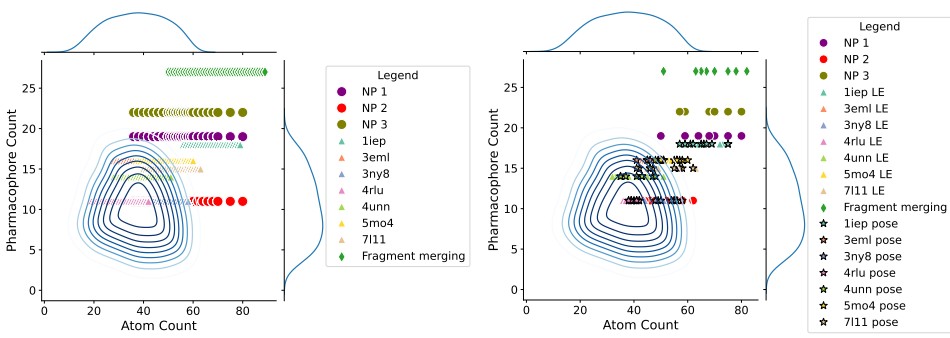

Figure 22: Distributions of the number of pharmacophores relative to the number of atoms (including hydrogens) for conditional samples from *ShEPhERD* when trained on *ShEPhERD*-MOSES-aq. For comparison, the kernel density plot of a randomly sampled subset of ∼320k molecules from the *ShEPhERD-MOSES-aq* dataset is shown in blue on both figures. (**Left**) Conditional *ShEPhERD*-generated samples from $P(\boldsymbol{x}_1|\boldsymbol{x}_3, \boldsymbol{x}_4)$ for our natural product (NP) ligand hopping, bioactive hit diversification, and bioisosteric fragment merging experiments. Note that the ordering of the NPs correspond to the order found in Fig. 4. Because we sweep over $n_1$ while specifying $n_4$ based on the target $\boldsymbol{x}_4$, the *ShEPhERD*-generated samples appear as horizontal lines. (**Right**) "Top-10" scoring samples generated by *ShEPhERD* for each experiment. The "Top-10" samples for the NP and fragment merging experiments are defined as the valid samples that have the highest combined ESP & pharmacophore score post xTB-relaxation and ESP-based realignment. The "Top-10" samples for the bioactive hit diversification experiments are defined by their Vina scores. We include the top samples when conditioning on both the lowest energy conformer (LE) and the bound pose of each PDB ligand. Top-scoring samples are found both in- and out-of-distribution in terms of $(n_1, n_4)$.

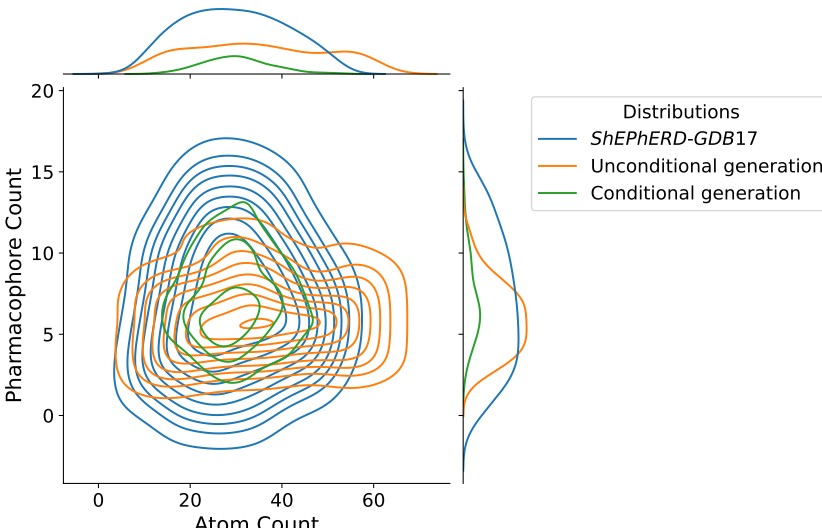

Figure 23: The distributions of the number of pharmacophores relative to the number of atoms (including hydrogens) for molecules in the *ShEPhERD*-GDB17 dataset and molecules sampled by *ShEPhERD* trained to learn $P(\boldsymbol{x}_1, \boldsymbol{x}_4)$. We include the distributions for $\sim$1M randomly sampled molecules from the *ShEPhERD*-GDB17 dataset, unconditionally-generated molecules from $P(\boldsymbol{x}_1, \boldsymbol{x}_4)$, and conditionally-generated molecules from $P(\boldsymbol{x}_1|\boldsymbol{x}_4)$. All distributions are visualized as kernel density estimate plots. All distributions show significant overlap.

### A.10 ADDITIONAL EXAMPLES OF GENERATED MOLECULES

Figures 24, 25, and 26 show additional random examples of unconditionally-generated molecules and their jointly generated interaction profiles from versions of *ShEPhERD* trained on *ShEPhERD*-GDB17. Figures 27, 28, and 29 show additional random examples of *ShEPhERD*-generated molecules when conditioning on target interaction profiles via inpainting. Although we are primarily interested in using the models trained to learn $P(\boldsymbol{x}_1, \boldsymbol{x}_3, \boldsymbol{x}_4)$ on *ShEPhERD*-MOSES-aq for conditional generation tasks relevant to ligand-based drug design, we also show unconditional samples from this model in Fig. 30.

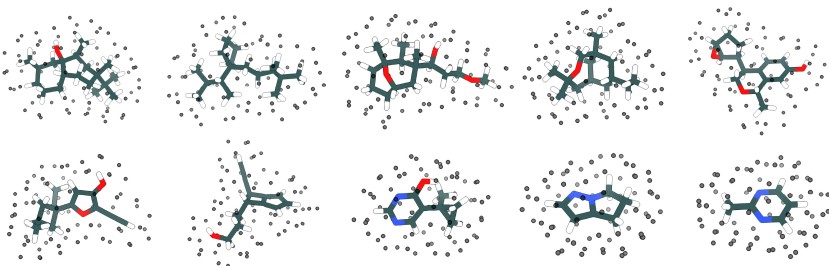

Figure 24: Random unconditional samples from *ShEPhERD* trained to learn $P(\boldsymbol{x}_1, \boldsymbol{x}_2)$ on *ShEPhERD*-GDB17. We show the natively generated molecular structures (without xTB-relaxation) and their jointly generated shapes. We show examples for both large and small $n_1$.

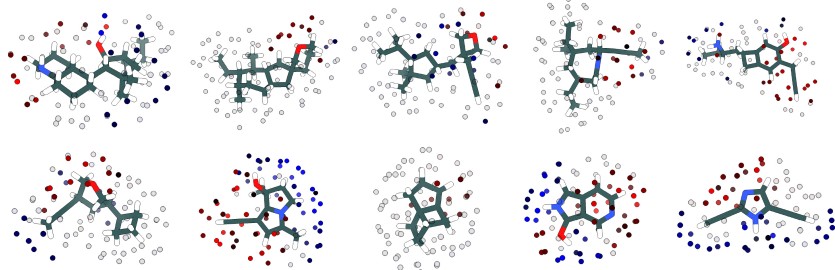

Figure 25: Random unconditional samples from *ShEPhERD* trained to learn $P(\boldsymbol{x}_1, \boldsymbol{x}_3)$ on *ShEPhERD*-GDB17. We show the natively generated molecular structures (without xTB-relaxation) and their jointly generated ESP surfaces. We show examples for both large and small $n_1$.

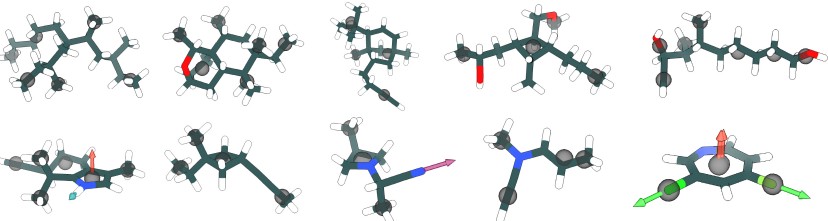

Figure 26: Random unconditional samples from *ShEPhERD* trained to learn $P(\boldsymbol{x}_1, \boldsymbol{x}_4)$ on *ShEPhERD*-GDB17. We show the natively generated molecular structures (without xTB-relaxation) and their jointly generated pharmacophores. We show examples for both large and small $n_1$. $n_4$ is sampled from the empirical data distribution $P(n_4|n_1)$. In these images, grey spheres represent hydrophobes (the most common pharmacophore in *ShEPhERD*-GDB17); purple and blue arrows represent HBAs and HBDs, respectively; orange arrows represent aromatic groups; and green arrows represent halogen-carbon bonds.

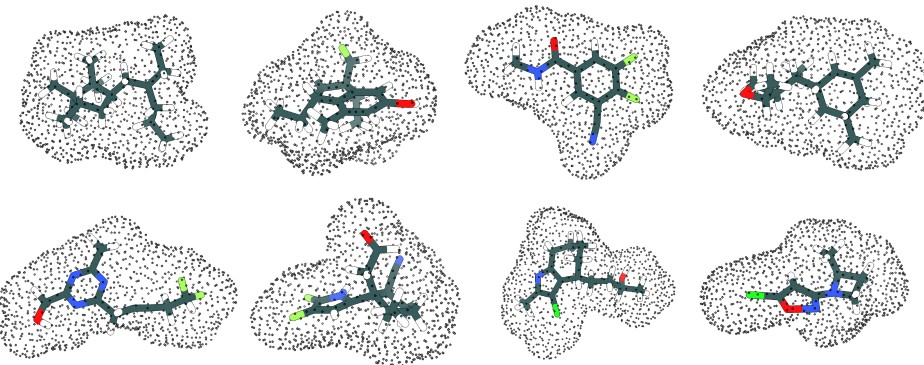

Figure 27: Conditional samples from *ShEPhERD* – trained to learn $P(\boldsymbol{x}_1, \boldsymbol{x}_2)$ on *ShEPhERD*-GDB17 – obtained by inpainting the 3D molecule given target shapes extracted from molecules held-out from training. We show the natively generated molecular structures (without xTB-relaxation or realignment) overlaid on the target shapes. The shape surfaces are upsampled for visualization.

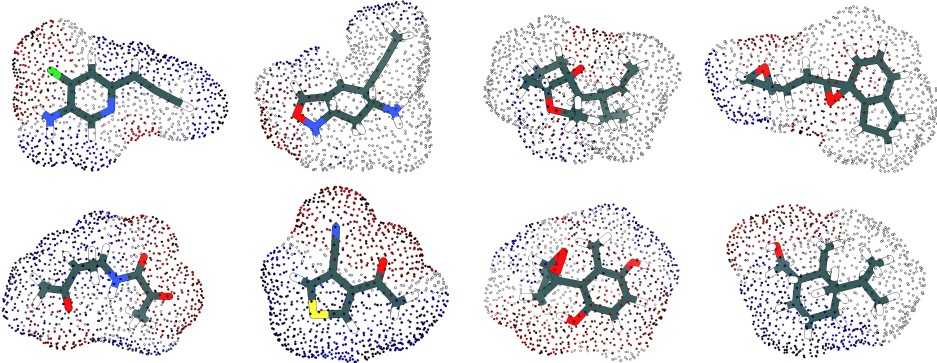

Figure 28: Conditional samples from *ShEPhERD* – trained to learn $P(\boldsymbol{x}_1, \boldsymbol{x}_3)$ on *ShEPhERD*-GDB17 – obtained by inpainting the 3D molecule given target ESP surfaces extracted from molecules held-out from training. We show the natively generated molecular structures (without xTB-relaxation or realignment) overlaid on the target ESP surfaces. The surfaces are upsampled for visualization.

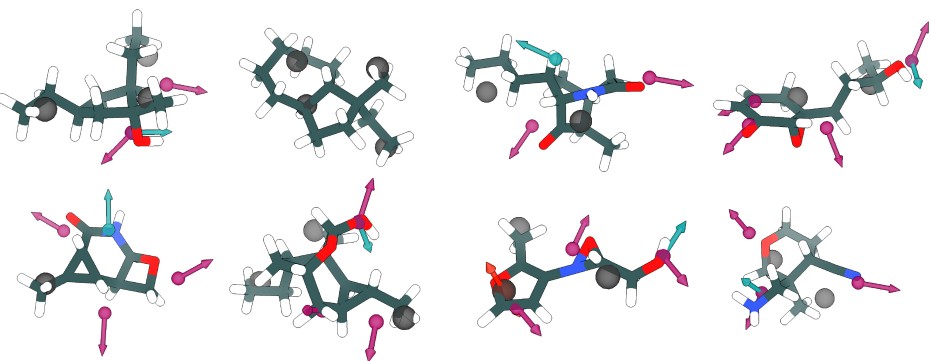

Figure 29: Conditional samples from *ShEPhERD* – trained to learn $P(\boldsymbol{x}_1, \boldsymbol{x}_4)$ on *ShEPhERD*-GDB17 – obtained by inpainting the 3D molecule given target pharmacophores extracted from molecules held-out from training. We show the natively generated molecular structures (without xTB-relaxation or realignment) overlaid on the target pharmacophores.

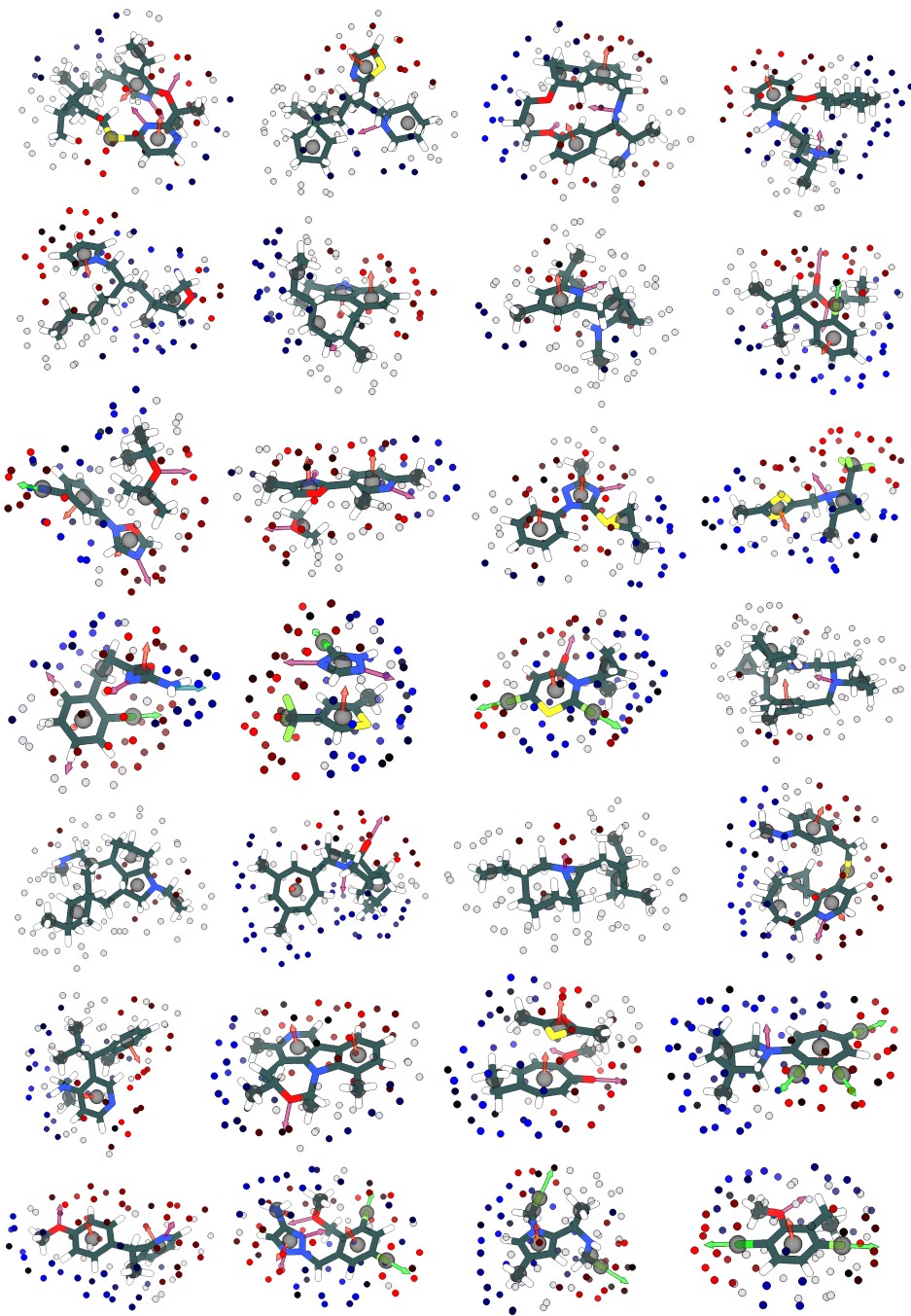

Figure 30: Random unconditional samples from *ShEPhERD* trained to learn $P(\boldsymbol{x}_1, \boldsymbol{x}_3, \boldsymbol{x}_4)$ on *ShEPhERD*-MOSES-aq. We show the natively generated molecular structures (without xTB-relaxation or realignment) and their jointly generated ESP surfaces and pharmacophores. Note that a number of these structures would not pass our validity criteria.

### A.11 Training and Inference Resources

**Training**. We train all models with V100 GPUs (32 GB memory). For both the *ShEPhERD*-GDB17 and *ShEPhERD*-MOSES-aq datasets, we train each of the $P(x_1, x_2)$, $P(x_1, x_3)$, and $P(x_1, x_3, x_4)$ models on 2 GPUs for approximately 2 weeks. We train the $P(x_1, x_4)$ model on 1 GPU for approximately 2 weeks. We configure the number of GPUs, nominal batch sizes, and the number of gradient accumulation steps to attain effective batch sizes of approximately 48 (molecules).

**Inference**. For each of the $P(x_1, x_2)$, $P(x_1, x_3)$, and $P(x_1, x_3, x_4)$ models, generating a batch of 10 independent samples (either unconditionally or via inpainting) takes approximately 3-4 minutes on a V100 GPU, when using iterative denoising with $T = 400$ steps. Sampling times are reduced by $\sim 50\%$ for the $P(x_1, x_4)$ models. Using harmonization when sampling increases the sampling time proportionally to the number of extra denoising steps. On a single CPU, inference requires 5-10 minutes per sample depending on system hardware and the choice of model.

Long sampling times are somewhat intrinsic to the DDPM paradigm due to needing numerous forward passes per sample. Our sampling is also slowed down by our use of memory-intensive E3NN modules throughout *ShEPhERD*'s denoising network. In this work, we did not attempt to accelerate sampling (e.g., by using fewer denoising steps at inference, reducing $T$, using less expressive networks, reducing $n_2$ and $n_3$, etc.). We leave such performance engineering to future work.

## A.12 ADDITIONAL EXPERIMENTS AND COMPARISONS TO RELATED WORK

### A.12.1 COMPARISON TO SQUID FOR SHAPE-CONDITIONED GENERATION

We compare *ShEPhERD* to two ligand-based models, SQUID (Adams & Coley, 2022) and Shape-Mol (Chen et al., 2023), for shape-conditioned molecular design. To perform this comparison, we apply the version of *ShEPhERD* trained on MOSES-aq to learn $P(x_1, x_3, x_4)$ to generate new molecules conditioned on the surface shape (e.g., the coordinates of $x_3$, which implicitly define $x_2$) of the 1000 3D molecules in SQUID's test set. Note that SQUID and *ShEPhERD* use the same training and test splits of MOSES, and hence there was no data leakage. We condition the $P(x_1, x_3, x_4)$ model on the molecules' shapes (only) by inpainting just the coordinates of $x_3$, and allowing *ShEP-hERD* to generate $x_1$, $x_4$, and the electrostatic potentials of $x_3$. We generate up to 10 samples from *ShEPhERD* and compare the average volumetric shape similarity using ROCS (the same shape scoring function that SQUID used). Since SQUID's generated molecules for each test molecule are publicly available for download via their GitHub, we re-evaluated the average shape similarity of SQUID-generated molecules to ensure a fair comparison.

Across the 1000 test-set molecules, the average shape similarity of SQUID-generated molecules is 0.70 when using $\lambda = 1.0$, and 0.74 when using $\lambda = 0.3$. In contrast, the average shape similarity of *ShEPhERD*-generated molecules is 0.80, a substantial improvement. Moreover, the *ShEPhERD*-generated molecules have an average 2D graph similarity of just 0.22 compared to the reference molecule, whereas the SQUID-generated molecules have average graph similarities of 0.25 ($\lambda = 1.0$) and 0.35 ($\lambda = 0.3$). Hence, *ShEPhERD* is able to generate less chemically similar molecules that have higher shape similarity to the target, on average. The full results are found in Table A.12.1.

We emphasize that *ShEPhERD* greatly exceeds the performance of SQUID even with this indirect way of doing shape-conditioned generation, i.e. via inpainting the joint $P(x_1, x_3, x_4)$ model. We could also train a $P(x_1, x_2)$ model directly for this task; however, we did not have time to retrain *ShEPhERD* from scratch within the rebuttal time period. We also note that *ShEPhERD* conditions on surface shapes, whereas we have compared *ShEPhERD* to SQUID using volumetric shape similarity (with ROCS). We expect that we could further improve *ShEPhERD*'s performance in shape-conditioned generation by retraining *ShEPhERD* with a smaller surface probe radius, thereby "shrinking" the surface shape to better model volumetric shapes.

Table 10: A comparison of *ShEPhERD* to default versions of SQUID ($\lambda = \{1.0, 0.3\}$) and Shape-Mol. We recompute the shape and graph similarities of SQUID using the structures provided by their GitHub. Note that the values for the evaluations of ShapeMol and ShapeMol+g were directly copied from (Chen et al., 2023) which use a different alignment protocol.

| Model | avgSim$_s$ ($\uparrow$) | avgSim$_g$ ($\downarrow$) | QED ($\uparrow$) |
|---|---|---|---|
| *ShEPhERD* | **0.799** $\pm$ 0.058 | **0.223** $\pm$ 0.065 | 0.723 |
| SQUID ($\lambda = 1.0$) | 0.700 $\pm$ 0.107 | 0.245 $\pm$ 0.078 | 0.760 |
| SQUID ($\lambda = 0.3$) | 0.735 $\pm$ 0.117 | 0.346 $\pm$ 0.161 | **0.766** |
| ShapeMol* | 0.689 $\pm$ 0.044 | 0.239 $\pm$ 0.042 | 0.748 |
| ShapeMol+g* | 0.746 $\pm$ 0.036 | 0.241 $\pm$ 0.050 | 0.749 |

### A.12.2 COMPARISONS AGAINST STRUCTURE-BASED DRUG DESIGN MODELS THAT EXPLICITLY ENCODE THE PROTEIN POCKET

While *ShEPhERD* is a ligand-only model and does not explicitly model protein pocket geometries, we were asked to compare to structure-based drug design (SBDD) models. We emphasize that since the methods differ in their respective applications and modeling strategies, the comparisons cannot be completely fair. Nevertheless, although *ShEPhERD* is not designed for structure-based drug design, it can still be roughly applied to SBDD by conditioning on the bound pose of a known ligand (but still ignoring the surrounding protein pocket). This bound pose can be obtained from crystallography (e.g., the co-crystal ligands in PDB entries) or from simulated docked poses (e.g., conditioning *ShEPhERD* on the top hit from a small docking-based virtual screen). In order to compare to SBDD generative models, we compare *ShEPhERD* against the benchmarking results presented by (Zheng et al., 2024), which report the average Vina docking scores amongst the top-

10 generated compounds (amongst ~1000 generated samples) for 7 PDB systems using generative SBDD methods like 3DSBDD, AutoGrow4, Pocket2Mol, PocketFlow, and ResGen. Specifically, we use *ShEPhERD* to condition on the electrostatic potential surface ($x_3$) and pharmacophores ($x_4$) of the PDB ligand in its crystallographic bound pose, generating up to 1000 valid molecules per PDB target. We dock the 1000 generated molecules (starting from their SMILES strings) using the same Vina program (and version) as used by Zheng et al. (2024), and compute the average Vina scores amongst the top-10 molecules per target. These average top-10 scores are compared against the results for the SBDD models evaluated by Zheng et al. (2024) in Table 11. We also evaluate the 2D graph similarities of *ShEPhERD*-generated compounds compared to the PDB ligand in each case, which are generally quite low (Fig. 31).

Table 11: Docking scores of *ShEPhERD*-generaed molecules and redocked PDB ligands, compared to ligands generated by various SBDD generative models. Scores for 3DSBDD, AutoGrow, Pocket2Mol, PocketFlow, and ResGen are copied from Zheng et al. (2024).

| Model | 1iep | 3eml | 3ny8 | 4rlu | 4unn | 5mo4 | 7l11 |
|---|---|---|---|---|---|---|---|
| Redocked PDB ligand | -9.58 | -9.14 | -8.62 | -8.70 | -9.24 | -9.95 | -9.10 |
| ***ShEPhERD*** | -12.25 ± 0.18 | -10.80 ± 0.23 | -10.69 ± 0.23 | -10.19 ± 0.23 | -10.60 ± 0.27 | -10.66 ± 0.20 | -9.44 ± 0.19 |
| 3DSBDD | -9.05 ± 0.38 | -10.02 ± 0.15 | -10.10 ± 0.24 | -9.80 ± 0.55 | -8.23 ± 0.30 | -8.71 ± 0.45 | -8.47 ± 0.18 |
| AutoGrow4 | -13.23 ± 0.11 | -13.03 ± 0.09 | -11.70 ± 0.00 | -11.20 ± 0.00 | -11.14 ± 0.12 | -10.38 ± 0.27 | -8.84 ± 0.33 |
| Pocket2Mol | -10.17 ± 0.53 | -12.25 ± 0.27 | -11.89 ± 0.16 | -10.57 ± 0.12 | -12.20 ± 0.34 | -10.07 ± 0.62 | -9.74 ± 0.38 |
| PocketFlow | -12.49 ± 0.70 | -9.25 ± 0.29 | -8.56 ± 0.35 | -9.65 ± 0.25 | -7.90 ± 0.78 | -7.80 ± 0.42 | -8.35 ± 0.31 |
| ResGen | -10.97 ± 0.29 | -9.25 ± 0.95 | -10.96 ± 0.42 | -11.75 ± 0.42 | -9.41 ± 0.23 | -10.34 ± 0.39 | -8.74 ± 0.24 |

Even though *ShEPhERD* only sees the PDB ligand and not the target protein pocket, the average top-10 Vina scores for *ShEPhERD*-generated molecules are very comparable to those obtained by these SBDD methods. Out of the 6 generative models (*ShEPhERD* and 5 SBDD methods), *ShEPhERD* ranks 1/6 for one target, 2/6 for one target, 3/6 for three targets, and 4/6 for two targets. These results confirm that *ShEPhERD* shows great promise for SBDD even though it is currently a ligand-only model.

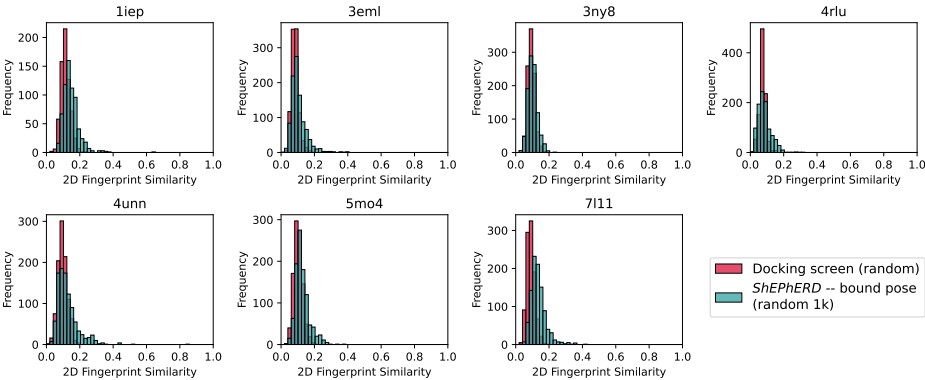

Figure 31: The graph similarity distributions of the 1000 valid *ShEPhERD*-generated molecules used in App. A.12.2 for each target. We also show the distributions of 1000 randomly selected samples from the 10k docking screen. Note that only 672 *ShEPhERD*-generated molecules were valid (and docked) for the target 1iep.

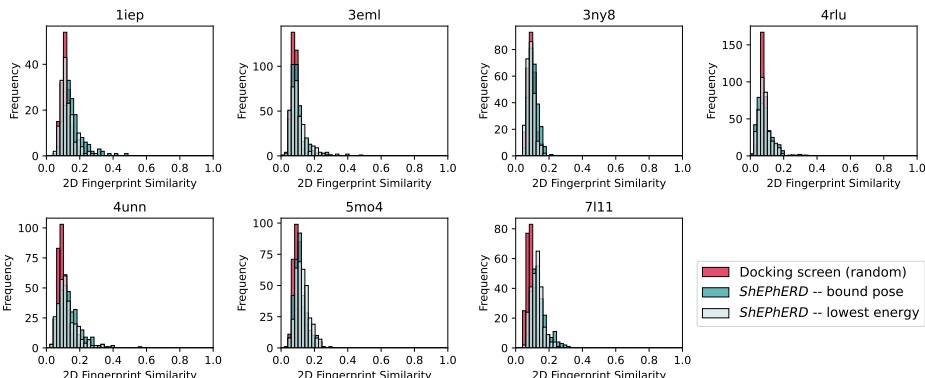

Figure 32: The graph similarity distributions of the valid *ShEPhERD*-generated molecules used in the main text. We also show the distributions of 500 randomly selected samples from the 10k docking screen.

### A.12.3  COMPARISONS AGAINST INPAINTING WITH DIFFSBDD

We also directly compare *ShEPhERD* to DiffSBDD, which also uses diffusion with inpainting to generate molecules, albeit in a structure-based drug design scenario where the pocket is directly encoded by the model. To compare *ShEPhERD* specifically against DiffSBDD, we applied *ShEPhERD* to generate analogs given the crystallographic bound pose for the ligand in the PDB entry 4tos (ligand PDB ID: 355), which DiffSBDD was evaluated on in Figure 2 of Schneuing et al. (2022). To ensure a fair comparison, we downloaded the 100 molecules generated by DiffSBDD for this PDB target, which are fortunately made publicly available by Schneuing et al. (2022). We used the molecules generated by the all-atom version of DiffSBDD that uses inpainting, as this is the model that generally performs the best according to Schneuing et al. (2022). We then re-docked these 100 molecules (starting from their SMILES strings) with AutoDock Vina (v1.1.2) to obtain their distribution of docking scores. We then applied *ShEPhERD* to generate 100 valid molecules given ($\mathbf{x}_3$, $\mathbf{x}_4$) of the bound ligand (still ignoring the actual protein pocket), and docked those molecules, again starting from their SMILES strings.

We compare the distributions of docking scores between the molecules generated by *ShEPhERD* and DiffSBDD in Figure 33. Even though *ShEPhERD* does not explicitly model the pocket structure, the average Vina score for *ShEPhERD*-generated molecules is approximately 2 kcal/mol lower (better) than the average Vina score for the DiffSBDD-generated molecules. Moreover, the *ShEPhERD*-generated molecules have significantly lower (better) SA scores, as well. Meanwhile, the *ShEPhERD*-generated molecules still have quite low graph similarity to the PDB ligand (vast majority are below 0.2). We also emphasize that this conditioning PDB ligand is clearly out-of-distribution for *ShEPhERD*, as the PDB ligand contains 42 non-hydrogen atoms, whereas *ShEPhERD* is only trained on molecules containing up to 27 non-hydrogen atoms. However, we do want to emphasize that, as previously mentioned, comparisons to SBDD methods are inherently unfair. In this case, the PDB ligand itself has a quite good docking score, which means that the *ShEPhERD*-generated analogs are likely to also have good docking scores, as *ShEPhERD* generates molecules that preserve 3D interactions. Nevertheless, this experiment shows *ShEPhERD*'s utility in SBDD, despite being a ligand-only model.

### A.12.4  COMPARISONS AGAINST SYNFORMER ON NATURAL PRODUCT LIGAND HOPPING

We compare against SynFormer (Gao et al., 2024) in our natural product ligand hopping experiments. However, note that the goal of our natural product ligand hopping experiments was to generate small-molecule analogues of structurally complex natural products that (1) preserve their electrostatic and pharmacophore 3D interactions, and (2) have simpler chemical structures as assessed via SA score. SynFormer is a non-3D generative model that can be used to generate synthesizable small-molecule analogues of an unsynthesizable reference molecule that have similar 2D chemical structures. We applied SynFormer using its publicly available code in its default settings to generate

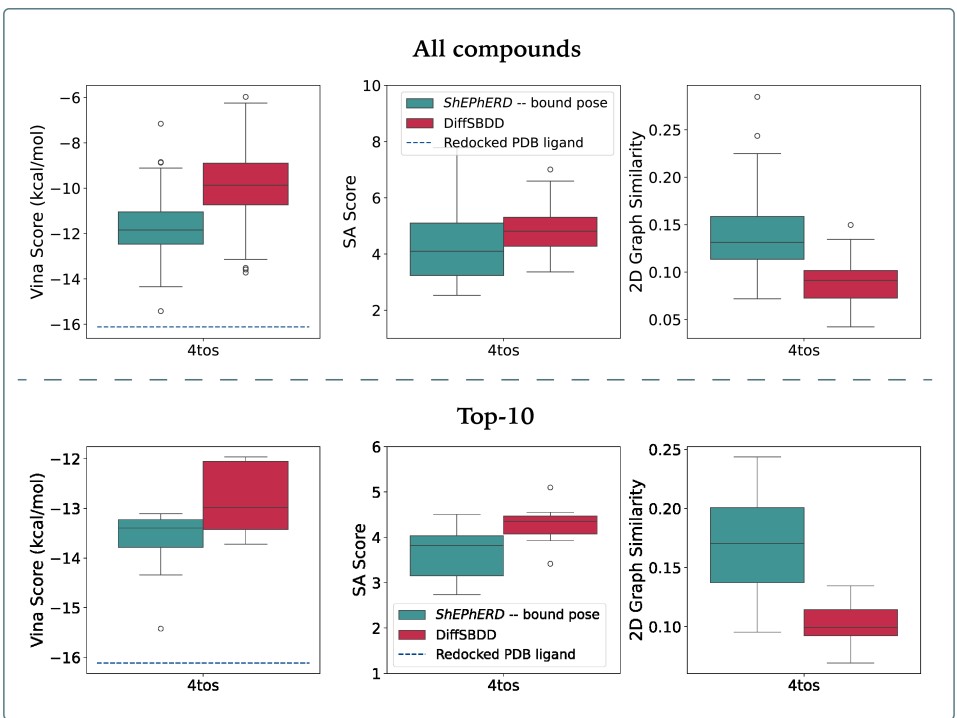

Figure 33: A comparison of distributions for molecules generated by DiffSBD and *ShEPhERD* for the target 4tos (PDB ligand ID 355). We show distributions of Vina Score (v1.1.2), SA score, and 2D graph similarity. (**Top**) The distributions of all 100 valid molecules generated by both models. (**Bottom**) The distributions of the top-10 docked molecules from both models.

analogs for each of the three natural products. We took each of the generated analogs, generated up to 10 xTB-optimized conformers, and scored their 3D similarities to the reference natural products after optimal alignment.

We compare the SynFormer-generated analogs against the *ShEPhERD*-generated analogs in Figure 34 on the basis of electrostatic and pharmacophore similarity to the reference natural products. In all cases, SynFormer underperforms *ShEPhERD*, which is not surprising since SynFormer was not designed to preserve 3D interaction similarity. We also note that very few (if any) of the SynFormer-generated molecules have SA scores below the threshold we enforce on *ShEPhERD* (SA<4.5), indicating that SynFormer's analogs are generally more complex, even if purportedly synthesizable.

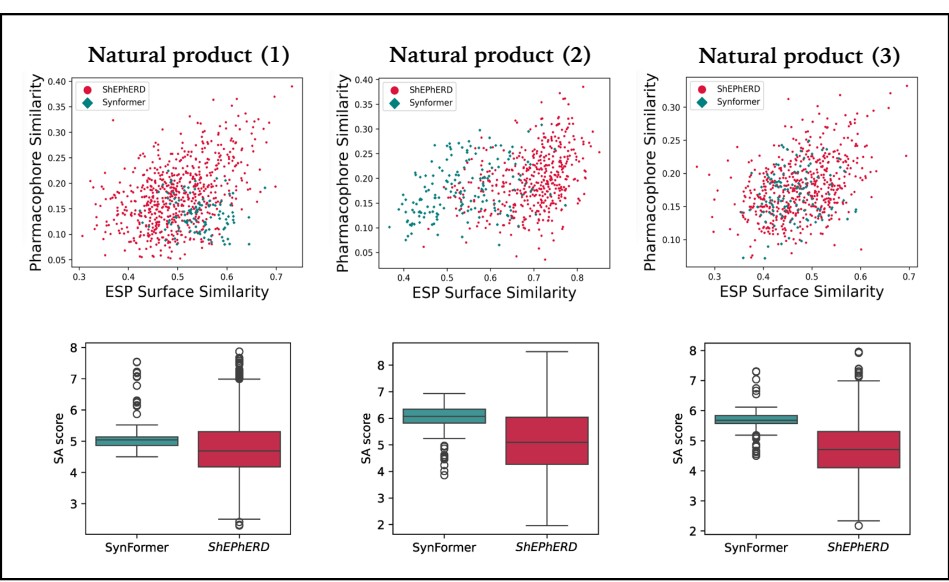

Figure 34: (**Top**) Plots of ESP *vs.* pharmacophore similarity for the molecules generated by Syn-Former and *ShEPhERD* in our natural product ligand hopping tasks. (**Bottom**) SA score distributions for each model's generated molecules.

