# OpenReview forum: "ShEPhERD: Diffusing shape, electrostatics, and pharmacophores for bioisosteric drug design"
_ICLR.cc/2025/Conference — ICLR 2025 Oral_

### Official Review · Reviewer_JQjD · 2024-11-02

**Soundness:** 3
**Presentation:** 3
**Contribution:** 3
**Rating:** 6
**Confidence:** 4

**Summary:**

The paper introduces the ShEPhERD method for generating 3D molecules that satisfy specific shape, electrostatic surface, and pharmacophore constraints. The method represents molecular shapes, electrostatic surfaces, and pharmacophores as point clouds and jointly models these properties with the 3D molecular structure. During sampling, inpainting techniques are used to generate molecules that meet the specified properties. This method enriches ligand-based drug design approaches and shows significant innovation in its application areas. However, it lacks substantial innovation in the basic structure of the generative algorithm.

**Strengths:**

1. The method provides a comprehensive summary of related work. To my knowledge, it is currently the only 3D generative model that simultaneously incorporates shape and pharmacophore constraints.
2. The authors achieved credible results in three specific tasks, partially demonstrating the method’s effectiveness.

**Weaknesses:**

1. The method represents shapes and electrostatic surfaces as point clouds and generates them together with the molecular structure using 3D graph neural networks, which may significantly impact computational efficiency. The generative model based on DDPM may also have efficiency drawbacks. The authors should provide relevant statistical results.
2. The method lacks comparisons with other methods, casting doubt on the superiority of the results.
    (1). For shape-based molecular generation, methods like ShapeMol and SQUID can achieve similar effects and should be compared. (2) For pharmacophore-based molecular generation, both pharmacophore type matching and structural matching should be considered and evaluated separately. If combined comparison is necessary, reference to 2D pharmacophore works such as the Match Score (Nature Communications | (2023)14:6234) is suggested. (3) The authors should provide the similarity of generated molecules to reference molecules for each specific task to ensure that the generated molecules are not simple variations of the reference molecules.

**Questions:**

My questions are the same as mentioned in Weaknesses.

---

> ### Author Response · Authors · 2024-11-20
> **Authors’ response to Reviewer JQjD (Part 1/n)**
>
> We thank the reviewer for their timely review and provide a point-by-point response to each of their concerns below. We also direct the reviewer to our *“Common Response to all reviewers highlighting new experiments and comparisons to related work”*, which highlights important additional experiments that we conducted in response to both this review, and the other reviews.
>
> ---
> ---
>
> > The method represents shapes and electrostatic surfaces as point clouds and generates them together with the molecular structure using 3D graph neural networks, which may significantly impact computational efficiency. The generative model based on DDPM may also have efficiency drawbacks. The authors should provide relevant statistical results.
>
> It is true that generating surfaces and pharmacophores jointly with the 3D molecule is relatively expensive, particularly with the DDPM paradigm, which is famously known to be inefficient due to needing a large number of time steps for best performance. Since our work is the first of its kind in jointly modeling shapes, electrostatics, and pharmacophores for ligand-based drug design objectives, we chose to de-emphasize computational efficiency to instead prioritize evaluating the absolute performance and merit of this approach. In our opinion, it makes more sense to emphasize efficiency only after demonstrating the objective utility of a newly proposed method to practical design tasks. We agree with the reviewer that engineering ShEPhERD to be more efficient is a worthwhile pursuit, especially now that ShEPhERD’s performance and potential impact has been demonstrated (in silico). But, efficiency engineering was not a focus of this initial work.
>
> We also note that there are many research and engineering efforts in the generative ML community that focus on improving the efficiencies of DDPMs. It is certainly possible to integrate these methods into ShEPhERD, but such performance engineering is simply not a priority for this particular submission.
>
> We do indeed report training times and typical sampling times in Appendix A.11 (“training and inference resources”). These reported times should be interpreted qualitatively, since sampling times will of course be hardware-specific, and will also depend on the context of the sampling (number of sampling steps, number of generated atoms, number of generated pharmacophores, etc).
>
> > The method lacks comparisons with other methods, casting doubt on the superiority of the results. (1). For shape-based molecular generation, methods like ShapeMol and SQUID can achieve similar effects and should be compared.
>
> Thank you for suggesting these comparisons. Please refer to our “Common Response…” which includes new comparisons to SQUID. SQUID publicly released their reference and generated test-set 3D molecules and hence made these fair comparisons possible. In summary, ShEPhERD decidedly outperforms SQUID in terms of generating chemically diverse molecules that have high shape-similarity to a reference molecule. ShapeMol performs similarly to SQUID according to the ShapeMol paper, and hence we can safely assume that ShEPhERD outperforms ShapeMol, as well.
>
> > (2) For pharmacophore-based molecular generation, both pharmacophore type matching and structural matching should be considered and evaluated separately. If combined comparison is necessary, reference to 2D pharmacophore works such as the Match Score (Nature Communications | (2023)14:6234) is suggested.
>
> We consider pharmacophore types and their positions & directions in 3D space jointly as that is standard for leading pharmacophore similarity scoring functions. For instance, ROCS (arguably the most commonly used ligand-based virtual screening tool) scores the volumetric overlap of “colored Gaussians” in 3D space in order to score shape and pharmacophore similarity. Pharmacophore models describe interaction patterns in 3D space precisely because ligand-protein interactions are 3-dimensional, and the precise spatial arrangement of those interactions matter. It isn’t practically useful to evaluate whether two molecules share the same pharmacophore types if their pharmacophores are spatially arranged in completely different ways. The reason other works will consider “2D” pharmacophore similarity is because it is usually cheaper (not better) than evaluating 3D pharmacophore similarity, which often requires conformational alignment. In this work, we perform full 3D pharmacophore scoring and alignments because it is state-of-the-art in pharmacophore similarity scoring. We also emphasize that our scoring function, which is based on  PheSA, goes beyond ROCS because it also considers the directionality of certain pharmacophore types (HBDs, HBAs, aromatic rings, and halogen bonds), making it a rigorous evaluation of 3D interaction pattern similarity.

---

> ### Author Response · Authors · 2024-11-20
> **Authors’ response to Reviewer JQjD (Part 2/n, n=2)**
>
> > (3) The authors should provide the similarity of generated molecules to reference molecules for each specific task to ensure that the generated molecules are not simple variations of the reference molecules.
>
> Thank you for this suggestion – we agree that it is necessary to evaluate whether conditionally generated molecules are chemically distinct from reference molecule. For the natural product and bioisosteric fragment merging demonstrations, we believe that it is sufficiently clear that the generated molecules are extremely distinct from the conditioning molecules. For natural product ligand hopping, we construct the task such that a molecule is only scored if its SA score is below 4.5 – this simulates the practical scenario where we want to find easier-to-synthesize, or at least less complex, drug-like small-molecule mimics of natural products. In contrast, the natural products all have SA scores exceeding 6.0, clearly indicating that the generated molecules are qualitatively distinct. For the fragment merging experiment, the generated molecules are obviously distinct because the conditioning structure is a set of overlapping fragments which do not form a coherent ligand.
>
> For the hit diversification / docking experiments using PDB ligands as the reference molecule, we have added distributions of 2D similarities between the generated molecules and the reference PDB ligand. You can find these distributions in Appendix B.2. In summary, the vast majority of generated molecules have graph similarities below 0.2, verifying that the generated samples are not simple variations of the reference molecule. We also perform these evaluations in our new comparisons to SQUID and DiffSBDD, and observe similar results.

---

> ### Author Response · Authors · 2024-11-25
> **Request for response**
>
> As the discussion period is coming to an end, we are checking in to ask whether the reviewer has seen our rebuttal, which includes multiple new experiments and evaluations that we performed at the suggestion of the reviewer (and the other reviewers) in order to significantly improve our work. We would be happy to address any last questions or concerns.

---

### Official Review · Reviewer_pbc2 · 2024-11-04

**Soundness:** 4
**Presentation:** 4
**Contribution:** 4
**Rating:** 10
**Confidence:** 5

**Summary:**

This paper presents a new method for the generation of small molecules conditioned on shape, electrostatic potential surface, or pharmacophores.  This development provides a useful and practical tool that can complement multiple conventional tasks during various stages in drug design projects.  The method is a joint diffusion model that predicts molecules in terms of graphs plus coordinates, surfaces sampled on a modest number of points (with or without an electrostatic potential), and pharmacophores (types and coordinates).  The conditional generation uses inpainting to sample the molecule graphs and coordinates conditioned on one or more other properties of the molecule, which can mimic a known target molecule.

**Strengths:**

The paper presents a substantial advance compared to previous related methods; it has a strong logical foundation with clear exposition of the method and its parameters, and a diverse set of demonstrations.  The conditional generation of molecules based on ligand-only properties is important in applied drug discovery and the method offers both a practical way for thinking about some of the concepts and a tool that can be readily applied.  In addition, it introduces useful and practical datasets, and it defines a simple but clear metric for comparing surface shapes.  I think that the work opens up a logical framework for building practical tools in drug discovery and will influence followup research.

**Weaknesses:**

Most of the weaknesses of the paper are somewhat superficial and do not limit the importance of this contribution.

All the conditional generation requires providing n1 a priori for each sample, and the pharmacophore application, which is particularly interesting and useful, requires providing both n1 and n4.  In addition to sampling n4 based on the chosen n1 and the training data distribution, have the authors considered exploring ways to automate or predict values for parameters like n1 and n4?  It would be very interesting to explore the regime where compute limitations are of no concern.

The demonstration of conditional generation in Fig 3 is performed on a random subsample of the training molecules.  I understand that there is no perfect split of molecules, however, the jump from a random subset, which work rather well, to natural products, which are completely out of distribution and don't appear to work well (albeit better than reinvent), appears a bit too dramatic.  To enhance the robustness of the evaluation and to gain additional insights, the authors could create additional, harder, splits by molecular properties such as numbers of atoms or rings, splits by scaffold, or splits by similarity clustering of the training data. Because the authors provide two new training datasets, it is perhaps more important than it would otherwise have been to provide and characterize a number of distinct, increasing difficulty splits.

The surfaces were fit on modest numbers of points and the graph embedding networks are surprisingly small, which are all probably a result of trying to keep the graph neural nets from collapsing.  It is possible that these shortcuts (and the usual problems of graph neural nets and scaling to larger systems) lead to the size-dependent effects demonstrated in the supplemental materials; these limitations might also underpin the examples of natural products.  It may be worth adding a brief note to discuss the scaling of the method with the size of the small molecules, even if the detailed analysis is reserved for the supplemental materials due to space limitations.

**Questions:**

Although it is not unreasonable, what is the logic for fitting the surfaces onto the volumes in order to obtain the parameters required for the method?  Also, why only fitting the particular set of discrete numbers of points (50, 100, 150, 200, 300, 400) and not, say, a function of the number of atoms, or of the total surface size?  Would there be a problem with scaling the method to a large number of points, or is it simply deemed unnecessary?  As a minor related question, what is the purpose of a separate set of surface points for x2 compared to x3?  Wouldn't the coordinates of x3 suffice for both tasks?

Other than following prior work, do the authors have a high level rationale for selecting the particular subset of targets from the PDB in Zheng et al (2024)?

---

> ### Author Response · Authors · 2024-11-20
> **Authors’ response to Reviewer pbc2 Part 1/n)**
>
> We thank the reviewer for their positive review; we are thrilled that other scientists see both the practical value of ShEPhERD as currently constructed, but also the generality of ShEPhERD’s framework and its potential across drug discovery and molecular design.
>
> We respond to each of the reviewer’s concerns/questions below. We also direct the reviewer to our *“Common Response to all reviewers highlighting new experiments and comparisons to related work”*, which provides a summary of the additional experiments we performed with ShEPhERD during this discussion phase.
>
> -----
> -----
>
> > All the conditional generation requires providing n1 a priori for each sample, and the pharmacophore application, which is particularly interesting and useful, requires providing both n1 and n4. In addition to sampling n4 based on the chosen n1 and the training data distribution, have the authors considered exploring ways to automate or predict values for parameters like n1 and n4? It would be very interesting to explore the regime where compute limitations are of no concern.
>
> Yes, it is certainly possible to automatically estimate or predict n1, n4, or both n1 and n4. For instance, one could train a simple ML regression model to predict n1 given {x3, x4} of a reference molecule. One could also train a simple model to predict n4 given {x1, x3} or other prior information. We did not explore such strategies because they will always be somewhat application specific, and we argue that it is better for the end-user to make an informed decision depending on their particular needs. For instance, in our bioisosteric fragment merging experiment, we chose n4 based on the number of pharmacophores left after manual interaction selection and pharmacophore clustering.
>
> We certainly agree that it would be interesting to exploring applications of ShEPhERD with more computational resources. Unfortunately we are currently unable to explore such directions ourselves.
>
> > The demonstration of conditional generation in Fig 3 is performed on a random subsample of the training molecules. I understand that there is no perfect split of molecules, however, the jump from a random subset, which work rather well, to natural products, which are completely out of distribution and don't appear to work well (albeit better than reinvent), appears a bit too dramatic. To enhance the robustness of the evaluation and to gain additional insights, the authors could create additional, harder, splits by molecular properties such as numbers of atoms or rings, splits by scaffold, or splits by similarity clustering of the training data.
>
> We used random splits for Figure 3 for multiple reasons:
> - To demonstrate ShEPhERD’s performance in-distribution, which we believe is important when introducing a new framework for modeling 3D interactions.
> - To highlight our evaluation/scoring methods in the context of in-distribution 3D molecular generation, including our (shape, electrostatic, pharmacophore) similarity scoring functions, 2D graph similarity evaluation, our self-consistency evaluations for unconditional joint generation, and our evaluations of geometric stability (e.g., RMSD before/after xTB relaxation).
> - Creating more difficult splits is computationally challenging and somewhat ill-defined in our setting, as assessing what an “in distribution” vs “out of distribution” interaction profile would potentially require screening the similarity of each candidate test set interaction profile against the millions of interaction profiles in the training set.
>
> We compensate for our use of random splits by testing ShEPhERD on challenging, often out-of-distribution tasks in Figure 4, including natural product ligand hopping and bioisosteric fragment merging. These tasks are harder than one could simulate by using arbitrary scaffold splits, while also testing ShEPhERD in tasks that are meaningful for drug discovery.
>
> We also somewhat disagree with the statement that ShEPhERD doesn’t “appear to work well” for natural product ligand hopping. While the 3D similarity scores *are* worse than those we get in Figure 3, it is uncertain what the upper bound is for this prospective task. Are there *any* drug-like small molecules with SA<4.5 that score substantially higher than the molecules found by ShEPhERD? Getting perfect scores on this task may be impossible. The best we can do is evaluate ShEPhERD against virtual screening and other tools like REINVENT, and Appendix A.1 demonstrates that ShEPhERD performs well compared to these existing methods for 3D ligand hopping.
>
> We also want to mention that our hit diversification experiments (Figure 4) offer a middle-ground between our in-distribution tasks and challenging design tasks. None of the PDB ligands are in MOSES, and many of them are larger and more complex than molecules in MOSES. Figure 20 shows that the top-scoring generated molecules are both in- and (slightly) out-of-distribution, indicating ShEPhERD’s ability to generalize.

---

> ### Author Response · Authors · 2024-11-20
> **Authors’ response to Reviewer pbc2 (Part 2/n)**
>
> > The surfaces were fit on modest numbers of points and the graph embedding networks are surprisingly small, which are all probably a result of trying to keep the graph neural nets from collapsing.
>
> Actually, both of these choices were made due to our limited GPU resources. We only trained ShEPhERD with two 32GB V100s, and had to choose small surfaces and modest embedding networks under this resource constraint. We are certainly eager to try increasing the scale of ShEPhERD. It’s worthwhile to point out that we have not yet noticed any significant training difficulties or instabilities, and our training protocols are very straightforward.
>
> > It is possible that these shortcuts (and the usual problems of graph neural nets and scaling to larger systems) lead to the size-dependent effects demonstrated in the supplemental materials; these limitations might also underpin the examples of natural products. It may be worth adding a brief note to discuss the scaling of the method with the size of the small molecules, even if the detailed analysis is reserved for the supplemental materials due to space limitations.
>
> We do already dedicate space for this type of analysis in our Appendix A.9 (“Characterizing the need for out-of-distribution performance”). We also analyze the validity of generated molecules as a function of the number of atoms in Figure 13 in Appendix A.3. We direct the reviewer to these existing analyses (which may have been missed due to our admittedly very long appendices), but would be happy to include more discussion on this topic if the reviewer believes it to be necessary.
>
>
> > Although it is not unreasonable, what is the logic for fitting the surfaces onto the volumes in order to obtain the parameters required for the method? Also, why only fitting the particular set of discrete numbers of points (50, 100, 150, 200, 300, 400) and not, say, a function of the number of atoms, or of the total surface size?
>
> We fit the surfaces onto the volumes to obtain the Gaussian width parameters because the volumetric representation has been shown to recapitulate exact shape overlap well (within 1% of hard-sphere calculations [Grant et al. 1996, "A fast method of molecular shape comparison..."]), and we want the surface similarity to recapitulate these well-established measures molecular shape similarity. Our surface similarity metric aims to reduce the overemphasis of existing volumetric similarity metrics on rewarding exact atomic overlaps.
>
> Also, visually inspecting the molecular alignments showed that the tuned parameters enable expected alignments.
>
> We only fit to a discrete number of points because we expected ShEPhERD to only operate on a fixed number of points, at least in this first version of ShEPhERD. The optimization is noisy and alignments can be expensive for pairwise calculations so a continuous function was not practically feasible for this initial study.
>
> We limited the number of surface points because the molecule training set distribution seemed to be well covered with 75 and due to computational resource limitations. If we could increase the number of points, it would very likely improve the ability to model larger molecules. This is certainly a promising direction, but just requires additional computational resources.
>
>
> > Would there be a problem with scaling the method to a large number of points, or is it simply deemed unnecessary?
>
> The issue with using a larger number of points for the diffused/denoised representations of x2/x3 is the large increase in GPU memory during both training and sampling. We only have access to 32GB GPUs, and hence simply could not use larger point clouds within ShEPhERD without significantly slowing down training.
>
> For scoring purposes, we found 400 points sufficient for scoring the shape/electrostatic similarities of small-to-medium-sized drug-like molecules with our surface-based scoring functions. When scaling to much larger structures (which we consider out-of-scope for this work), 400 points may not be sufficient. As the surface clouds become less densely sampled, the variance in the computed similarity scores increases. This is why we used 400 points for scoring rather than 75 points, which is used internally by ShEPhERD when diffusing/denoising x2 and x3 for computational efficiency.

---

> > ### Comment · Reviewer_pbc2 · 2024-11-21
> >
> > Thanks for all the answers, which are all clear and satisfactory.  It is your choice in the end, and perhaps my wording was unclear, but I meant to say:
> >
> > "It is possible that these shortcuts (and the usual problems of graph neural nets and scaling to larger systems) lead to the size-dependent effects demonstrated in the supplemental materials; these limitations might also underpin the examples of natural products. It may be worth adding a brief note *[EDIT: in the main text of the paper]* to discuss the scaling of the method with the size of the small molecules, even if the detailed analysis is reserved for the supplemental materials due to space limitations."
> >
> > In other words, I think that the scaling and behavior of the method as a function of molecule size is of enough interest to the reader so you might want to consider adding a brief note in your main text and not only in the appendix.

---

> ### Author Response · Authors · 2024-11-20
> **Authors’ response to Reviewer pbc2 (Part 3/n, n=3)**
>
> > As a minor related question, what is the purpose of a separate set of surface points for x2 compared to x3? Wouldn't the coordinates of x3 suffice for both tasks?
>
> Yes, the coordinates of x3 do suffice for both tasks (e.g., shape-conditioned generation and electrostatics-conditioned generation). We indeed made this subtle point in our main text: “Note that since x3 defines an (attributed) surface S_3, jointly modeling (x2, x3) is redundant; x3 implicitly models x2.” In both our methods section and our code, we treat these variables separate so that it is more straightforward to model P(x1, x2) or P(x1 | x2) if one wishes to. Note that modeling x3 also requires ESP calculations, which may not be desired or relevant for certain applications. In practice, we would never model P(x1, x2, x3) for the reason the reviewer cited.
>
> > Other than following prior work, do the authors have a high level rationale for selecting the particular subset of targets from the PDB in Zheng et al (2024)?
>
> We chose these targets for four reasons.
> 1. As the reviewer suggested, a primary motivation was to follow existing work.
> 2. These PDB targets are conveniently included in Therapeutic Data Commons, and have their protein pockets already prepped for docking. This made these pockets simply convenient to choose for our evaluations with AutoDock Vina.
> 3. Zheng et al (2024) use these targets to evaluate a number of generative models explicitly designed for SBDD. This gave us the opportunity to easily compare ShEPhERD against such generative SBDD models – an opportunity we took in our new experiments described in our “Common Response …”.
> 4. Zheng et al (2024) pointed out that these protein targets are involved in various different biological functions and are diverse in structure.

---

### Official Review · Reviewer_HW8X · 2024-11-07

**Soundness:** 2
**Presentation:** 4
**Contribution:** 4
**Rating:** 8
**Confidence:** 5

**Summary:**

This paper introduces ShEPhERD, a generative SE(3)-equivariant diffusion model for molecular drug design that incorporates 3D structural features inspired by medicinal chemistry, namely shape, electrostatics, and pharmacophoric properties. The authors demonstrate ShEPhERD's utility for ligand-based drug design tasks, including ligand hopping, bioactive hit diversification, and bioisosteric fragment merging.

I appreciate the approach of leveraging prior target and chemical information to enhance generative models and commend the authors for advancing in this direction. While the case studies on specific targets are excellent, the general evaluation and particularly the comparative analysis with other methods could be significantly strengthened. I am inclined to give a strong score if these concerns are addressed.

**Strengths:**

- The paper is comprehensive and well-written, with excellent figures and molecular visualizations.
- Although conditioning on pharmacophoric features is not new in the field, this is the most thoughtful implementation of the approach to date and a very important direction more works should look into. The authors obviously have a very good understanding of drug discovery and SBDD that is not open seen at ML conferences.
- In my opinion, the field is overly obsessed with only focusing on de novo design models that cannot leverage any target specific data. Therefore, I really like the fact they were able to use experimental fragments screens to get better designed molecules.
- The inclusion of comparisons to REINVENT and random docking screens as baselines is a valuable addition, as these are seldom seen in DDPM literature. (However, I believe there is still room for improvement in the comparisons and evaluations—see below).

**Weaknesses:**

**Main limitations:**

- While the method is strong, main limitation of the paper as it currently stands is the extreme lack of comparisons to related works in the experiments conducted. I kindly request the authors to benchmark against previous works in the relevant tasks. I am not looking for SOTA on all metrics, but better benchmarking is essential to understand the strengths and limitations of the work and I cannot recommend a strong accept at ICLR without more comparisons.
    - There are multiple works in a similar flavour that propose inpainting and/or conditioning on pharmacophore features of some kind. E.g.  DiffSBDD+inpainting [1], MolSnapper [2] and Diffint [3].
    - Compare to DiffHopp [4]/TurboHopp [5] in case of hit diversification. Obviously these models are only in the structure-based context.
    - Compare to ChemProjector [6]/SynFormer [7] incase of ligand hopping.
- Furthermore, the metrics could be greatly improved:
    - Heavy reliance on only Vina scores and shape/tanimoto similarity.
    - The work cites the PoseCheck [8] paper but does not use any of the metrics. For example, clashes are not measured. In the target conditioned cases, does ShEPhERD provide poses without clashes before relaxation? Is the model able to design hydrogen bonding networks better than previous SBDD models?
    - Strain energy is calculated in A.3 and the model appeared to perform well. However, the impact of this is greatly diminished by not comparing to other works. I would be grateful if the authors could include this to give context. Do the authors use the implentation in PoseCheck or their own? Is the strain energy calculated for poses bound inside the protein pocket?
    - PoseBusters [9] validity could also be used to access the quality of the structures.
- An expanded related work section covering models that perform ligand hopping, scaffold hopping and fragment merging would base the work in the literature much better.
- Would be nice to see anonymous code in the review stage.
- In the hit diversiation section, the authors claim that ShephERD produces enriched sampled “Despite having no knowledge of the protein targets”. While it is technically true the model has never directly see the receptor coordinates, a large amount of structural information is leaked/provided based on the complimentarity of the 7 experimental ligands. This is not a weakness of the method but I believe the language to be misleading. A simple rewording should do.
- The joint model/inpainting method used to condition on the target profile is not novel, as this was done in DiffSBDD 2 years ago [1]. A citation to that work and the original RePaint paper reproducing the ‘replacement’ method of inpainting in the main text would suffice.
- In the interaction-conditioned generation section you only generate 20 samples per target and filter based on interaction similarity. How many samples are left after filtering? How can you ensure statistical significance?
- It is not clear to me if any attempt was made to reduce data leakage between train and test sets (except for size) as the splits seem random. How can you be sure that the target interaction profile on the test samples are not the same/similar to those seem during training?
- Comparison to REINVENT.
    - Was REINVENT trained to convergence? 10,000 oracle calls seems quite low in my experience? (Happy to be proved wrong)
    - Can the authors clarify exactly which version of REINVENT they use? They seem to use an old version from the PMO benchmark. This is the 2017 implementation I believe? I would encourage them to use the largest REINVENT4, this is easy to use and a strong baseline in my experience (https://github.com/MolecularAI/REINVENT4).
    - Little measure of synthetic accessibility in the work. REINVENT does surprisingly well on SA score in my experience. Can this be benchmarks quantitively?

**Minor points:**

- Some background on various inpainting/conditioning strategies for DDPMs would be nice (i.e. replacement method [what is used here], classifier guidance, reconstruction guidance, Doobs-H transform)
- The generated samples in Appendix A.10 are quite small. It would be nice to include some larger samples as well to see how the model handles larger structures.
- The neural network architecture is described in great detail. In my opinion taking too much space. I would prefer to see this in the appendix with more discussion on experiments in the main text.
- What was the intuition for scaling the pharmacophore feature vector by 2.0?
- Can the target profile guidance be more or less ‘weakly’ enforced?
- It would be nice to see the fragment merging work replicated on more than one target. I would recommend the Mpro screen (on Fragalysis already) for example. You can then compare for novelty against the molecules derived from than screen in the COVID moonshot- To what extent does conditioning with the ShEPhERD method limit diversity of new samples? Could be that previous DDPMs produce on average worse samples but allow for the exploration of novel chemotypes? (My guess is this method is still very useful but I would like to see this discussed/measured)
- Have you tried an ablation of the joint noising regime? I believe in DiffSBDD its benefits was found to be marginal. [1]
- Could your pharmacophore similarity metric be seen as related to interaction fingerprint similarity?
- Is it possible to use ShEPhERD completely de novo? I.e. for a new target without example molecules in the PDB? Could pharmacophore features be constructed from hotspot maps/manual feature engineering?
- The fragment merging protocol is nice, can a semi-automated pipeline be provided in the code?
- Section A.9 correctly points out the disparity between the size of ligand seen during training and those seen during evaluation. Why not include larger ligands in the training set?
- Given the large training set size, it would be nice to see some scaling law experiments in terms of dataset and model size. (Given that I am not aware of the resources available to the authors this will not impact my review score - I am just interested if its possible)..
    - Can you provide a list of the 7 PDBs used to perform these experiments?
- Figure 1 + 2:
    - Are both very good: I would consider merging to safe space and add more analysis/discussion.
- Figure 3:
    - Can ‘dataset samples’ be added to the histograms in the right of the figure?
- Figure 4:
    - The PyMol in this paper is generally very good but the overlapping molecules in this figure is highly confusing. A key/color map would be very helpful. (Especially for hit diversification section).
    - REINVENT molecules seem to consistently outperform ShEPhERD on SA scores. I would be good to include an example structure instead of just listing the metrics.
    - The PDB targets are quite well chosen (Kinases/COVID MPro etc), adding target names in brackets next to PDBs would be helpful.


1. Schneuing, Arne, et al. "Structure-based drug design with equivariant diffusion models." arXiv preprint arXiv:2210.13695 (2022).
2. Ziv, Yael, Brian Marsden, and Charlotte M. Deane. "Molsnapper: Conditioning diffusion for structure based drug design." bioRxiv (2024): 2024-03.
3. Sako, Masami, Nobuaki Yasuo, and Masakazu Sekijima. "DiffInt: A Pharmacophore-Aware Diffusion Model for Structure-Based Drug Design with Explicit Hydrogen Bond Interaction Guidance." (2024).
4. Torge, Jos, et al. "Diffhopp: A graph diffusion model for novel drug design via scaffold hopping." arXiv preprint arXiv:2308.07416 (2023).
5. Yoo, Kiwoong, et al. "TurboHopp: Accelerated Molecule Scaffold Hopping with Consistency Models." arXiv preprint arXiv:2410.20660 (2024).
6. Luo, Shitong, et al. "Projecting Molecules into Synthesizable Chemical Spaces." arXiv preprint arXiv:2406.04628 (2024).
7. Gao, Wenhao, Shitong Luo, and Connor W. Coley. "Generative Artificial Intelligence for Navigating Synthesizable Chemical Space." arXiv preprint arXiv:2410.03494 (2024).
8. Harris, Charles, et al. "Posecheck: Generative models for 3d structure-based drug design produce unrealistic poses." NeurIPS 2023 Generative AI and Biology (GenBio) Workshop. 2023.
9. Buttenschoen, Martin, Garrett M. Morris, and Charlotte M. Deane. "PoseBusters: AI-based docking methods fail to generate physically valid poses or generalise to novel sequences." Chemical Science 15.9 (2024): 3130-3139.

**Questions:**

- To what extent does conditioning with the ShEPhERD method limit diversity of new samples? Could be that previous DDPMs produce on average worse samples but allow for the exploration of more novel chemotypes? (My guess is this method is still very useful but I would like to see this discussed/measured)
- Have you tried an ablation of the joint noising regime? I believe in DiffSBDD its benefits was found to be marginal.
- Could your pharmacophore similarity metric be seen as related to interaction fingerprint similarity?
- Is it possible to use ShEPhERD completely de novo? I.e. for a new target without example molecules in the PDB? Could pharmacophore features be constructed from hotspot maps/manual feature engineering?
- The fragment merging protocol is nice, can a semi-automated pipeline be provided in the code?
- Section A.9 correctly points out the disparity between the size of ligand seen during training and those seen during evaluation. Why not include larger ligands in the training set?
- Given the large training set size, it would be nice to see some scaling law experiments in terms of dataset and model size. (Given that I am not aware of the resources available to the authors this will not impact my review score - I am just interested if its possible).

---

> ### Author Response · Authors · 2024-11-20
> **Authors’ response to Reviewer HW8X (Part 1/n)**
>
> We thank the reviewer for their remarkably comprehensive review and many suggestions for improving our work. We have strived to respond to each of the raised concerns/questions below, but we also direct the reviewer to our “Common Response to all reviewers highlighting new experiments and comparisons to related work”, which discusses the new experiments and evaluations that we have performed at the request of this reviewer and the other reviewers. We believe that these additional experiments thoroughly address many of the valid concerns that the reviewer raised, while also substantially enhancing the quality of the submission.
>
> ---
> ---
>
> > While the method is strong, main limitation of the paper as it currently stands is the extreme lack of comparisons to related works in the experiments conducted. I kindly request the authors to benchmark against previous works in the relevant tasks. I am not looking for SOTA on all metrics, but better benchmarking is essential to understand the strengths and limitations of the work and I cannot recommend a strong accept at ICLR without more comparisons.
> > - There are multiple works in a similar flavour that propose inpainting and/or conditioning on pharmacophore features of some kind. E.g. DiffSBDD+inpainting [1], MolSnapper [2] and Diffint [3].
> > - Compare to DiffHopp [4]/TurboHopp [5] in case of hit diversification. Obviously these models are only in the structure-based context.
> > - Compare to ChemProjector [6]/SynFormer [7] incase of ligand hopping.
>
> We thank the reviewer for prompting this opportunity for us to perform additional experiments that showcase the capabilities of ShEPhERD relative to other related (even if not directly comparable) generative modeling works. Our *“Common Response to all reviews…”* discusses our new experiments in full. In summary, we newly:
> - Compare to SQUID in shape-conditioned generation
> - Compare to DiffSBDD in SBDD for the PDB target 4tos, which was reported on in the DiffSBDD paper.
> - Compare to 5 other SBDD generative models on our 7 PDB targets, using benchmark results reported in Zheng et. al., “Structure-based drug design benchmark: Do 3d methods really dominate?”.
> - Compare to SynFormer on our natural product ligand hopping tasks
>
> It is also important to note that ShEPhERD is addressing very different, often orthogonal, tasks compared to the models that the reviewer cited:
> - *ShEPhERD does not condition on protein pockets*, in contrast to SBDD methods like DiffSBDD, MolSnapper, DiffInt, and DiffHopp/TurboHopp. Nevertheless, as our new experiments show, ShEPhERD can often perform better than these (or other) SBDD generative models via its LBDD formulation, at least when the reference ligand is itself a high-scoring hit.
> - MolSnapper guides a 3D DDPM to form Nitrogen or Oxygen atoms that could potentially be hydrogen bond acceptors or donors. *MolSnapper does not consider other highly relevant pharmacophores* like aromatic groups or hydrophobes, and it is unclear how it could be adapted to do so. Hence, ShEPhERD is far more expressive and general in terms of pharmacophore-based design than MolSnapper. We also note that *ShEPhERD could incorporate MolSnapper’s guidance strategy* by also inpainting the exact positions and types of a subset of particular atoms, if one wished to do so.
> - DiffInt, which was preprinted ~2 months before the ICLR submission deadline, also only considers hydrogen bonding interactions, and they only consider hydrogen bonding interactions in the context of protein-pocket-conditioned molecular design. ShEPhERD is far more generally applicable because it can condition on pharmacophores of ligands in any chemical context, or in the absence of an external chemical environment. For instance, DiffInt (and MolSnapper, and any existing SBDD generative model) could not be used for interaction-preserving natural product ligand hopping.
> - DiffHopp/TurboHopp are designed for straightforward fragment linking, but *their fragment linking strategies do not attempt to preserve explicitly-defined 3D interactions* beyond implicitly conditioning on a protein structure. DiffHopp/TurboHopp cannot be used for bioisosteric fragment merging, nor could they be used for full ligand hopping (e.g., natural product ligand hopping), especially in the absence of a protein pocket. ShEPhERD is simply addressing different tasks (and we focus on tasks like that other methods cannot perform), so it does not make sense to compare them.
> - Although we do perform additional comparisons to SynFormer, SynFormer/ChemProjector are addressing completely orthogonal tasks compared to ShEPhERD. SynFormer/ChemProjector aim to find *synthesizable analogues that preserve 2D chemical similarity*, whereas ShEPhERD aims to find *chemically dissimilar* analogues that preserve 3D interaction similarity.

---

> ### Author Response · Authors · 2024-11-20
> **Authors’ response to Reviewer HW8X (Part 2/n)**
>
> > Furthermore, the metrics could be greatly improved:
> > - Heavy reliance on only Vina scores and shape/tanimoto similarity.
>
> We use Vina scores and shape/electrostatic/pharmacophore Tanimoto similarity scores as our primary metrics of in silico success because this is standard for structure-based and ligand-based drug design works, respectively. We also report extensive validity metrics, RMSDs upon xTB relaxation, strain energies, and 2D graph similarities where appropriate.
>
> > - The work cites the PoseCheck [8] paper but does not use any of the metrics. For example, clashes are not measured. In the target conditioned cases, does ShEPhERD provide poses without clashes before relaxation? Is the model able to design hydrogen bonding networks better than previous SBDD models?
>
> We do not use the PoseCheck metrics because ShEPhERD does not generate molecules inside protein pockets, and does not encode pocket structure. Hence, measuring steric clashes is not applicable. Our “target conditioned cases” are not explicitly protein target-conditioned and we do not relax in the context of a protein pocket – our model is ligand conditioned only. All of our reported protein-bound poses are obtained upon docking (from a SMILES), and hence measuring steric clashes would just be evaluating how good Vina is at avoiding steric clashes. As for comparisons to SBDD models, we perform explicit comparisons in terms of Vina scores in our new experiments (see our *“Common Response...”*). We note that ShEPhERD does not condition on hydrogen bonding networks of protein-ligand interactions since we condition only on the pharmacophores of the ligand unbiased by its specific binding interactions with a protein target.
>
>
> > - Strain energy is calculated in A.3 and the model appeared to perform well. However, the impact of this is greatly diminished by not comparing to other works. I would be grateful if the authors could include this to give context. Do the authors use the implementation in PoseCheck or their own? Is the strain energy calculated for poses bound inside the protein pocket?
>
> We do not compare to other works because to our knowledge, we are the first to compute strain energies with xTB (semi-empirical DFT). Other works like PoseCheck use UFF (a classical force field) to evaluate strain energies, and the strain energies computed with xTB vs. UFF are generally incomparable. We use xTB strain energies (and RMSDs upon xTB relaxation) because we train on xTB-optimized geometries, and hence we should evaluate ShEPhERD’s ability to generate 3D structures at the same level of theory. We use our own implementation of strain energy for this reason.
>
> We also note that PoseCheck’s use of UFF for calculating strain energies is not exactly rigorous because most of the generative models that they evaluate are not trained on UFF-optimized geometries. As a point of comparison, if we compute the xTB-based strain energies of MMFF94-optimized structures, they are generally quite large (see our Table 3 in Appendix A.3).
>
> We do not calculate strain energies for protein-bound poses because ShEPhERD does not generate protein-bound poses.
>
> > - PoseBusters [9] validity could also be used to access the quality of the structures.
>
> We implement our own validity checks (which use a very rigorous definition of validity; see Appendix A.3) because we generate molecules with explicit hydrogens included (most generative models in 3D do not do this, or perform substantially worse when doing so because of the additional complexity). We generate molecules with explicit hydrogens, as explicit hydrogens are needed to appropriately define hydrogen bond donors/acceptors and their directionalities. We also implement our own validity checks because we are the only method (to our knowledge) that assess validity/quality with xTB relaxations. We use xTB relaxations because such relaxations are agnostic to graph structure and are of higher quality than classical force fields.
>
> PoseBusters also evaluates quality of structures by referencing empirical histograms of bond angles, etc. from the Cambridge Structural Database. While this may be appropriate for models that train on crystal structures of protein-bound ligands, we train on ligands *only* that are optimized with xTB. Hence, we should evaluate the geometric quality ShEPhERD’s samples with respect to the geometric quality of the training set (e.g., by using xTB strain energies).

---

> ### Author Response · Authors · 2024-11-20
> **Authors’ response to Reviewer HW8X (Part 3/n)**
>
> > Would be nice to see anonymous code in the review stage.
>
> We did submit our anonymized code along with our initial paper submission. This is also highlighted in our Reproducibility Statement.
>
> > In the hit diversification section, the authors claim that ShephERD produces enriched sampled “Despite having no knowledge of the protein targets”. While it is technically true the model has never directly see the receptor coordinates, a large amount of structural information is leaked/provided based on the complementarity of the 7 experimental ligands. This is not a weakness of the method but I believe the language to be misleading. A simple rewording should do.
>
> The reviewer has highlighted a subtle but important point. Unlike models trained explicitly for structure-based drug design (e.g, DiffSBDD, Pocket2Mol, etc.), ShEPhERD does not ever encode, see, or condition-on protein structures, protein sequences, etc. ShEPhERD is strictly a ligand-only model that sees interaction profiles of a ligand provided in a potentially arbitrary conformation. In Figure 4, we conduct experiments where ShEPhERD conditions on the lowest-energy conformation of the experimental PDB ligand. This conformation does not leak any information about the binding pose itself, nor do we bias the extraction of the interaction profiles based on the protein pocket. This task simulates the real-world ligand-based drug design scenario where you only know that a “hit” ligand displays bioactivity, but you don’t know which protein the ligand binds to. This is a common in phenotypic experimental screens. Hence, it is true that in this scenario, ShEPhERD “has no knowledge of the protein targets” — all it knows is that a reference ligand shows bioactivity towards *something*.
>
> Having said this, we have changed our language to “having no **explicit** knowledge of the protein target…”.
>
> > The joint model/inpainting method used to condition on the target profile is not novel, as this was done in DiffSBDD 2 years ago [1]. A citation to that work and the original RePaint paper reproducing the ‘replacement’ method of inpainting in the main text would suffice.
>
> We do cite both DiffSBDD and RePaint in our paper, but have added these citations to our “Sampling” section.
>
> > In the interaction-conditioned generation section you only generate 20 samples per target and filter based on interaction similarity. How many samples are left after filtering? How can you ensure statistical significance?
>
> We only filter samples that are invalid or have graph similarity above 0.3 (which would inflate 3D similarity metrics). We *do not* filter based on interaction similarity. As reported in App. A.3, the validity rates for our conditional generation experiments on GDB17 are above 90%, and nearly all the samples have graph similarity below 0.3 (see 2D Fingerprint Similarity distributions in Figure 3). Hence, the vast majority of samples (>90%) remain after filtering.
>
> > It is not clear to me if any attempt was made to reduce data leakage between train and test sets (except for size) as the splits seem random. How can you be sure that the target interaction profile on the test samples are not the same/similar to those seen during training?
>
> We used random splits for Figure 3 for multiple reasons:
> - To demonstrate ShEPhERD’s performance in-distribution, which is important when introducing a new framework for modeling 3D interactions.
> - To highlight our evaluation/scoring methods in the context of in-distribution 3D molecular generation, including our (shape, electrostatic, pharmacophore) similarity scoring functions, 2D graph similarity evaluation, our self-consistency evaluations for unconditional joint generation, and our evaluations of geometric stability (e.g., RMSD before/after xTB relaxation).
> - Creating more difficult splits is computationally challenging and somewhat ill-defined in our setting, as assessing what an “in distribution” vs “out of distribution” interaction profile would potentially require screening the similarity of each candidate test set interaction profile against the millions of interaction profiles in the training set.
>
> We try to compensate for our use of random splits by testing ShEPhERD on many challenging, often clearly out-of-distribution tasks in Figure 4, including natural product ligand hopping and bioisosteric fragment merging. These tasks are clearly harder than one could simulate by using arbitrary scaffold splits, while also testing ShEPhERD in tasks that are actually meaningful for drug discovery.
>
> We also want to mention that our hit diversification experiments (also in Figure 4) offer a middle-ground between our in-distribution tasks and challenging design tasks. None of the PDB reference ligands are in MOSES, and many of them are slightly larger and more complex than molecules in MOSES. Figure 20 shows that the top-scoring generated molecules are both in- and (slightly) out-of-distribution, indicating ShEPhERD’s ability to generalize.

---

> ### Author Response · Authors · 2024-11-20
> **Authors’ response to Reviewer HW8X (Part 4/n)**
>
> > Comparison to REINVENT.
> > - Was REINVENT trained to convergence? 10,000 oracle calls seems quite low in my experience? (Happy to be proved wrong)
> > - Can the authors clarify exactly which version of REINVENT they use? They seem to use an old version from the PMO benchmark. This is the 2017 implementation I believe? I would encourage them to use the largest REINVENT4, this is easy to use and a strong baseline in my experience (https://github.com/MolecularAI/REINVENT4).
> > - Little measure of synthetic accessibility in the work. REINVENT does surprisingly well on SA score in my experience. Can this be benchmarked quantitatively?
>
> We used REINVENT 2.0 (https://pubs.acs.org/doi/10.1021/acs.jcim.0c00915 ) from 2020. We used the implementation used in the PMO benchmark precisely because the PMO benchmark extensively evaluated this implementation and found it (in general) to be superior to most other molecular optimization methods. We have no reason to believe that this particular version of REINVENT is unsatisfactory. We also capped the number of oracle calls to 10,000 because this is what was done in the PMO benchmark, which again found REINVENT to be quite performant in this setting. In theory, one could run both REINVENT and ShEPhERD for any number of iterations. To establish a fair comparison, we had to pick some number of oracle calls. We emphasize that although we allowed 10,000 oracle calls for REINVENT, we limited ShEPhERD to 2500 generated molecules and only scored the molecules that were valid (even though validity-based filtering doesn’t strictly use up an “oracle call”). We do expect that REINVENT would find better scoring molecules if we ran it for longer; we also expect the same for ShEPhERD, as we have no reason to believe that ShEPhERD has converged in performance, either.
>
> To explicitly check for convergence, we plot the average score of each REINVENT batch vs. the iteration number for each natural product target in the new Appendix B.5. For 1 natural product, REINVENT appears to have converged. For another natural product, REINVENT appears close to converged. For the last natural product, REINVENT could likely find better scoring molecules if trained for longer. Again, we emphasize that we could also run ShEPhERD for longer, too.
>
> In practice, even allowing for just 10,000 REINVENT samples required multiple days on a GPU per target because of the expensive conformer sampling, xTB optimization, and alignment with our scoring functions. We also note that in contrast to REINVENT, ShEPhERD is not specifically trained to optimize similarity for any natural product (or for any particular molecule).
>
> We also plot SA score distributions in the new Appendix B.5. REINVENT does indeed do surprisingly well on generating molecules with lower SA scores, but also performs less well at optimizing 3D similarity. We note that we don’t focus on synthetic accessibility because synthesizability metrics are imperfect and because ShEPhERD is not designed with synthesizability in mind (as opposed to works like SynNet, SynFormer, etc.). We do show SA scores of ShEPhERD-generated molecules where relevant, and note that in our natural product ligand hopping experiments, we only score molecules that have SA scores below 4.5, as the task involves trying to generate structurally simpler small molecules that mimic the 3D interactions as the natural products.
>
>
> > An expanded related work section covering models that perform ligand hopping, scaffold hopping and fragment merging would base the work in the literature much better.
>
> To our knowledge, only Wills et al. (2024) have considered fragment merging in the style that we do, and we do highlight Wills et al. (2024) in our introduction. Other works are limited to more straightforward fragment linking, and we do cite some of these approaches in our Related Work (e.g., Schneuing et al., 2022). As for ligand/scaffold hopping, we extensively discuss models that have been designed for 3D-aware ligand/scaffold hopping in our Related Work section on “3D interaction-aware molecular generation”. We also already discuss some works that do interaction-aware design for scaffold-hopping in 1D/2D (Skalic et al., 2019; Imrie et al., 2021; Zhu et al., 2023; Xie et al., 2024, Papadopoulos et al., 2021). These works are most related to our own, as they approach scaffold/ligand hopping from a shape/pharmacophore perspective. Many other methods approach scaffold/ligand hopping through virtual screening, and we also extensively discuss these approaches in our Related Work section on “3D similarity scoring for ligand-based drug design”. If the reviewer could list particular works that we missed that they feel are especially relevant to ShEPhERD without going out-of-scope, we’d be happy to include them.

---

> ### Author Response · Authors · 2024-11-20
> **Authors’ response to Reviewer HW8X (Part 5/n)**
>
> > Some background on various inpainting/conditioning strategies for DDPMs would be nice (i.e. replacement method [what is used here], classifier guidance, reconstruction guidance, Doobs-H transform)
>
> Our work is not intended to be a review of guidance strategies for DDPMs, as we only use a basic form of inpainting guidance from RePaint (which we do describe in practical detail, and visualize in Fig 2). We did not consider these other guidance strategies that the reviewer mentioned, and including their discussion in this work would be out-of-scope. There are other works that describe guidance strategies much better than what we could describe here while remaining on-topic.
>
> > The generated samples in Appendix A.10 are quite small. It would be nice to include some larger samples as well to see how the model handles larger structures.
>
> Figure 9 does contain four examples of larger structures that were generated in our fragment merging experiments. Figure 4 also contains examples of medium-sized molecules (which are still typically larger than molecules we trained on) that were generated for the natural product ligand hopping experiments. Beyond these practical applications, we hesitate to overemphasize performance on larger structures because ShEPhERD isn’t currently trained on large structures. Nevertheless, we’d be happy to include more if these already reported structures are not sufficient for conveying ShEPhERD’s ability to generalize to (moderately) larger structures.
>
> > The neural network architecture is described in great detail. In my opinion taking too much space. I would prefer to see this in the appendix with more discussion on experiments in the main text.
>
> We thank the reviewer for this perspective. We include these descriptions of the network for two reasons:
> - Our paper is quite interdisciplinary, and is likely to be read by those with varying levels of familiarity with diffusion modeling and 3D network design. Whereas some may not need certain descriptions to understand the work, others may appreciate their inclusion. We personally believe the descriptions of the diffusion model and the network design improve the paper’s accessibility.
> - We include the descriptions of the diffusion model and (high-level) network architecture to emphasize ShEPhERD’s simplicity and extensibility. By including these descriptions, we aim for others to take inspiration from ShEPhERD’s framework in their own work on interaction-aware molecular design.
>
> > What was the intuition for scaling the pharmacophore feature vector by 2.0?
>
> We took inspiration from the EDM paper, which scaled atom types under the intuition that “when the features h are defined on a smaller scale than the coordinates x, the denoising process tends to first determine rough positions and decide on the atom types only afterwards”.
>
> Similarly, we hypothesized that it would be beneficial for pharmacophore-conditioned generation for the pharmacophores to become resolved earlier in the denoising process. Hence, we scaled the features by 2.0. This was not a rigorous choice, and we did particularly tune this hyperparameter.
>
> > Can the target profile guidance be more or less ‘weakly’ enforced?
>
> Yes, this can be easily done by only inpainting up to a certain time step $t^* > 0$. Presumably, this would increase ligand diversity while reducing 3D similarity to the target profile.
>
> > Can you provide a list of the 7 PDBs used to perform these experiments?
>
> These PDB ligand IDs are already listed in the first paragraph of A.2.4.
>
> > Figure 1 + 2: Are both very good: I would consider merging to safe space and add more analysis/discussion.
>
> Thanks for this suggestion. We prefer to keep them separate, as Figure 1 provides a schematic overview of the paper – and should be present near the beginning of the paper – whereas Figure 2 includes technical information that is not discussed in the main text until page 4 or 5.
>
> > Figure 3: Can ‘dataset samples’ be added to the histograms in the right of the figure?
>
> Can the reviewer clarify exactly what subfigure they are referring to? We would be happy to make adjustments.
>
> > The PyMol in this paper is generally very good but the overlapping molecules in this figure is highly confusing. A key/color map would be very helpful. (Especially for hit diversification section).
>
> For the “Overlaid docked poses” subfigure in Figure 4, we did consider using color-coded molecules, but this was difficult to interpret as heteroatoms were no longer readily visible. The main purpose of this subfigure is to qualitatively show that the top-3 generated ligands for each PDB target explore docking poses that closely align with the crystal pose of the PDB ligand. We have included the individual poses of each generated ligand in the new Figure 34 in Appendix B.6, for clarity.

---

> ### Author Response · Authors · 2024-11-20
> **Authors’ response to Reviewer HW8X (Part 6/n)**
>
> > REINVENT molecules seem to consistently outperform ShEPhERD on SA scores. I would be good to include an example structure instead of just listing the metrics.
>
> We do indeed provide example structures of REINVENT-generated molecules in Figure 5, Appendix A.1.
>
> > The PDB targets are quite well chosen (Kinases/COVID MPro etc), adding target names in brackets next to PDBs would be helpful.
>
> We are in the process of updating these figures while preserving formatting, and will incorporate these suggestions. Some of these changes may not appear until the final version of the paper.
>
> > To what extent does conditioning with the ShEPhERD method limit diversity of new samples? Could be that previous DDPMs produce on average worse samples but allow for the exploration of more novel chemotypes? (My guess is this method is still very useful but I would like to see this discussed/measured)
>
> Conditioning molecule generation on a target shape, electrostatics, or pharmacophores certainly limits diversity of new samples compared to an unconditional generative model, by definition. However, we emphasize that ShEPhERD does not just generate “me-too” compounds that display only minor chemical graph variations compared to the reference ligand. In practically all of our evaluations of graph similarity, the majority of conditionally generated molecules have graph similarities to the reference below 0.2. Qualitatively, Figure 4 displays the 2D molecular graphs of conditionally generated molecules, and it is evident that ShEPhERD is generating diverse molecules with substantial chemical variety. We attribute this result to our choice of using surface based shape/electrostatic representations (which don’t specify exact atomic coordinates), our use of joint diffusion / inpainting, and the fact that interaction profiles themselves are a lossy abstraction of the molecular structure.
>
> We also note that if one wishes to relax the shape/electrostatic/pharmacophore constraints, they may easily do so by only inpainting up to a certain time step $t^* > 0$, and then permitting the model to finish generation for $t < t^*$ without conditioning.
>
> > Have you tried an ablation of the joint noising regime? I believe in DiffSBDD its benefits was found to be marginal.
>
> If we understand correctly, the reviewer is referring to the observation that DiffSBDD performs similarly when conditioning only on the clean protein pocket (during both training and inference) versus jointly denoising both the ligand+pocket and using inpainting for pocket-conditioning at inference. In early model development, we did consider just conditioning the generation of x1 on the exact {x2, x3, x4}, but found that this conditioning strategy drastically reduces the diversity of the generated ligands. This is presumably because a well-trained model can easily generate the exact x1 (or a close match) from un-noised {x2, x3, x4}. By learning the joint distribution with DDPMs and using inpainting, ShEPhERD generates much more diverse ligands that still score highly with regard to shape, electrostatic, and pharmacophore similarity. Hence, we focused on the joint diffusion strategy.
>
> > Could your pharmacophore similarity metric be seen as related to interaction fingerprint similarity?
>
> They are related but have subtle differences. Interaction fingerprints usually represent explicit interactions present in a protein-ligand complex. Our pharmacophore representations are ligand-centric, and do not consider external binding partners. In most of our experiments, our pharmacophore similarity scoring metric considers all possible pharmacophores on the ligand, even if some of those modes may not be activated upon binding with a particular target. That said, our pharmacophore similarity metric can also be used for interaction fingerprint similarity if you only consider those pharmacophores that are directly involved in binding, like what we do in our fragment merging experiment.
>
> > Is it possible to use ShEPhERD completely de novo? I.e. for a new target without example molecules in the PDB? Could pharmacophore features be constructed from hotspot maps/manual feature engineering?
>
> In principle this is certainly possible, and would be an interesting strategy to directly adopt ShEPhERD to de novo structure-based drug design. Because we focus on ligand-based drug design in this submission, we did not explore this particular application, which would likely require considerable algorithm development for constructing pharmacophore and/or electrostatic features from a protein pocket. We are hopeful that others will find ShEPhERD useful in these types of applications.

---

> ### Author Response · Authors · 2024-11-20
> **Authors’ response to Reviewer HW8X (Part 7/n, n=7)**
>
> > It would be nice to see the fragment merging work replicated on more than one target. I would recommend the Mpro screen (on Fragalysis already) for example. You can then compare for novelty against the molecules derived from than screen in the COVID moonshot
>
> > The fragment merging protocol is nice, can a semi-automated pipeline be provided in the code?
>
> Our fragment merging experiment required considerable manual effort to select and cluster the fragments’ pharmacophores that we observed to be involved in binding. While certain aspects of this workflow could in principle be automated (e.g., interaction detection with Fragalysis and using heuristic clustering rules), we expect that the protocol to extract relevant interactions from an experimental fragment screen will be (and perhaps should be) case-specific. We ourselves are also not experts in fragment-based drug design, and our protocols could likely be improved. We argue that developing a more streamlined and broadly-applicable fragment-merging tool – and evaluating on more fragment merging case studies –  would be better suited for a stand-alone paper with a different primary audience than ICLR. Our primary motivation in this initial paper is to just highlight that fragment merging is readily feasible with ShEPhERD.
>
> > Section A.9 correctly points out the disparity between the size of ligand seen during training and those seen during evaluation. Why not include larger ligands in the training set?
>
> We used the MOSES dataset (which includes molecules up to 27 non-hydrogen atoms) because it has been used by other works in shape-conditioned molecular generation (SQUID, ShapeMol, etc.) and because it is a commonly used dataset for benchmarking molecular generative models. We also note that other works have observed that 3D DDPMs for molecular generation struggle to generate large molecules while maintaining validity, and we wanted to focus on our primary task of bioisosteric drug design rather than attempting to improve the baseline performance of unconditional generative models. Now that ShEPhERD has proven capable of molecular design on small-to-medium sized drug-like molecules – and indeed shows the capacity to generate larger ligands than it was trained on – it makes sense for future work to consider scaling ShEPhERD and explicitly training on larger ligands. This, of course, will demand greater computational resources.
>
> > Given the large training set size, it would be nice to see some scaling law experiments in terms of dataset and model size. (Given that I am not aware of the resources available to the authors this will not impact my review score - I am just interested if its possible).
>
> We are also interested in seeing how the performance of ShEPhERD scales with model size, particularly given its promising capabilities even at relatively small model sizes (4M - 6M learnable parameters). For the moment, we are unable to perform these experiments ourselves due to our limited access to GPU resources.

---

> > ### Comment · Reviewer_HW8X · 2024-11-22
> >
> > I thank the authors for their remarkably comprehensive rebuttal! I agree with or am very happy with 90% of points made in the rebuttal.
> >
> > While, there still are a few points I would push back on (and I will endeavour to write a further rebuttal to some of your points over the weekend), I already think the work is far beyond the usual standard for a drug discovery paper at an ML conference and have therefore raised my score to 8.

---

### Author Response · Authors · 2024-11-20
**Common Response to all reviewers highlighting new experiments and comparisons to related work (Part 1/n)**

We appreciate the reviewers’ careful consideration of our submission and for their many detailed comments and suggestions to improve our work. We thank the reviewers for their patience as we’ve performed a number of new experiments and comparisons in order to highlight the flexibility and performance of ShEPhERD across ligand-based and structure-based drug design tasks. While we will respond to each reviewer separately, this comment provides an overview of the new experiments and evaluations, which we believe should be highlighted for all the reviewers. These new comparisons include:

- Comparisons to SQUID for shape-conditioned molecular generation on MOSES.
- Comparisons against structure-based drug design models that explicitly encode the protein pocket.
- Comparisons specifically against DiffSBDD.
- Comparisons against SynFormer on natural product ligand hopping.

These comparisons are reported in the new Appendices B.1, B.2, B.3, and B.4, which are included at the end of the submission file so that they can be easily found during this review period. These new sections will be better integrated into the relevant Appendices for the final version of the paper.

**Comparisons to SQUID for shape-conditioned molecular generation on MOSES**

Reviewer JQjD suggested that we compare to SQUID (Adams and Coley, ICLR 2023) to evaluate the superiority of ShEPhERD for shape-conditioned molecular design. To perform this comparison, we apply the version of ShEPhERD trained on MOSES-aq to learn P(x1,x3,x4) to generate new molecules conditioned on the surface shape (e.g., the coordinates of x3, which implicitly define x2) of the 1000 3D molecules in SQUID’s test set. Note that SQUID and ShEPhERD use the same training and test splits of MOSES (which are directly provided by the original MOSES paper), and hence there was no data leakage. We condition the P(x1,x3,x4) model on the molecules’ shapes only by inpainting just the coordinates of x3, and allowing ShEPhERD to generate x1, x4, and the electrostatic potentials of x3. We generate up to 10 samples from ShEPhERD and compare the average volumetric shape similarity using ROCS (the same shape scoring function that SQUID used). Since SQUID’s generated molecules for each test molecule are publicly available for download on their github/figshare, we re-evaluated the average shape similarity of SQUID-generated molecules to ensure a fair comparison.

Across the 1000 test-set molecules, the average shape similarity of SQUID-generated molecules is 0.70 using lambda=1.0, and 0.74 using lambda=0.3. In contrast, the average shape similarity of ShEPhERD-generated molecules is 0.80, a substantial improvement. Moreover, the ShEPhERD-generated molecules have an average 2D graph similarity of just 0.22 compared to the reference molecule, whereas the SQUID-generated molecules have average graph similarities of 0.25 (lambda=1.0) and 0.35 (lambda=0.3). Hence, ShEPhERD is able to generate less chemically similar molecules that have higher shape similarity to the target, on average.

We emphasize that ShEPhERD greatly exceeds the performance of SQUID even with this indirect way of doing shape-conditioned generation, i.e. via inpainting the joint P(x1,x3,x4) model. We could also train a P(x1,x2) model directly for this task; however, we did not have time to retrain ShEPhERD from scratch within the rebuttal time period. We also note that ShEPhERD conditions on surface shapes, whereas we have compared ShEPhERD to SQUID using volumetric shape similarity (with ROCS). We expect that we could further improve ShEPhERD’s performance in shape-conditioned generation by retraining ShEPhERD with a smaller surface probe radius, thereby “shrinking” the surface shape to better model volumetric shapes.

---

> ### Author Response · Authors · 2024-11-20
> **Common Response to all reviewers highlighting new experiments and comparisons to related work (Part 2/n)**
>
> **Comparisons against structure-based drug design models that explicitly encode the protein pocket**
>
> Reviewer HW8X suggested that we compare ShEPhERD against 3D generative models for structure-based drug design, which explicitly model protein pocket geometries. We want to emphasize upfront that ShEPhERD is a ligand-only model, and does not explicitly model the structures of protein pockets. Hence, any comparison to SBDD methods cannot be completely fair. Nevertheless, in our submission we do demonstrate that although ShEPhERD is not designed for structure-based drug design, it can still be roughly applied to SBDD by conditioning on the bound pose of a known ligand (but still ignoring the surrounding protein pocket). This bound pose can be obtained from crystallography (e.g., the co-crystal ligands in PDB entries) or from simulated docked poses (e.g., conditioning ShEPhERD on the top hit from a small docking-based virtual screen). In order to compare to SBDD generative models (which do model protein pockets), we compare ShEPhERD against the benchmarking results presented by Zheng et al. (2024), which report the average Vina docking scores amongst the top-10 generated compounds (amongst ~1000 generated samples) for 7 PDB systems using generative SBDD methods like 3DSBDD, AutoGrow4, Pocket2Mol, PocketFlow, and ResGen. Specifically, we use ShEPhERD to condition on the electrostatic potential surface (x3) and pharmacophores (x4) of the PDB ligand in its crystallographic bound pose, generating up to 1000 valid molecules per PDB target. We dock the 1000 generated molecules (starting from their SMILES strings) using the same Vina program (and version) as used by Zheng et al. (2024), and compute the average Vina scores amongst the top-10 molecules per target. The new Appendix. B.2 reports these average top-10 scores, compared against the results for the SBDD models reported in Zheng et al. (2024). We also evaluate the 2D graph similarities of ShEPhERD-generated compounds compared to the PDB ligand in each case, which are generally quite low.
>
> In summary, even though ShEPhERD only sees the PDB ligand and not the target protein pocket, the average top-10 Vina scores for ShEPhERD-generated molecules are very comparable to those obtained by these SBDD methods. Out of the 6 generative models (ShEPhERD and 5 SBDD methods), ShEPhERD ranks 1/6 for one target, 2/6 for one target, 3/6 for three targets, and 4/6 for two targets. These results confirm that ShEPhERD shows great promise for SBDD even though it is currently a ligand-only model.
>
> We also want to emphasize that while ShEPhERD performs comparably to these SBDD methods on these particular PDB targets, ShEPhERD is a far more general model as it can be applied to natural product ligand hopping, biosisosteric fragment merging, shape- and pharmacophore-conditioned scaffold hopping, etc., all in context-free environments.

---

> ### Author Response · Authors · 2024-11-20
> **Common Response to all reviewers highlighting new experiments and comparisons to related work (Part 3/n)**
>
> **Comparisons specifically against DiffSBDD.**
>
> We note that DiffSBDD was not one of the SBDD generative models evaluated in Zheng et al. (2024). However, Reviewer HW8X specifically requested that we compare against DiffSBDD, which also uses diffusion + inpainting to generate molecules, albeit in a structure-based drug design scenario where the pocket is directly encoded by the model. To compare ShEPhERD specifically against DiffSBDD, we applied ShEPhERD to generate analogs given the crystallographic bound pose for the ligand in the PDB entry 4tos, which DiffSBDD was evaluated on in Figure 2 of the DiffSBDD paper (https://arxiv.org/abs/2210.13695). To ensure a fair comparison, we downloaded the 100 molecules generated by DiffSBDD for this PDB target, which are fortunately made publicly available by the authors of DiffSBDD. We used the molecules generated by the all-atom version of DiffSBDD that uses inpainting, as this is the model that generally performs the best according to the DiffSBDD paper. We then re-docked these 100 molecules (starting from their SMILES strings) with AutoDock Vina (version 1.1.2) to obtain their distribution of docking scores. We then applied ShEPhERD to generate 100 valid molecules given (x3, x4) of the bound ligand (still ignoring the actual protein pocket), and docked those molecules, again starting from their SMILES strings.
>
> We compare the distributions of docking scores between the molecules generated by ShEPhERD and DiffSBDD in the new Appendix B.3. Even though ShEPhERD does not explicitly model the pocket structure, the average Vina score for ShEPhERD-generated molecules is approximately 2 kcal/mol lower (better) than the average Vina score for the DiffSBDD-generated molecules. Moreover, the ShEPhERD-generated molecules have significantly lower (better) SA scores, as well. Meanwhile, the ShEPhERD-generated molecules still have quite low graph similarity to the PDB ligand (vast majority are below 0.2). We also emphasize that this conditioning PDB ligand is clearly out-of-distribution for ShEPhERD, as the PDB ligand 42 non-hydrogen atoms, whereas ShEPhERD is only trained on molecules containing up to 27 non-hydrogen atoms. However, we do want to emphasize that, as previously mentioned, comparisons to SBDD methods are inherently unfair. In this case, the PDB ligand itself has a quite good docking score, which means that the ShEPhERD-generated analogs are likely to also have good docking scores, as ShEPhERD generates molecules that preserve 3D interactions. Nevertheless, this experiment shows ShEPhERD’s utility in SBDD, despite being a ligand-only model.

---

> ### Author Response · Authors · 2024-11-20
> **Common Response to all reviewers highlighting new experiments and comparisons to related work (Part 4/n, n=4)**
>
> **Comparisons against SynFormer on natural product ligand hopping.**
>
> Reviewer HW8X also requested that we compare against SynFormer in our ligand hopping experiments. As a reminder, the goal of our natural product ligand hopping experiments was to generate small-molecule analogs of structurally complex natural products that (1) preserve their electrostatic and pharmacophore 3D interactions, and (2) have simpler chemical structures as assessed via SA score. SynFormer is a non-3D generative model that can be used to generate synthesizable small-molecule analogs of an unsynthesizable reference molecule that have similar 2D chemical structures. We applied SynFormer using its publicly available code in its default settings to generate analogs for each of the three natural products. We took each of the generated analogs, generated up to 10 xTB-optimized conformers, and scored their 3D similarities to the reference natural products after optimal alignment.
>
> We compare the SynFormer-generated analogs against the ShEPhERD-generated analogs in the new Appendix B.4 on the basis of electrostatic and pharmacophore similarity to the reference natural products. In all cases, SynFormer vastly underperforms ShEPhERD, which is not surprising since SynFormer was not designed to preserve 3D interaction similarity. We also note that very few (if any) of the SynFormer-generated molecules have SA scores below the threshold we enforce on ShEPhERD (SA<4.5), indicating that SynFormer’s analogs are generally more complex, even if purportedly synthesizable.
>
>
> **Summary**
>
> We again thank the reviewers for suggesting these new experiments, as we believe they have substantially improved our submission by placing the performance of ShEPhERD in context with existing models in the literature that perform related, if not always directly comparable, tasks. We are happy to answer any questions regarding these additional experiments.

---

### Meta-Review · Area_Chair_boFK · 2024-12-22

**Metareview:**

In this work, authors introduce ShEPhERD, an SE(3)-equivariant diffusion model for 3D molecular generation that jointly models molecular graphs, shapes, electrostatic surfaces, and pharmacophores. The model enables conditional generation of molecules with desired 3D interaction properties through inpainting, facilitating tasks like natural product ligand hopping and bioisosteric fragment merging.

The main scientific contribution is demonstrating that a joint diffusion model over multiple molecular representations enables generating novel molecules that preserve key 3D interaction patterns of reference compounds. The authors validate this through experiments showing ShEPhERD can: 1) generate molecules with similar shapes to references while maintaining low graph similarity, outperforming SQUID as shown in additional experiments, 2) generate bioisosteric analogs of natural products that maintain interaction patterns while having lower synthetic accessibility scores, and 3) merge fragments based on their pharmacophoric interactions.

The reviewers identified several key strengths. Reviewer HW8X highlighted the thoughtful implementation of pharmacophore conditioning and practical utility for drug discovery applications. Reviewer pbc2 noted the strong logical foundation and clear exposition. The extensive comparisons to baselines like REINVENT and response to reviewer requests for additional comparisons to methods like SQUID, DiffSBDD, and SynFormer demonstrate robust performance.

Some limitations were raised, particularly around computational efficiency and scaling to larger molecules, as noted by Reviewer JQjD. However, the authors acknowledge these engineering challenges as future work, appropriately focusing first on establishing the method's utility. Questions about dataset splits and evaluation protocols were addressed through additional experiments showing good performance on challenging out-of-distribution tasks.

I recommend acceptance for several reasons: First, the paper makes a clear technical contribution by introducing a novel joint modeling approach for 3D molecular properties. Second, the extensive experimental validation, including new comparisons added during discussion, demonstrates practical utility across multiple drug design tasks. Third, the method enables important capabilities not possible with existing approaches, particularly around preserving 3D interactions in ligand-based design. While there are opportunities for improving efficiency and scaling, these do not diminish the core methodological advance. The thorough author response and additional experiments effectively addressed the key reviewer concerns about comparisons and evaluation protocols.

In summary, this work represents an important contribution to molecular generation methods that will benefit both the machine learning and drug discovery communities. The clear exposition, comprehensive evaluation, and demonstrated practical utility make it worthy of presentation at ICLR.

**Additional Comments On Reviewer Discussion:**

Some of the key aspects that emerged during the rebuttal period are as follows. First, multiple reviewers requested additional comparisons to related methods. The authors conducted extensive new experiments comparing ShEPhERD to SQUID for shape-conditioned generation, DiffSBDD and other models for structure-based design, and SynFormer for ligand hopping. These comparisons demonstrated ShEPhERD's superior performance in shape-based generation and competitive performance even against specialized structure-based models.

Second, pbc2 raised concerns about dataset splits and generalization capabilities. The authors clarified their rationale for using random splits in basic evaluations while demonstrating generalization through challenging out-of-distribution tasks like natural product hopping. The hit diversification experiments provided additional evidence of generalization to moderately larger molecules.

Third, JQjD questioned computational efficiency and evaluation metrics. The authors acknowledged efficiency limitations but prioritized establishing method utility first. They added 2D similarity analyses showing generated molecules are distinct from references, addressing concerns about novelty.

The discussion was particularly productive in establishing ShEPhERD's capabilities relative to existing methods. Notably, HW8x substantially increased their score after reviewing the comprehensive experimental additions. The authors' thorough responses and willingness to conduct extensive new experiments strengthened the paper's evaluation significantly.

---

### Decision · Program_Chairs · 2025-01-22

Accept (Oral)